Technical Report

# Uncovering protein glycosylation dynamics and heterogeneity using deep quantitative glycoprofiling (DQGlyco)

Clément M. Potel [1,5,6] ✉, Mira Lea Burtscher[1,2,5], Martin Garrido-Rodriguez[1,3,5], Amber Brauer-Nikonow[4], Isabelle Becher[1], Cecile Le Sueur[1], Athanasios Typas [1], Michael Zimmermann [4] & Mikhail M. Savitski [1,6] ✉

Protein glycosylation regulates essential cellular processes such as signaling, adhesion and cell–cell interactions; however, dysregulated glycosylation is associated with diseases such as cancer. Here we introduce deep quantitative glycoprofiling (DQGlyco), a robust method that integrates high-throughput sample preparation, highly sensitive detection and precise multiplexed quantification to investigate protein glycosylation dynamics at an unprecedented depth. Using DQGlyco, we profiled the mouse brain glycoproteome, identifying 177,198 unique *N*-glycopeptides—25 times more than previous studies. We quantified glycopeptide changes in human cells treated with a fucosylation inhibitor and characterized surface-exposed glycoforms. Furthermore, we analyzed tissue-specific glycosylation patterns in mice and demonstrated that a defined gut microbiota substantially remodels the mouse brain glycoproteome, shedding light on the link between the gut microbiome and brain protein functions. Additionally, we developed a novel strategy to evaluate glycoform solubility, offering new insights into their biophysical properties. Overall, the in-depth profiling offered by DQGlyco uncovered extensive complexity in glycosylation regulation.

Protein glycosylation involves the covalent linkage of sugar molecules to amino acid side chains. About one third of human proteins go through the secretory pathway, wherein a vast majority are *N*-glycosylated on asparagine side chains in the endoplasmic reticulum (ER), before further glycan modification in the Golgi apparatus[1,2]. Over 200 different glycosylation enzymes can act sequentially to create glycans of heterogeneous composition and structure[2], meaning that, in addition to glycosylation macroheterogeneity (glycosylation at different sites), glycosylation microheterogeneity accounts for the variety of glycans that can be attached to the same site. Although the regulatory role of glycosylation microheterogeneity is a relatively recent concept, it is now acknowledged that the alteration of protein glycoforms can impact protein functions by modulating protein–protein or protein–ligand interactions[3].

Protein glycosylation regulates key biological processes including cell signaling, cell adhesion or endocytosis[1–4] and its deregulation has been linked to many pathologies, including cancer and neuronal disorders[5,6]. A major obstacle in our understanding of

[1]Genome Biology Unit, European Molecular Biology Laboratory, Heidelberg, Germany. [2]Faculty of Biosciences, Heidelberg University, Heidelberg, Germany. [3]Institute for Computational Biomedicine, Heidelberg University and Heidelberg University Hospital, Heidelberg, Germany. [4]Structural and Computational Biology Unit, European Molecular Biology Laboratory, Heidelberg, Germany. [5]These authors contributed equally: Clément M. Potel, Mira Lea Burtscher, Martin Garrido-Rodriguez. [6]These authors jointly supervised this work: Clément M Potel, Mikhail M Savitski. ✉e-mail: clement.potel@embl.de; mikhail.savitski@embl.de

protein glycosylation has been the lack of highly selective enrichment strategies of the glycoproteome and high-throughput quantitative approaches for precise glycosylation measurement. Various methods have been developed to enrich glycopeptides from protein digests[7], including lectin affinity chromatography[8] and hydrophilic interaction chromatography[9]. While highly useful, these methods have technical limitations, including biases toward certain types of glycan compositions or structures and low enrichment specificities.

Chemical coupling of glycopeptides to beads functionalized with reactive groups generally leverages the reversible reaction between phenylboronic acid (PBA) derivatives and diols present in sugar molecules. This results in the formation of a covalent bond between functionalized beads and glycopeptides at high pH, followed by elution of the glycopeptides at low pH. In principle, this approach offers two key advantages: the glycopeptide–bead covalent linkage facilitates stringent washes to increase glycopeptide enrichment specificity and it allows unbiased enrichment because nearly all glycopeptides contain reactive diol groups. However, chemical coupling has historically identified fewer glycopeptides than alternative enrichment techniques and often requires in-house preparation of functionalized beads. The use of binding buffers containing a low nucleophilic base was recently shown to enhance the capture of glycopeptides using beads functionalized with a PBA derivative[10], suggesting that PBA could compete with other methods for glycopeptide analysis.

Here, we developed an enrichment and prefractionation strategy that substantially increases the coverage of the glycoproteome, integrated with a multiplexed and cost-effective quantification strategy. In the mouse brain, we identified a total of 177,198 unique *N*-glycopeptides, representing a 25-fold improvement over previous state-of-the-art glycoproteomics studies. We quantified approximately 50,000 unique *N*-glycopeptides per experiment in mouse tissues, achieving a more than tenfold increase in glycoproteome coverage when compared to previous large-scale studies, allowing us to quantify, on average, ten glycoforms per site. This coverage enabled a detailed exploration of site microheterogeneity alterations in response to perturbations. Our improved methodology facilitated the examination of protein structural features dictating glycosite microheterogeneity and maturation processes. By treating intact living human cells with two enzymes, we could identify surface-exposed, mature glycoforms. When profiling the glycoproteome across three mouse tissues, we discovered that glycosylation tissue specificity is driven by a subset of glycosites exhibiting extensive differences in their glycosylation patterns. These glycosites are enriched in functional domains involved in immunity, adhesion and cell signaling. We quantified glycosylation dynamics in human cells treated with a fucosylation inhibitor and in the brains of mice colonized with defined gut microbiomes.

The composition of the gut microbiome is known to affect behavior and brain physiology but the molecular mechanisms remain unclear. We observed substantial alterations in protein glycoform abundance on proteins involved in axon guidance or neurotransmission. Strikingly, this in-depth quantitative profiling of glycosylation upon perturbations revealed widespread site-specific modulation of glycoforms on glycosites belonging to the same proteins, thus indicating that mechanisms of glycosylation regulation are more complex than previously appreciated. Lastly, we present an approach that enables the systematic biophysical characterization of protein glycosylation. This revealed widespread solubility differences among glycoforms at the same glycosylation sites, suggesting that glycosylation microheterogeneity has a proteome-wide impact on protein states in vivo. All results and data can be accessed through an interactive data visualization application.

## Results

### Sensitive and specific enrichment of protein glycosylation
We developed a glycoproteomics workflow (Fig. 1a) using commercially available, economical silica beads functionalized with PBA to selectively enrich intact glycopeptides. All steps of the workflow, from sample preparation to glycopeptide enrichment, were performed in 96-well (filter) plates, increasing sample processing throughput. In addition to glycopeptides, other diol-containing biomolecules, such as RNA, can also bind to PBA beads, interfering with the detection of glycopeptides. This is illustrated by the high abundance of an RNA marker ion (330.06 *m/z*)[11] in glycopeptide-enriched samples after 2% strong denaturing detergent (SDS) lysis followed by protein precipitation (Supplementary Fig. 1a). To address this, we optimized the sample lysis buffer by incorporating a high concentration of chaotropic salts and organic solvent to induce nucleic acid precipitation while proteins remained in solution. Nucleic acid aggregates were then filtered out on 96-well filter plates, followed by protein precipitation by further increasing organic solvent concentration before tryptic enzymatic digestion. This fast sample preparation workflow (1 h for 96 samples) enabled efficient removal of RNA molecules (Supplementary Fig. 1a), increasing the number of unique *N*-glycopeptides identified by 60% (Supplementary Fig. 1b and Supplementary Data 1).

Next, we adjusted the full scan (first-level mass spectrometry (MS1)) range to preferentially target glycopeptides that are higher in mass compared to nonmodified peptides that nonspecifically bind to PBA beads (Supplementary Fig. 1c). This increased the number of identified unique *N*-glycopeptides by 18% and the enrichment specificity of *N*-glycopeptides by 13% compared to a commonly used mass over charge scan range (Supplementary Fig. 1d,e and Supplementary Data 1). In total, using MSFragger[12] as a search engine, we were able to identify an average of 10,294 unique glycopeptides, 1,746 glycosites and 774 glycoproteins in human cell lines (HeLa and HEK293T; 1% false discovery rate (FDR) at the *N*-glycopeptide level, glycan *q* values ≤ 0.05) per single-shot replicate. These numbers increased to 16,090 unique glycopeptides, 2,431 glycosites and 1,057 glycoproteins in mouse brain samples (Fig. 1b). Overall, deep quantitative glycoprofiling (DQGlyco) enables high-throughput sample preparation and enrichment of hundreds of samples per day. Moreover, DQGlyco markedly improves the detected glycoproteome coverage when compared to previous studies (Supplementary Fig. 1f) without the need for prefractionation. Remarkably, the enrichment selectivity exceeded 90% for all samples (Fig. 1c and Supplementary Data 1).

### Deep mouse brain glycoproteome analysis
To increase the glycoproteome coverage in mouse brain samples, we used porous graphitic carbon (PGC) as a first dimension of chromatographic separation for glycopeptides before a second online C18-based reversed-phase separation. Leveraging a mixed-mode retention mechanism[13], PGC can more efficiently resolve different glycan species compared to traditional reversed-phase chromatography[14,15]; thus, as a first dimension of separation, PGC should facilitate a better capture of glycosylation microheterogeneity. Our approach identified 177,198 unique *N*-glycopeptides and 8,245 *N*-glycosites located on 3,741 *N*-glycoproteins (Fig. 1d and Supplementary Data 2). This represents a greater than 25-fold improvement in the identification of unique *N*-glycopeptides compared to current state-of-the-art *N*-glycoproteomics studies on mouse brain samples[16,17] (Supplementary Fig. 1f). Notably, this improvement persists when searching previous datasets with our analytical pipeline (Supplementary Fig. 1g) or searching one of our datasets with another search engine (Byonic[18]; Supplementary Fig. 1h).

The majority of identified glycoforms were on annotated plasma membrane or extracellular proteins (Supplementary Fig. 1i). No differences in glycopeptide molecular weight and peptide length were observed between the single-shot and fractionated samples, indicating that PGC fractionation does not introduce a bias for the detection of shorter, more readily eluted glycopeptides (Supplementary Fig. 1j).

Our dataset encompasses 65% of the proteins that have been previously identified in various regions of the mouse brain[19] and annotated

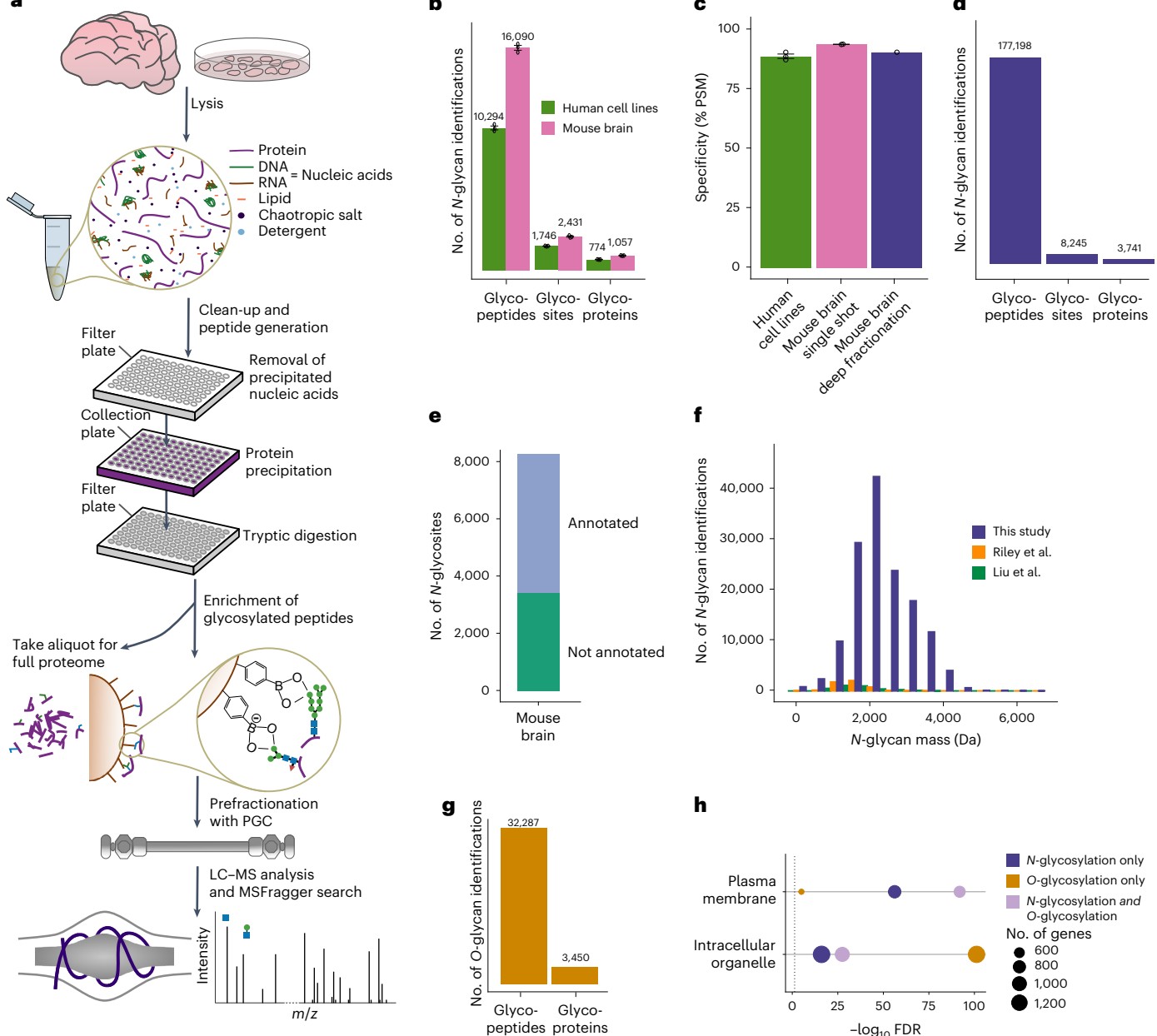

**Fig. 1 | Deep profiling of protein glycosylation in human cell lines and mouse brain. a**, DQGlyco workflow. **b**, Average numbers of unique glycopeptides, glycosites and glycoproteins identified from LC–MS/MS triplicates of unfractionated samples from human cell lines (HelaK and HEK293T cells) and mouse brain. Data are presented as the mean values ± s.d. **c**, Glycopeptide enrichment specificity, calculated as the ratio of glycopeptide spectrum matches to all peptide spectrum matches (PSM) for unfractionated samples from human cell lines (HeLaK and HEK293T; $n = 3$ technical replicates per cell line) and mouse brain ($n = 3$ technical replicates), as well as from PGC-fractionated mouse brain samples. Data are presented as the mean values ± s.d. **d**, Total number of identified $N$-glycopeptides (unique sequence and glycan

composition), $N$-glycosites and $N$-glycoproteins from PGC-fractionated mouse brain samples. **e**, UniProt and PhosphoSitePlus annotation of $N$-glycosites identified in mouse brain samples. **f**, Total number of unique glycopeptides of the PGC-fractionated mouse brain sample per binned $m/z$ range compared to recent large-scale glycoproteomics studies (Riley et al.[16] and Liu et al.[17]). **g**, Total number of $O$-glycopeptides and $O$-glycoproteins identified in the PGC-fractionated mouse brain sample. **h**, GO enrichment analysis (STRINGdb) for proteins found to be only $N$-glycosylated, only $O$-glycosylated or both $N$-glycosated and $O$-glycosylated for two selected terms (intracellular organelle and plasma membrane).

as glycosylated in UniProt (Supplementary Fig. 2a). Overall, around half of the glycosites and glycoproteins identified in this study were reported as glycosylated in the UniProt and PhosphoSitePlus databases, with only 6% having experimental validation evidence (Fig. 1e and Supplementary Fig. 2b). Thus, this study substantially expands the known mouse brain $N$-glycoproteome. Proteins reported as glycosylated in UniProt are predominantly annotated as part of the plasma membrane while previously unreported glycosylated proteins are

mainly associated with subcellular organelles (Supplementary Fig. 2c). The identification of 332 glycoproteins in our dataset that were not found in the mouse proteome atlas[19] study (Supplementary Fig. 2d) highlights the sensitivity of our method as these proteins are likely expressed at low levels. Typically, glycopeptide enrichment methods exhibit biases toward certain glycan compositions. We compared the different glycan compositions or masses identified by alternative enrichment strategies and found that $N$-glycans identified by our

PBA-based approach encompass those previously identified by different approaches (Fig. 1f and Supplementary Fig. 2e) while greatly improving the glycoproteome coverage, suggesting a low enrichment bias toward specific glycopeptides.

Understanding the stoichiometry of different glycoforms is important for interpreting the biological implications of glycoform alterations. In this study, we can only perform relative quantification as glycan composition can impact glycopeptide ionization efficiency (Supplementary Fig. 2f). Using MS1 intensities as a proxy to calculate glycoform fractional intensity (ratio of glycoform intensity to the sum of all glycoform intensities at a given site) revealed that most glycoforms exhibit a low fractional intensity (Supplementary Fig. 2g). When focusing on the glycoform having the highest fractional intensity per site (here, only sites with at least ten glycoforms were considered, representing 1,580 sites), it becomes clear that glycosylation of a specific site is rarely dominated by a single glycoform of high stoichiometry (Supplementary Fig. 2h).

*N*-glycopeptides are more abundant than *O*-glycopeptides in mammalian cells; hence, *O*-glycopeptide identification usually necessitates dedicated analytical strategies[7]. We reasoned that the deep glycoproteome coverage obtained by DQGlyco could enable the simultaneous identification of both *N*-glycopeptides and *O*-glycopeptides. In total, we identified 32,287 *O*-glycopeptides on 3,450 *O*-glycoproteins in mouse brain samples (Fig. 1g and Supplementary Data 2), constituting an extensive characterization of intact *O*-glycosylation in a mammalian tissue. It has to be noted that, as higher-energy collisional dissociation (HCD) fragmentation was used in this study, we could not confidently assign the site of glycosylation within a given glycopeptide[20]. We could identify over 494 unique *O*-glycan compositions, with a molecular weight distribution shifted to lower masses when compared to *N*-glycans (Supplementary Fig. 2i). A total of 1,853 proteins were only *O*-glycosylated and not *N*-glycosylated. Proteins that were solely *O*-glycosylated were enriched in intracellular membrane-bound organelles (Fig. 1h). In accordance with previous reports, *O*-linked *N*-acetylhexosamine (*O*-HexNAc) modification was enriched in intracellular organelles, whereas no difference in glycan elongation was observed between proteins belonging to the plasma membrane and to intracellular organelles (Supplementary Fig. 2i). In addition, using oxonium ion ratios for analysis[21], we observed that *O*-glycans on proteins annotated as intracellular were predominantly modified by *N*-acetylglucosamine (GlcNAc), while *O*-glycans on proteins annotated as extracellular or belonging to the plasma membrane were predominantly modified by *N*-acetylgalactosamine (GalNAc)[22] (Supplementary Fig. 2j). Conversely, *N*-glycans were predominantly modified by GlcNAc, regardless of the protein subcellular localization (Supplementary Fig. 2k).

## DQGlyco reveals extensive microheterogeneity

Although half of the sites identified were modified by one or two glycoforms, numerous sites exhibited high microheterogeneity, with an average of 17.4 glycoforms per site (Fig. 2a) and extreme cases such as the N205 site on the excitatory amino acid transporter 2 protein (Eaat2, also referred to as Slc1a2), on which 667 glycoforms were identified. No correlation was observed between the detected number of glycosites per protein or number of glycoforms per site and the abundance of the respective protein (Supplementary Fig. 3a,b). Similarly, no major differences in isoelectric point and peptide length were observed for glycopeptides bearing sites of different microheterogeneity (Supplementary Fig. 3c,d). This suggests that the glycosylation heterogeneity observed is not skewed toward easily detected or abundant glycoproteins and glycopeptides.

The different glycan compositions were grouped into eight classes, which can have varying impacts on protein–protein or protein–ligand interactions through steric or electrostatic effects and can serve as epitopes for specific lectin receptors[1–4]. *N*-glycans are added in the ER as high mannose and can be processed in the Golgi into glycans classified as high mannose, small, paucimannose, phosphorylated, fucosylated and sialylated, while the remaining compositions were considered as complex or hybrid (Fig. 2b and Supplementary Fig. 3e,f; details in Methods). Compared to the Riley et al. dataset[16], we identified markedly less high-mannose *N*-glycosylation (fewer than 15% of all glycopeptides identified versus 60%). This confirms that the commonly used conA lectin-based enrichment is biased toward high-mannose glycosylation and underscores the need for deep analytical coverage to achieve comprehensive characterization of mature glycoforms (Fig. 2c). In this study, close to half of the sites modified by only one glycoform were found to be modified by high mannose, implying that these sites are indeed subjected to a low number of processing events (Supplementary Fig. 3g).

The distribution of different classes of *N*-glycosylation differs between the two human cell lines and the mouse brain (Supplementary Fig. 3h), likely reflecting differences in the regulation of glycosylation processes. Despite the fact that little difference in enriched Gene Ontology (GO) terms was observed between the different glycan classes (Supplementary Fig. 3i), meaning that different types of glycosylation in general do not target proteins with specific functions, significant distinctions were observed in the enriched protein domains for the different *N*-glycosylation classes (Fig. 2d). Na$^+$/K$^+$-transporting adenosine triphosphatase domains were, for example, strongly enriched in sialylated and paucimannose glycans, while saposin domains (specific to lysosomal proteins) were enriched for phosphoglycans. In general, phosphoglycans showed strong enrichment on lysosomal proteins (Supplementary Fig. 3j), aligning with the role of mannose-6-phosphate as a tag directing enzymes to the lysosome[23].

As the different sites of *N*-glycoproteins should be exposed to the same glycoenzymes, we investigated if microheterogeneity across different sites within the same protein was more similar than across the glycoproteome. We observed no clear pattern of co-occurrence of different classes of *N*-glycosylation either at the glycosite or at the glycoprotein level (Supplementary Fig. 4a,b). However, when correlating the exact *N*-glycoform compositions between sites localized on the same protein before and after random reshuffling of all glycoforms across all identified sites, we concluded that, while there is substantial variation in microheterogeneity between different sites on the same protein, this variation is significantly lower than across the glycoproteome as a whole (Fig. 2e).

Differences were observed when classifying *N*-glycoproteins according to the level of site microheterogeneity, independently of glycan compositions; sites with high heterogeneity were enriched on extracellular proteins, involved in processes such as cell-surface receptor signaling pathways and cell adhesion, both known to be modulated by glycosylation[2] (Fig. 2f). Interestingly, the level of site microheterogeneity within a given glycoprotein tended to be site specific (Supplementary Fig. 4c).

No specific protein annotation was associated with a class of *O*-glycosylation, with intracellular or plasma membrane *O*-glycoproteins exhibiting relatively similar glycosylation patterns (Supplementary Fig. 4d). However, *O*-glycopeptides on which only one glycoform was identified were predominantly modified by a single HexNAc sugar (Supplementary Fig. 4e) and *O*-HexNAc modification occurred mostly on intracellular proteins (Supplementary Fig. 4f). Similarly, proteins with only one identified *O*-glycoform were enriched in intracellular organelle proteins while proteins with higher heterogeneity were enriched in membrane proteins involved in signaling (Supplementary Fig. 4g). It has been reported that *O*-GlcNAcylation (addition of a single sugar) and phosphorylation have a propensity to co-occur on the same protein regions, resulting in posttranslational modification (PTM) crosstalk[24]. Here, we show that around 25% of *O*-glycosylated peptides were also previously reported to be phosphorylated, independently of the glycan mass and composition, suggesting

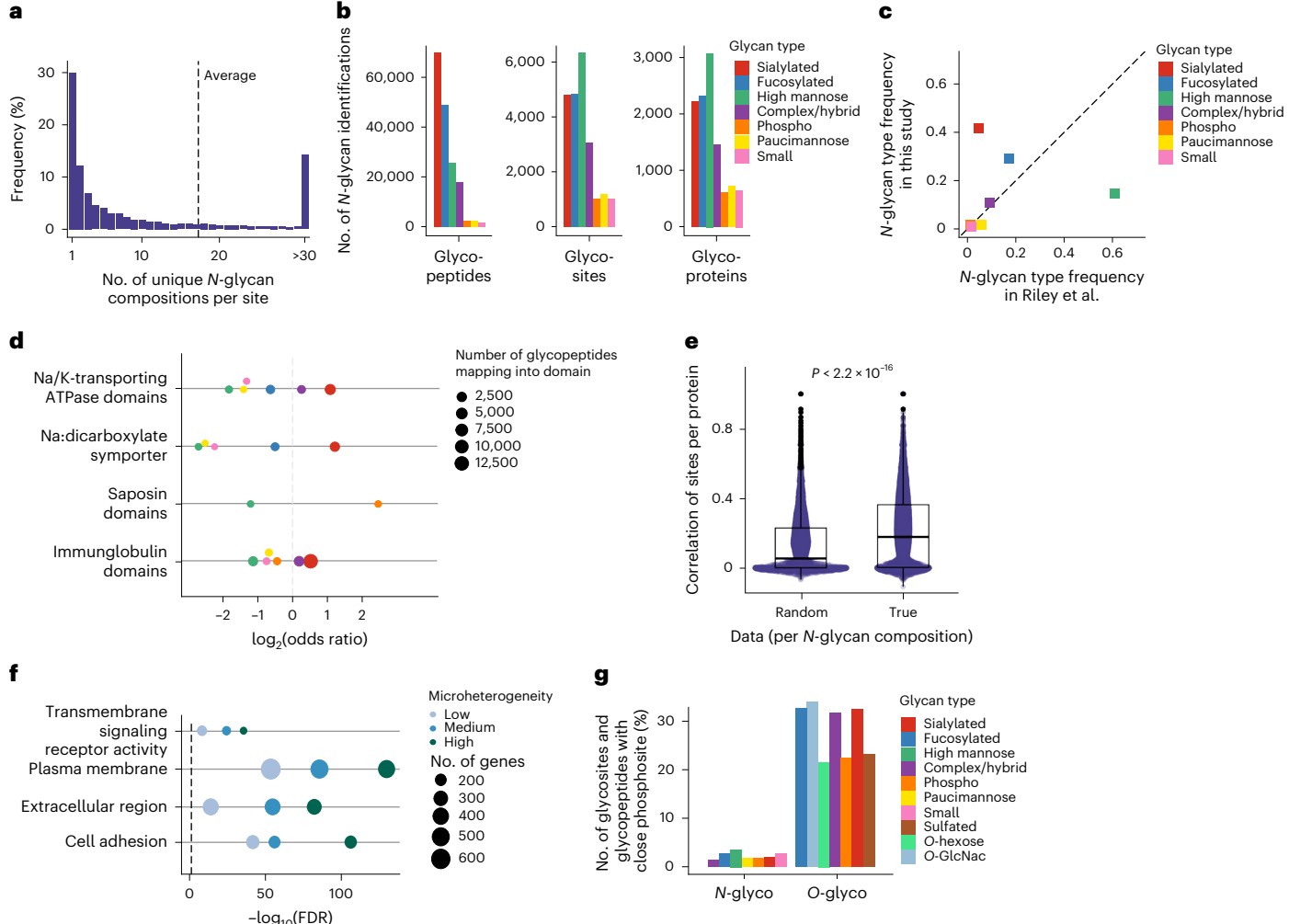

**Fig. 2 | Protein glycosylation composition and microheterogeneity in the mouse brain. a,** Frequency of the number of unique glycan compositions per site in the PGC-fractionated mouse brain $N$-glycosylation data (on average, 18 glycan compositions were identified per site). **b,** Total number of unique glycopeptides, glycosites and glycoproteins per glycan class for the PGC-fractionated mouse brain sample. **c,** Frequencies of glycan classes identified in this study and in a recent glycoproteomics study using lectin-based enrichment[16]. **d,** Distinct glycan classes are significantly overrepresented or underrepresented on specific functional protein domains from the InterPro database (log$_2$ odds ratio > 0 or log$_2$ odds ratio < 0, Fisher's exact test, adjusted $P$ value < 1 × 10$^{-3}$; examples shown). **e,** Glycosylation profiles of sites belonging to the same protein are significantly more correlated compared to when glycan compositions are shuffled across

the whole glycoproteome (Kendall rank correlation coefficient for $n$ = 29,932 site pairs per distribution, two-sided Wilcoxon rank-sum test, $P < 2.2 × 10^{-16}$). Box plots indicate the median and the first and third quartiles. Whiskers extend from the hinges to the largest value no further than 1.5 × the interquartile range. Data points beyond the end of the whiskers are plotted individually. **f,** Results of the GO enrichment analysis (STRINGdb) of proteins with sites displaying low, medium or high microheterogeneity ($n$ = 1–2, 3–11 and 11+ unique glycan compositions per site, respectively) for selected significant terms (two-sided Fisher's exact test, FDR < 0.05). **g,** Frequency of glycosites or glycopeptides with a close known phosphosite (±5 residues) per glycan class for $O$-glycosylation and $N$-glycosylation data.

that the existence of crosstalk can be extended to all glycoforms on $O$-glycosites. In comparison, fewer than 2% of $N$-glycosites were in close proximity to reported phosphosites (Fig. 2g).

## Machine learning prediction of site microheterogeneity

Using AlphaFold predictions[25], we extracted structural variables for glycosylated asparagine residues, including prediction-aware part-sphere exposure (pPSE), which calculates how many amino acid residues are in the structural vicinity of each site while accounting for AlphaFold prediction error. Additionally, AlphaFold predictions were used to annotate each glycosite with its predicted secondary structure[25,26]. We used this information to define intrinsically disordered regions (IDRs), while topological domain annotations were retrieved from UniProt. Using glycan classes to group the sites in different categories, we found little differences between sites, except for those where we only detected high-mannose glycans, which tend to be located in less

structured regions (Supplementary Fig. 5a). We then investigated the variability of structural features after grouping sites on the basis of their level of microheterogeneity, revealing that sites of high microheterogeneity tended to be located in more structured regions, bends and turns but not in α-helices (Supplementary Fig. 5b). These observations are exemplified on the adenovirus receptor protein CXADR (Fig. 3a,b).

Using these variables as input, we built a model to discriminate between sites of low and high microheterogeneity (Methods). Assuming that a structural signal would be conserved between mouse and human, we fitted our model on sites identified in the mouse brain glycoproteome and validated it using data from human cell lines. This model achieved a validation area under the receiver operating characteristic curve (AUROC) of 0.707 (Fig. 3c). The feature importance analysis revealed that pPSE (24 Å, 180°) and pPSE (12 Å, 70°) were the most relevant features for the predictive power of the model (Fig. 3d). As both variables directly inform about the proportion of residues

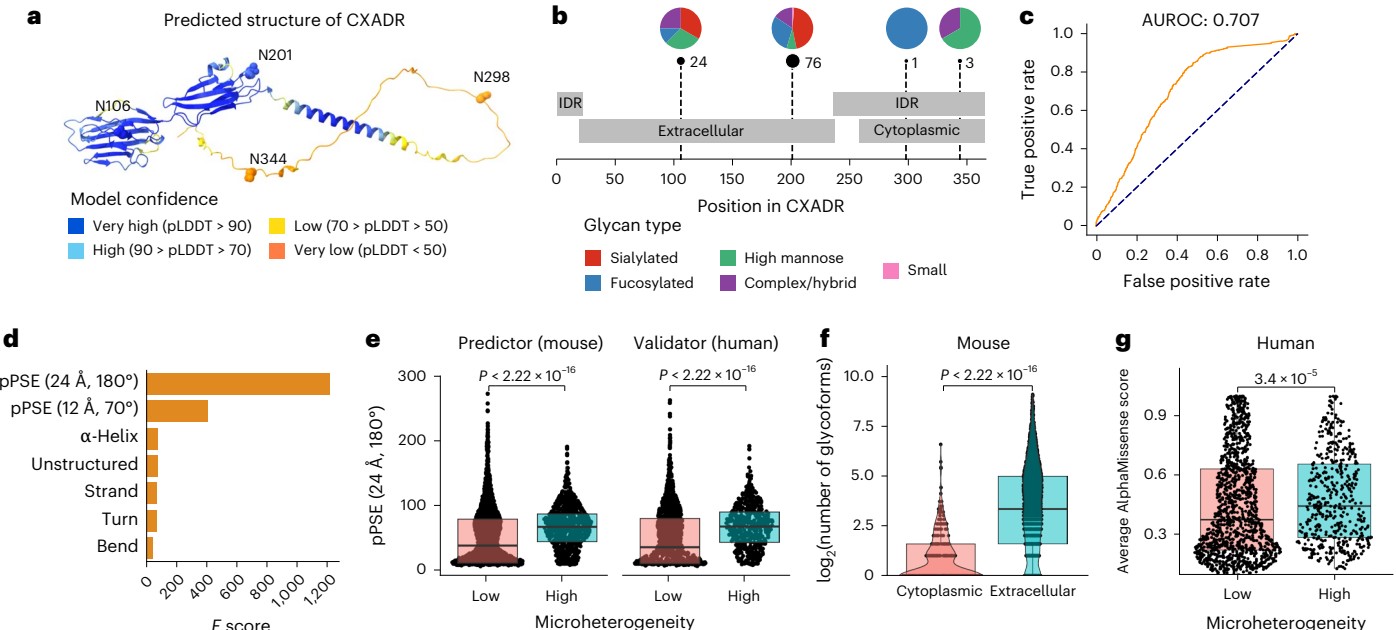

**Fig. 3 | N-glycosylation in a structural context. a**, Glycosylated asparagines highlighted as spheres in the AlphaFold-predicted structure of the coxsackievirus and adenovirus receptor (P78310). Colors indicate the model predicted local distance difference test (pLDDT). **b**, Total number of identified glycans and relative glycan type composition per site in the same protein. Topological domains and IDRs are also highlighted (IDR definition in Methods). **c**, Validation AUROC for the predictive model of lowly and highly glycosylated asparagines that uses structural features as input (details in Methods; low-$N = 2,203$ and high-$N = 747$ in mouse; low-$N = 1,196$ and high-$N = 488$ in human). **d**, Feature importance plot. The x axis indicates the F score for every feature

on the y axis. **e**, pPSE (24 Å, 180°) for highly and lowly glycosylated asparagines (low-$N = 2,203$ and high-$N = 747$ in mouse; low-$N = 1,196$ and high-$N = 488$ in human). **f**, Log-transformed number of glycoforms on asparagines within extracellular ($N = 2,813$) or cytoplasmic ($N = 335$) topological domains. **g**, Average AlphaMissense score for lowly ($N = 1,196$) and highly ($N = 488$) glycosylated asparagines on data from human cell lines. All box plots indicate the median and the first and third quartiles. Whiskers extend from the hinges to the largest value no further than $1.5 \times$ the interquartile range. Data points beyond the end of the whiskers are plotted individually. All P values indicate the comparison of distributions using a two-sided Wilcoxon rank-sum test.

surrounding glycosites, we found that sites of high microheterogeneity tended to be localized in more structured regions compared to sites exhibiting low microheterogeneity (Fig. 3e). Interestingly, when focusing on accessibility at non-IDRs, we found the inverse pattern, with sites of high microheterogeneity being more exposed (Supplementary Fig. 5c). This suggests that specific ordered structures combined with high site solvent accessibility are needed for extensive glycan processing by glycosyltransferases.

As expected, we found that extracellular glycosites exhibited a larger extent of glycan microheterogeneity than their cytoplasmic counterparts on the same proteins, in both mice (Fig. 3f) and human (Supplementary Fig. 5d). The above observations are exemplified by the detected glycosylation pattern on the adenovirus receptor protein CXADR (Fig. 3a). Lastly, we leveraged the recently published AlphaMissense model, which predicts the effect of missense variants in the human genome[27]. We found that asparagine residues exhibiting high microheterogeneity had, on average, a higher AlphaMissense score than low microheterogeneity sites, indicating their relevance for normal protein function and their potential pathogenicity when substituted by other residues (Fig. 3g).

## Deep multiplexed quantification of glycoforms

To unravel mechanisms governing glycosylation regulation, robust and high-throughput quantification methods are required. So far, most of the quantitative glycoproteomics studies have relied on label-free approaches, which come with inherent limitations as these strategies are incompatible with the different prefractionation techniques that improve analytical depth. Alternatively, tandem mass tag (TMT) labeling, enabling multiplexing, compatibility with prefractionation and low number of missing values, has been applied in glycoproteomics[28–30]. However, the prohibitive cost of TMT reagents has hindered its routine

adoption, as pre-enrichment labeling, where both the glycopeptides and the nonmodified peptides are labeled together, is recommended for accurate quantification. Here, we showcase the feasibility of decreasing the number of TMT reagents used 200-fold through postenrichment labeling, as only the substoichiometric glycopeptides need to be labeled. This cost-effective quantification method allows for the multiplexing of up to 18 conditions with high quantitative reproducibility, as shown in Figs. 4c and 5b and Supplementary Data 4 (average correlation > 0.984 between biological replicates of different mouse tissue samples (liver, kidney and brain) at the glycopeptide level). This underscores the high reproducibility achieved through our sample preparation workflow and enrichment platform. Moreover, this highlights that the glycoproteome is tightly regulated and conserved between biological replicates (see Fig. 7b), in contrast to other tissues such as plasma, for which high interindividual differences were reported[31].

## Fucosylation inhibition reveals glycoform-specific kinetics

We first used our quantitative workflow to study the inhibition of fucosylation over time following treatment with 2-fluorofucose (2FF; Fig. 4a). 2FF treatment leads to a general suppression of protein fucosylation, inhibiting several biological processes such as tumor growth, adhesion and metastasis[32]. The profiling of the glycoproteome over 3 days of treatment of HEK293f cells revealed a gradual decrease in fucosylation, alongside extensive remodeling across all glycan classes, while no global effect on protein expression was observed (Fig. 4b–d, Supplementary Fig. 6a–c and Supplementary Data 3). Notably, the downregulation of fucosylation occurred at varying rates, which we grouped into three main clusters (Fig. 4e), revealing a widespread site-specific and glycoform-specific modulation. This is illustrated by fucosylated glycoforms of the same protein belonging to different downregulated

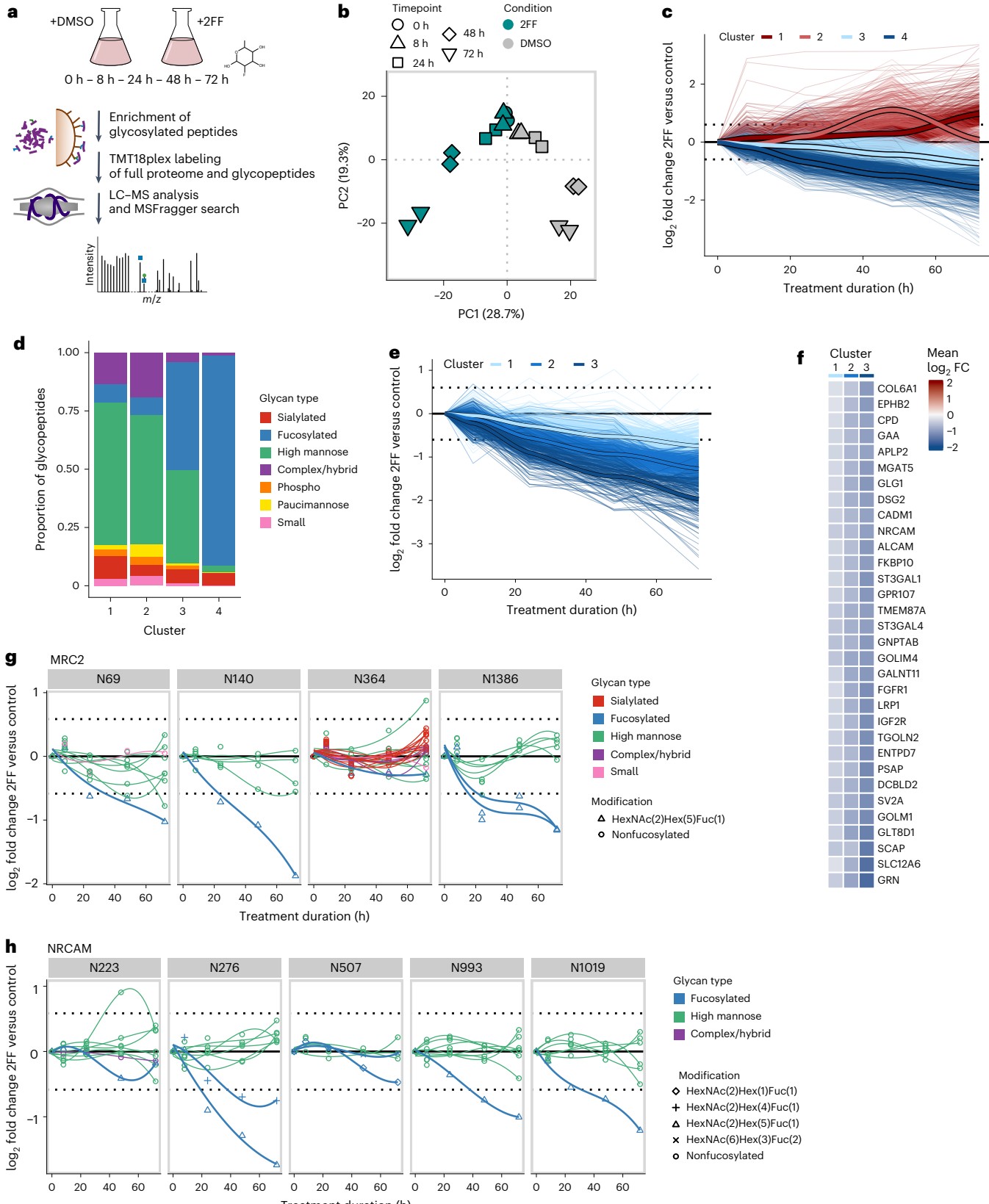

**Fig. 4 | Time course of fucosylation inhibition. a**, Experimental design. **b**, PCA of the normalized TMT glycopeptide intensities per treatment condition, time point and replicate. PC, principal component. **c**, Glycopeptide modulation over time. Glycopeptides significantly regulated for at least one time point were grouped in four clusters depending on their kinetics of regulation using neural gas clustering. **d**, Proportion of glycan classes present in the four clusters. **e**, Fucosylated peptides being significantly downregulated for at least one time point were grouped into three clusters using neural gas clustering. **f**, Glycoproteins having fucosylated glycoforms belonging to the three clusters. Here, the mean log₂ fold changes of the fucosylated glycopeptides belonging to each cluster are displayed. FC, fold change. **g,h**, The fucosylated glycoforms of the C-type mannose receptor 2 (MRC2) and the neuronal cell adhesion molecule (NRCAM) exhibit different rates of downregulation.

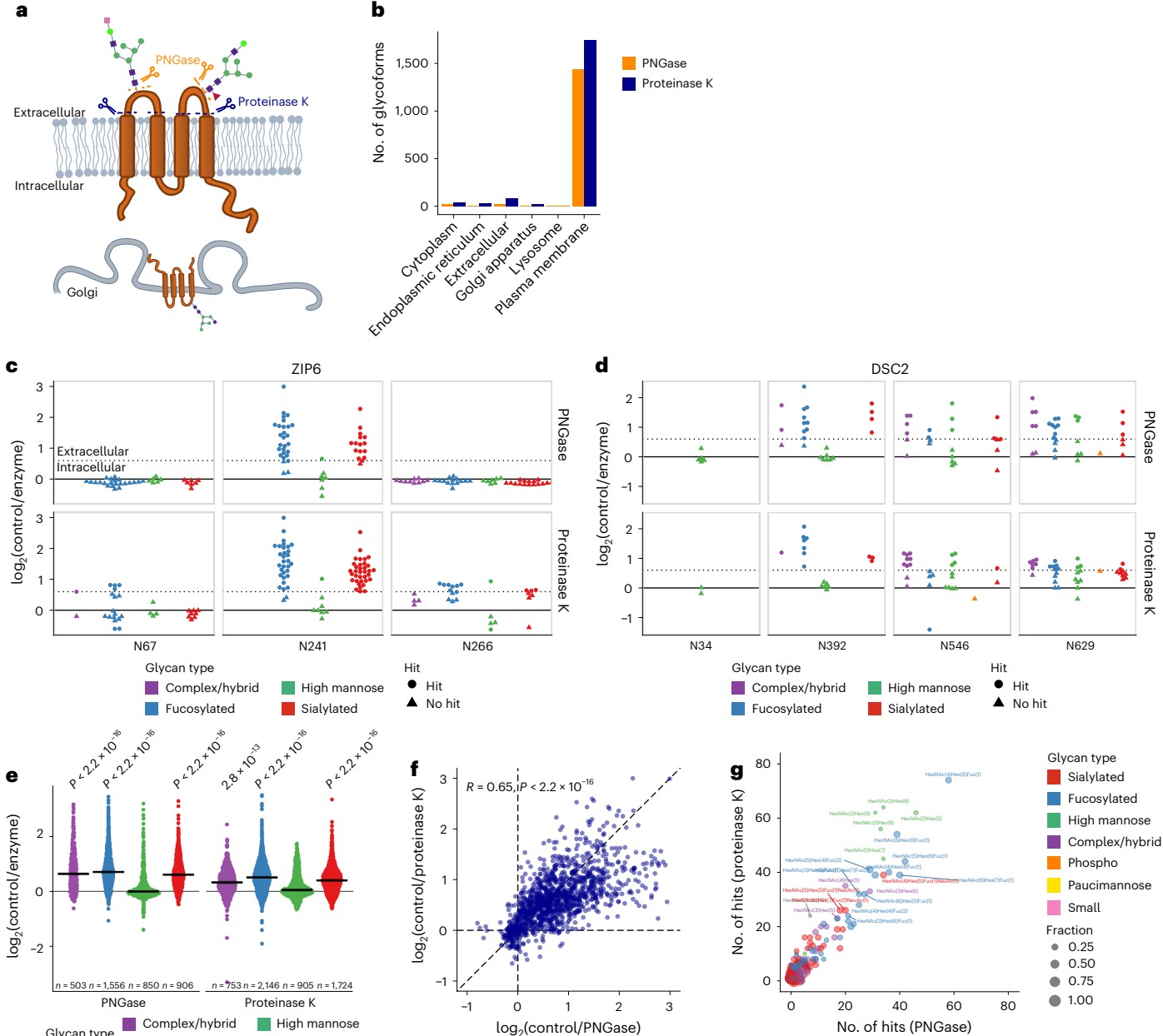

**Fig. 5 | Systematic characterization of surface-exposed glycoforms. a,** Intact living HEK293 cells are treated with either PNGase or proteinase K. Surface-exposed glycoforms are affected while intracellular glycoforms remain protected. **b,** Number of glycoforms on affected sites (at least two glycoforms changing) significantly changing in abundance upon enzymatic treatments, per subcellular compartment annotation of proteins. **c,d,** Characterization of surface-exposed glycoforms belonging to the zinc ion channel ZIP6 and to the cell adhesion protein DSC2. Each data point represents a unique glycoform, with the color representing the glycan class. The shape of the data point reflects significance, while the dotted line represents the effect size cutoff ($\log_2 0.6 = 1.5$ fold change). **e,** Fold change of glycoforms, separated per glycan class, on sites for which at least two glycoforms were affected by the enzymatic treatments. High-mannose glycans were significantly less exposed than the other glycan classes (two-sided $t$-test). The horizontal lines represent the median. **f,** Comparison of fold changes of glycopeptide abundance upon proteinase K ($y$ axis) and PNGase treatments ($x$ axis), for sites affected by both enzymes (sites with at least two glycopeptides significantly changing in abundance upon treatment) and glycopeptides identified in both experiments. **g,** Number of times a given glycan composition was identified on a glycopeptide significantly changing in abundance upon proteinase K ($y$ axis) and PNGase ($x$ axis) treatments. The size of the dot represents the frequency with which this composition was identified as surface exposed.

clusters (Fig. 4f) or the fact that, in general, fucosylated glycoforms at the same site are not coregulated (Supplementary Fig. 6d). This means that such differences in the kinetics of fucosylation downregulation cannot be solely attributed to differences in protein turnover rates. A clear example of this observation is the modulation of glycoforms on the proteins encoded by MRC2 and NRCAM, where fucosylated glycoforms on different sites follow distinct kinetics (Fig. 4g,h).

It is generally accepted that changes in protein glycosylation occur mainly through the modulation of glycoenzyme abundance, localization or activity. In such cases, similar trends would be expected at the glycoproteome level, especially for glycosites within the same protein. This site-specific and glycoform-specific modulation could be explained by glycoform-specific turnover rate or by more complex modulation at the fucosyltransferase level.

## Determination of cell-surface-exposed glycoforms

*N*-glycans are added to asparagines as high mannose in the ER before being further processed in the Golgi. Once the cells are lysed, it becomes impossible to differentiate mature, functional glycoforms from intermediate glycoforms going through the secretory pathway. Because of the complexity of the glycosylation machinery, it is not possible to predict the composition of mature glycans and it is postulated that glycoforms of any composition or structure could potentially be surface exposed[2]. Differences have been observed between cell-surface and intracellular glycans[33] but obtaining a comprehensive site-specific overview of mature glycoforms has been challenging so far and the compositions or the proportion of mature glycoforms in a given glycoproteomics experiment remains an open question. We applied our quantitative workflow to intact living human HEK293 cells subjected to either peptide:*N*-glycosidase F (PNGase F) treatment for 30 min (glycosidase targeting *N*-glycans) or proteinase K for 1 min (broad-specificity enzyme used in limited proteolysis experiments[34]). Our rationale was that only cell-surface-exposed glycoforms would be affected while glycoforms present in intracellular organelles would be protected, resulting in unchanged abundance (Fig. 5a). As a quality control, we demonstrated that the vast majority of proteins affected by the enzymes were annotated as belonging to the plasma membrane (Fig. 5b).

While some sites remained inaccessible and were not affected by the enzymatic treatments, we identified 3,990 glycopeptides on 826 sites and 513 proteins showing significant changes in abundance upon treatment (Supplementary Data 4 and Supplementary Fig. 7a,b). To enhance confidence, we decided to consider for subsequent analysis only sites for which at least two glycoforms were affected (3,113 glycopeptides on 299 sites and 168 proteins; Supplementary Fig. 6b). For example, on the zinc transporter ZIP6, the N241 site was consistently affected by both enzymatic treatments while the N67 and N266 sites remained unaltered (Fig. 5c). Regarding site N67, this can be explained by the fact that this site is located on a propeptide, the N-terminal cleavage triggering the location of ZIP6 to the plasma membrane[35]. Similarly, site N34 of desmocollin 2 (DSC2) remained unaffected because it is localized on the propeptide region, whereas glycans on the N392, N546 and N629 sites were revealed to be surface exposed (Fig. 5d).

Analyzing these accessible sites, we could demonstrate that high-mannose glycoforms were, in general, less surface exposed, although this tendency seemed to be site dependent. Conversely, most of the fucosylated or sialylated glycoforms were, at least partially, surface exposed, aligning with recent literature findings[33] (Fig. 5e). Sites exhibiting at least two high-mannose surface-exposed glycoforms were in general less accessible, supporting the notion that site accessibility may influence glycan maturation (Supplementary Fig. 7c). We could, in addition, identify glycan compositions predominantly characterized as surface exposed in our dataset (Supplementary Fig. 7d). For both enzymes, no discernible bias toward preferential cleavage of given glycan compositions or molecular weight was observed (Supplementary Fig. 7e,f). This suggests that the measured fold change upon treatment likely reflects the degree of surface exposition of a given glycoform.

We observed good correlation between the two experiments for sites impacted by both treatments, at both the glycopeptide level (fold change of a given glycopeptide between treatment and control conditions) and the glycan composition level (number of times a given glycan composition was observed on a glycopeptide significantly changing upon treatment) (Fig. 5f,g). PNGase and proteinase K were also found to be complementary (Supplementary Fig. 7g,h), with proteinase K effectively targeting less accessible glycosites (Supplementary Fig. 7i), likely because of the fact that cleavage at any amino acids in the vicinity of the glycosite would result in a change in glycopeptide abundance, whereas PNGase needs access to the glycosylated asparagine. In summary, we introduce a method for determining surface-exposed glycans and provide a system-wide, site-specific view of mature and immature glycans in a human cell line.

## Characterization of glycosylation tissue specificity

In 1987, the Thy1 membrane glycoprotein was identified as the first protein exhibiting tissue-specific glycosylation (in brain and liver), with one main site driving specificity and two passenger sites with a largely similar glycosylation profile across tissues[36]. Here, we conducted a proteome-wide quantitative profiling of *N*-glycosylation across three tissues (brain, liver and kidney) and two germ-free mice (C57BL/6) colonized with a human stool sample for 14 days (Fig. 6a and Supplementary Data 5). We demonstrated that the overall glycoproteomes exhibited tissue specificity after correction for differences in protein abundance (Fig. 6b,c), with high conservation of the glycoproteome within tissue across individuals. Notable differences in the liver glycoproteome were identified across individuals, whereas the kidney and brain glycoproteomes showed higher conservation (Supplementary Fig. 8a).

To assess tissue specificity at the glycosite level, we calculated the pairwise Pearson correlation of glycosylation patterns (correlating the relative intensities of all glycopeptides on a given glycosite between tissues) for sites having more than five quantified glycopeptides (Fig. 6d). This revealed that the tissue specificity arises from a subset of poorly correlated glycosites, while many glycosites exhibit highly correlated glycosylation patterns across tissues, extending Parekh et al.'s single-protein observation to a proteome-wide scale (Fig. 6e,f). Importantly, no link between the level of site microheterogeneity and tissue specificity was observed (Supplementary Fig. 8b). For further analysis, we categorized sites with high and low tissue specificity on the basis of the correlation below and above the mean correlation value (0.654), respectively. We also demonstrated that correlation values for glycosites within a given glycoprotein are site-specific, meaning that a low (or high) correlation on one site does not imply low (or high) correlations on other sites of the same protein (Fig. 6g). This is exemplified by the three glycosites profiled on the insulin receptor, with the glycosylation patterns of N138 and N910 being relatively similar between tissues, whereas N445 exhibits high tissue specificity (Fig. 6h). Interestingly, site N445 is located on a ligand-binding domain, suggesting that differences in glycosylation patterns may regulate protein activity in different tissues.

Although no structural differences were noted between these two groups, sites with high tissue specificity (low correlation across tissues) were significantly enriched in domains crucial for cell adhesion (cadherin and fibronectin domains), cell signaling (plexin and immunoglobulin-like fold domains) and immunity (immunoglobulin-like fold domain) (Fig. 6i). When considering only one type of glycan for correlation calculation, we observed that high-mannose glycoforms exhibit higher correlation than more processed glycoforms (Fig. 6j). This can be rationalized by the initial addition of all *N*-glycans as high mannose, with subsequent glycan processing being more tissue specific. Similarly, sites on proteins belonging to the ER tended to have a higher correlation than sites on proteins annotated as extracellular or belonging to the plasma membrane (Fig. 6k). We used summed TMT intensities of all quantified glycopeptides as a proxy for the absolute abundance of different glycan classes in the different tissues. While overall abundance is relatively similar across tissues, differences exist, such as the lower prevalence of fucosylation in the liver, consistent with a previous mouse *N*-glycomics atlas[37] (Supplementary Fig. 8c). Additionally, we identified tissue-specific glycan compositions with the highest fractional intensities in a given tissue (Supplementary Fig. 8d). Analyzing the top 1% of glycopeptides with the highest fractional abundance in a given tissue, we found that the most abundant glycan compositions in the kidney were of higher molecular weight (Supplementary Fig. 8e) and more fucosylated (Supplementary Fig. 8f). Furthermore, the most abundant sialylated glycan compositions contained more *N*-acetylneuraminic acid (NeuAc) in the brain, whereas, in the kidney and liver tissues, sialylated glycans containing *N*-glycolylneuraminic acid (NeuGc) were

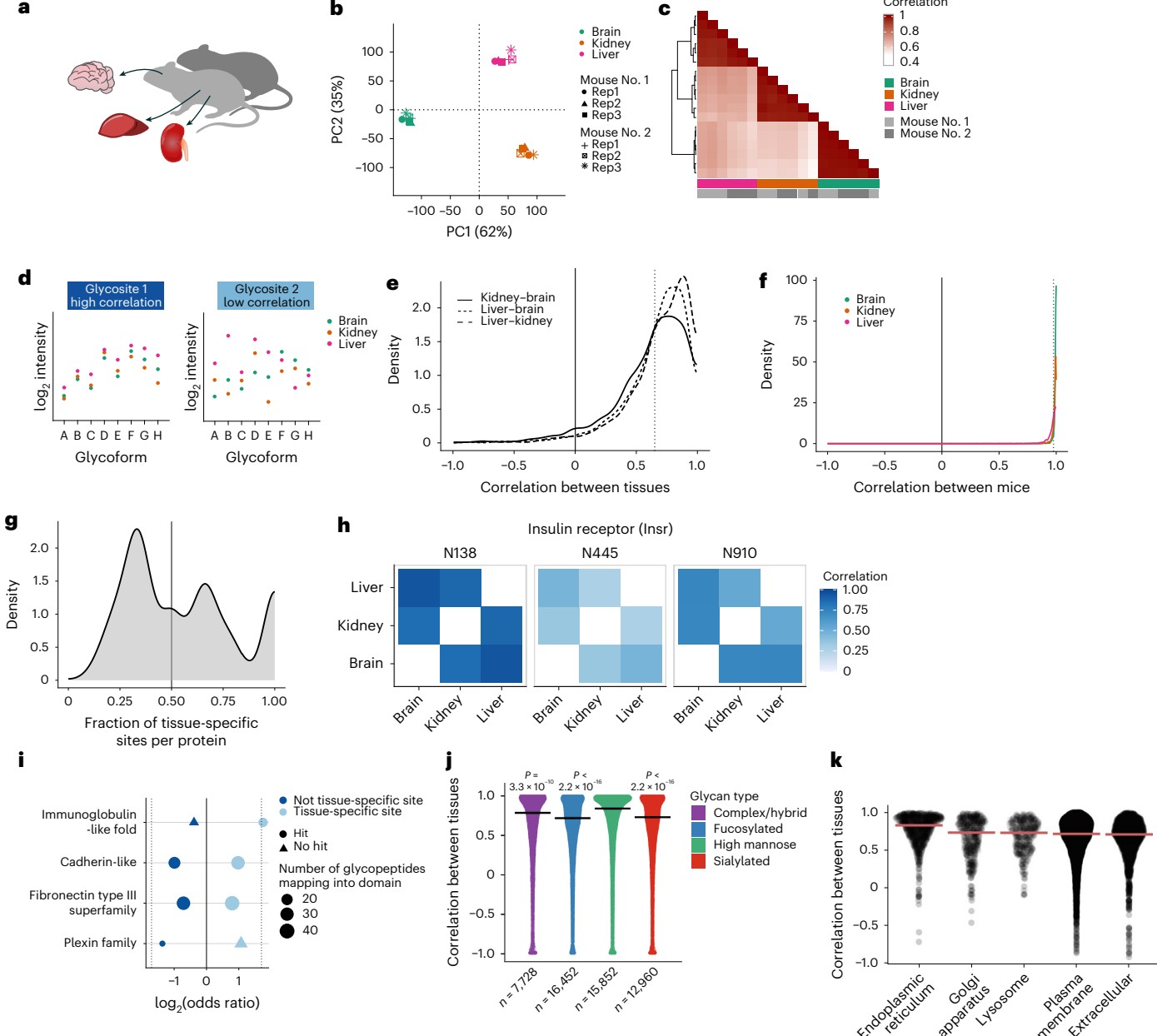

**Fig. 6 | Quantitative profiling of the glycoproteome across tissues.**
**a**, Experimental design. Glycoproteomes of the brain, liver and kidney tissues belonging to two male mice colonized (each in technical triplicates) with a human gut microbiome were profiled. **b**, PCA of the glycopeptide intensities after protein abundance normalization. **c**, Spearman correlation of glycopeptide intensities among different samples after protein abundance normalization. **d**, Pairwise Pearson correlation of the relative intensities of all glycopeptides on a given glycosite between tissues (for glycosites with at least five glycopeptides quantified) identifying glycosites with low tissue specificity (high correlation) and sites with high tissue specificity (low correlation). **e**, Density plot of site correlation values across tissues. The mean (0.669) is represented as a dotted line and chosen as a cutoff value for glycosite tissue specificity. **f**, Density plot of

site correlation values for the same tissue across two mice. **g**, Fraction of tissue-specific sites (correlation < 0.669) within a given glycoprotein (for proteins with at least three glycosites with a correlation value). **h**, Correlation values across tissues for the three sites quantified on the insulin receptor protein. **i**, Functional domain enrichment depending on glycosite tissue specificity (InterPro database, $\log_2$ odds ratio > 0 or $\log_2$ odds ratio < 0, two-sided Fisher's exact test, adjusted *P* value < 0.001; examples shown). **j**, Glycosite correlation values across tissues when only one type of glycan class is considered (two-sided, Wilcoxon rank-sum test). The horizontal lines represent the median. **k**, Glycosite correlation values across tissues per subcellular compartments. The horizontal lines represent the median.

more prevalent (Supplementary Fig. 8g), once again aligning with a previous glycomics study[37].

**Biophysical glycoproteomics**
Modulation of site microheterogeneity has been shown to regulate many cellular processes but determining its functional impact remains challenging. This is mainly because of the fact that only few

glycosylation events have a functional annotation, whereby it is currently impossible to selectively alter a given glycoform on a given glycosite both in vitro and in vivo and one cannot perform mutagenesis to assess the functionality of glycosite microheterogeneity. The quantitative glycoproteomics workflow developed in this study enables us to systematically determine biophysical properties of glycoforms in vivo by assessing the solubility of more than 30,000 glycoforms in

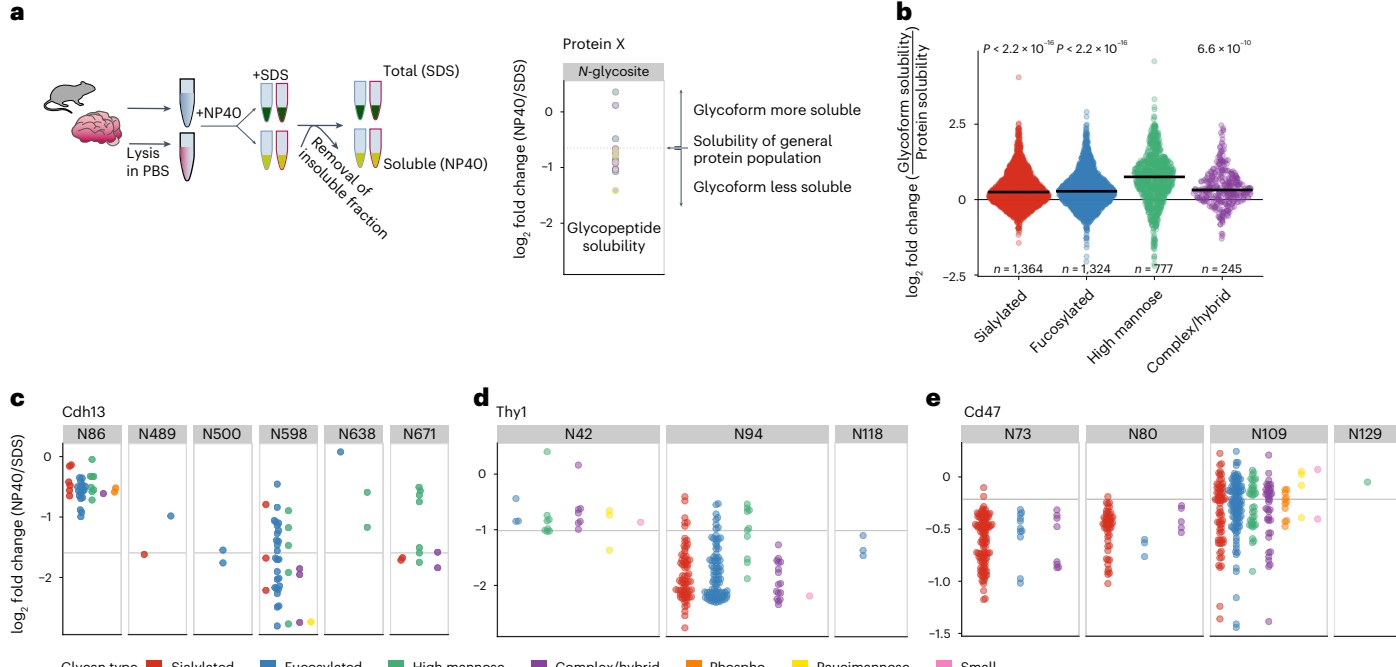

**Fig. 7 | Systematic characterization of glycoform biophysical properties.**
**a**, Solubility proteome profiling experimental workflow, enabling the proteome-wide characterization of the in vivo biophysical properties of glycoforms by assessing their solubility in a nondenaturing detergent (NP40) and in a SDS. **b**, Glycoform solubility compared to the solubility of the general protein population per glycan class (two-sided *t*-test). The horizontal lines represent the median. **c**–**e**, Solubility profiling of glycoforms of proteins encoded by Cdh13 (**c**), Thy1 (**d**) and Cd47 (**e**). Each dot represents a glycoform while the solid lines represent the solubility of the general protein population, based on nonmodified peptides.

the mouse brain using solubility proteome profiling[38] (Supplementary Data 6). Here, the differential solubility of a given proteoform in a mild detergent (NP40) and a SDS was determined (Fig. 7a), an approach previously used to systematically study the functional relevance of protein phosphorylation[39]. Although this approach does not directly reveal the impact of a change in glycosylation on specific protein function or activity, variations in protein solubility because of a specific glycosylation likely suggest differences in protein state, such as localization or interaction.

Application of glyco solubility proteome profiling showed that high-mannose glycoforms tend to be more soluble than more processed glycans (Fig. 7b), likely because of the fact that mature glycoforms are mostly less soluble than intracellular glycoforms, as a result of interactions with the extracellular matrix. This is particularly well illustrated in the case of the cadherin-13 protein (Cdh13), involved in cell adhesion; glycoforms on site N86, belonging to the propeptide, were more soluble than glycoforms on other sites belonging to the mature proteoforms (Fig. 7c). Overall, different glycoforms at the same site displayed varying solubility, indicating that glycosylation microheterogeneity has a widespread impact on protein states in vivo. Notably, we observed several instances of site-specific impact of glycosylation on solubility, wherein most glycosylation events on a given site tended to influence protein solubility in the same direction, as illustrated for site N94 of the Thy1 membrane glycoprotein or sites N73 and N80 of the leukocyte surface antigen Cd47 (Fig. 7d,e).

**Gut microbiome modulates the mouse brain glycoproteome**
The alteration in gut microbiome composition has been associated with behavioral changes and demonstrated to impact brain development and function through nervous or chemical signaling along the gut–brain axis[40]. However, underlying molecular mechanisms remain poorly characterized. Given the known importance of glycosylation for neuronal functions, we decided to assess the impact of different gut microbiome compositions on the brain proteome and

glycoproteome. Three groups of six adult germ-free C57BL/6 mice (three males and three females) were monocolonized for 2 weeks with *Bacteroides uniformis*, one of the most prevalent human gut microbes, colonized by a defined eight-member community composed of human commensal gut bacteria or kept germ-free (Fig. 8a). Changes in both protein abundance and glycoform levels were quantified (Fig. 8b,c and Supplementary Fig. 9a). In total, 303 of 10,865 quantified proteins in the unenriched sample and 2,575 of 45,724 quantified glycopeptides on 482 glycoproteins in the enriched sample exhibited significant changes at the proteome and glycoproteome levels, respectively (Fig. 8d and Supplementary Fig. 9b), with changes at the glycoform level predominantly occurring on sites exhibiting high microheterogeneity (Supplementary Fig. 9c). Only two proteins (encoded by Cntnap5A and Erbb3) exhibited significant changes at both the abundance and the glycosylation levels (Fig. 8e and Supplementary Data 7). A clear separation was observed among the three mice groups after principal component analysis (PCA), indicating gut microbiome-specific remodeling of the brain proteome and glycoproteome at the adult stage (Fig. 8c).

Proteins significantly changing in abundance were enriched in mitochondrial proteins (Fig. 8f), aligning with previous findings on gut microbiome–mitochondria crosstalk[41]. This interaction, likely through the secretion of small molecules, could constitute an interesting avenue to investigate in the context of the gut microbiome–brain axis, considering that mitochondrial function deregulation has been associated with neurological pathologies[42]. In addition, alterations in abundance were observed for nine of the 12 G protein γ-subunits, with the eight moderate-translocating and slow-translocating G protein γ-subunits identified being upregulated and the fast-translocating γ-subunit 11 being downregulated (Fig. 8f). G proteins associate with transmembrane receptors to modulate and relay signals to the inside of the cell and G-protein γ-subunit translocation kinetics has been associated with cell sensitivity and adaptation to extracellular signals[43], hinting at a change in receptor activity.

To further our understanding of the impact of gut colonization on the brain proteome of adult mice, we used thermal proteome profiling[44] (Supplementary Fig. 9d). This approach measures the melting behavior of proteins in vivo on a proteome-wide scale, linking protein thermal stability change with the alteration of protein state, such as changes in protein interaction or localization[45]. Many proteins involved in axon guidance and neuronal migration (Supplementary Fig. 9e and Supplementary Data 8) exhibited alterations in thermal stability when comparing the brains of mice colonized with the eight-member community to germ-free mice. In particular, components of the cytoskeleton, essential for supporting axonal growth[46], displayed changes in thermal stability, potentially stemming from a change in protein polymerization (Fig. 8g). Most of these proteins, part of the GO term 'structural molecule activity' showed a decrease in protein thermal stability upon colonization (Fig. 8h).

Proteins involved in axon guidance also showed changes in glycoform levels (Fig. 8i). The absence of a maternal microbiome has previously been linked to defects in fetal neurodevelopment, with reduced axonogenesis and downregulation of genes involved in axon guidance[47]. Given the crucial role of glycosylation in axon guidance[48] because of its importance in the regulation of cell adhesion and migration processes, our findings provide another layer of evidence to illustrate how the gut microbiome could impact axonogenesis. In addition, N-glycosylation has been linked with the modulation of neurotransmission by regulating the affinity binding, oligomerization states, trafficking or cell-surface retention of neurotransmitter receptor ligands[49]. Amongst proteins exhibiting changes at the glycosylation levels, 13 glutamate and six GABA G-protein-coupled receptor proteins were identified (Fig. 8i). Glutamate and GABA are the main excitatory and inhibitory neurotransmitters, respectively. We additionally identified changes in glycoform abundance on several ion channel proteins (Fig. 8i). N-glycosylation has been demonstrated to regulate neurotransmission by modulating membrane excitability through the regulation of ion channel proteins[49]. Overall, the group of proteins showing changes in glycosylation upon microbiome colonization exhibited alterations in thermal stability when compared to all glycoproteins identified (Fig. 8h). For example, the protein Contactin-associated protein 1 (Cntnap1), which enables the regeneration of the electrical signal along the axon[50] and for which 22 glycopeptides were identified as being regulated, displayed a clear decrease in thermal stability (Fig. 8j and Supplementary Fig. 9f). While this does not provide direct proof of the functionality of the observed glycosylation changes, this suggests that these alterations could be linked to biological changes.

No trend in specific regulation of one class of glycoform was observed (that is, all glycan classes followed the same trend; Supplementary Fig. 9g) and glycopeptides belonging to the same glycan class localized on the same site were not coregulated (Supplementary Fig. 9h,i). Regulated glycoforms belonging to the same site, regardless of glycan composition, tended to change in abundance in the same direction (Supplementary Fig. 9j,k), suggesting a regulation of the overall level of glycosylation of specific glycosites. This is well illustrated by the regulation of glycoforms on the prolow-density lipoprotein receptor-related protein 1 (Lrp1; Supplementary Fig. 9l), where some sites exhibited general upregulation or downregulation of glycoforms upon gut microbiome colonization. Furthermore, our study revealed extensive site-specific regulation of glycoforms wherein only specific glycoforms on specific glycosites within a protein were regulated while others remained unchanged, as shown for the glutamate receptor ionotropic NMDA 2A (Grin2a; Fig. 8k). Notably, only two glycosyltransferase enzymes changed significantly in abundance in this study (Alg12 and Mgat4e, unlikely to be solely responsible for the vast changes in microheterogeneity observed) and no glycoenzymes showed a significant alteration in thermal stability. Taken together, these observations suggest the existence of intricate posttranslational mechanisms of regulation governing site-specific glycosylation dynamics.

A compelling example involves the two glycosites identified on Eaat2 (Fig. 8l), on which a similar number of glycoforms were quantified (345 on N205 and 349 on N215). While only ten amino acids apart, the N205 site exhibits extensive upregulation of glycoforms in the brain of colonized mice, while only four glycoforms were found to significantly change on the N215 site (Fig. 8m). This family of transporters has an essential role in the brain by maintaining low levels of extracellular excitatory neurotransmitter glutamate in the synaptic cleft to avoid excitotoxicity and their deregulation has been associated with various neurological diseases[51]. A general decrease in glycosylation level on Eaat2 has been observed in brain samples of persons with schizophrenia[52], illustrating how the gut microbiome could impact brain physiology through the modulation of protein glycosylation. In addition, substitution of the N205 site to a serine was shown to decrease the uptake activity of the Eaat2 transporter[53]. Mapping the modulated glycoforms to the solubility data revealed that upregulated glycoforms upon gut colonization were more soluble than the average glycoform (Fig. 8n), hinting that the remodeling of Eaat2 glycosylation may be linked to changes in protein state. Interestingly, Lrp1, which shows widespread glycosylation alterations upon gut colonization (Supplementary Fig. 9l), has been shown to have key roles in tau uptake[54] and to exhibit abnormal glycosylation in the brain of persons suffering from Alzheimer disease[55]. The observed shift in thermal stability of Lrp1 upon colonization (Supplementary Fig. 9m), along with the altered solubilities of its regulated glycoforms (Supplementary Fig. 9n), suggests that gut microbiome colonization induces a change in protein state of Lrp1.

**Fig. 8 | Impact of the gut microbiome on the mouse brain glycoproteome.**
**a**, Three groups of six adult germ-free mice (three males and three females) were colonized by *B. uniformis*, by a defined eight-member community of human gut commensal or remained germ-free for 2 weeks. **b**, Spearman correlation of glycopeptide intensities between biological replicates after protein abundance normalization. **c**, PCA of normalized reporter intensities of the 18 brain samples. **d**, N-glycopeptide regulation in the mouse brain depending on gut microbiome composition. **e**, Overlap of significantly regulated proteins at the glycoproteome and proteome level. **f**, Mitochondrial proteins involved in cellular respiration and G protein γ-subunits regulated in protein abundance. **g**, Cytoskeleton proteins change in thermal stability upon gut microbiome colonization (*n*, number of proteins in each family; actins: Actb, Acta2 and Actg1; dyneins: Dnm1, Dnm2 and Dnm3; myosins: Myh10, Myh11, Myh9 and Myl12b; α-tubulins: Tuba1b, Tuba4a, Tuba8 and Tubal3; β-tubulins: Tubb1, Tubb2, Tubb2b, Tubb3, Tubb4a, Tubb4b, Tubb5 and Tubb6; γ-tubulins: Tubg1 and Tubg2). **h**, ΔAUC of thermal stability of proteins in brains of germ-free and eight-member colonized mice. Glycoproteins with at least one glycopeptide changing significantly in abundance upon colonization and proteins with GO term 'structural molecule activity' had higher thermal stability compared to other glycoproteins (two-sided *t*-test) The horizontal lines represent the median. **i**, N-glycosylation modulation on proteins involved in neurotransmission. **j**, Cntnap1 changes in thermal stability upon gut microbiome colonization. **k**, Site-specific regulation of N-glycosylation on the Grin2a glutamate receptor. Each vertical line represents one glycopeptide and each dot represents one replicate (3 conditions × 6 biological replicates). **l**, Eaat2 encodes a membrane-bound transporter, glycosylated at the N205 and N215 sites on an extracellular loop (blue). Human structure combined with AlphaFold predictions for the extracellular region is shown. The light-gray structure represents the homotrimer (Protein Data Bank 7VR8). Yellow planes depict the membrane orientation; dark-blue and dark-gray regions depict the predicted structures aligned to the template structure. Spheres indicate the N205 and N215 glycosites. **m**, The glycoforms on the two glycosylation sites of the Eaat2 protein are differentially modulated upon gut microbiota colonization. **n**, Solubility of the Eaat2 glycoforms from the solubility experiment. Gut-microbiome-modulated glycoforms are highlighted. The horizontal lines represent the median.

## Discussion

Deep profiling of protein glycosylation in the mouse tissues and human cell lines characterized site microheterogeneity to an unprecedented extent. Overall, regarding the interplay among the different optimized parameters, we believe that the improvement of MS acquisition methods and PGC fractionation would enhance all glycoproteomics enrichment platforms. However, nucleic acid precipitation should mostly be beneficial for sample preparation before PBA enrichment. Indeed, PBA will form a covalent bond with diols present in nucleic acids but lectin and, to a lesser extent, enrichment based on hydrophilic interaction chromatography should remain largely unaffected by the presence of nucleic acids in the sample.

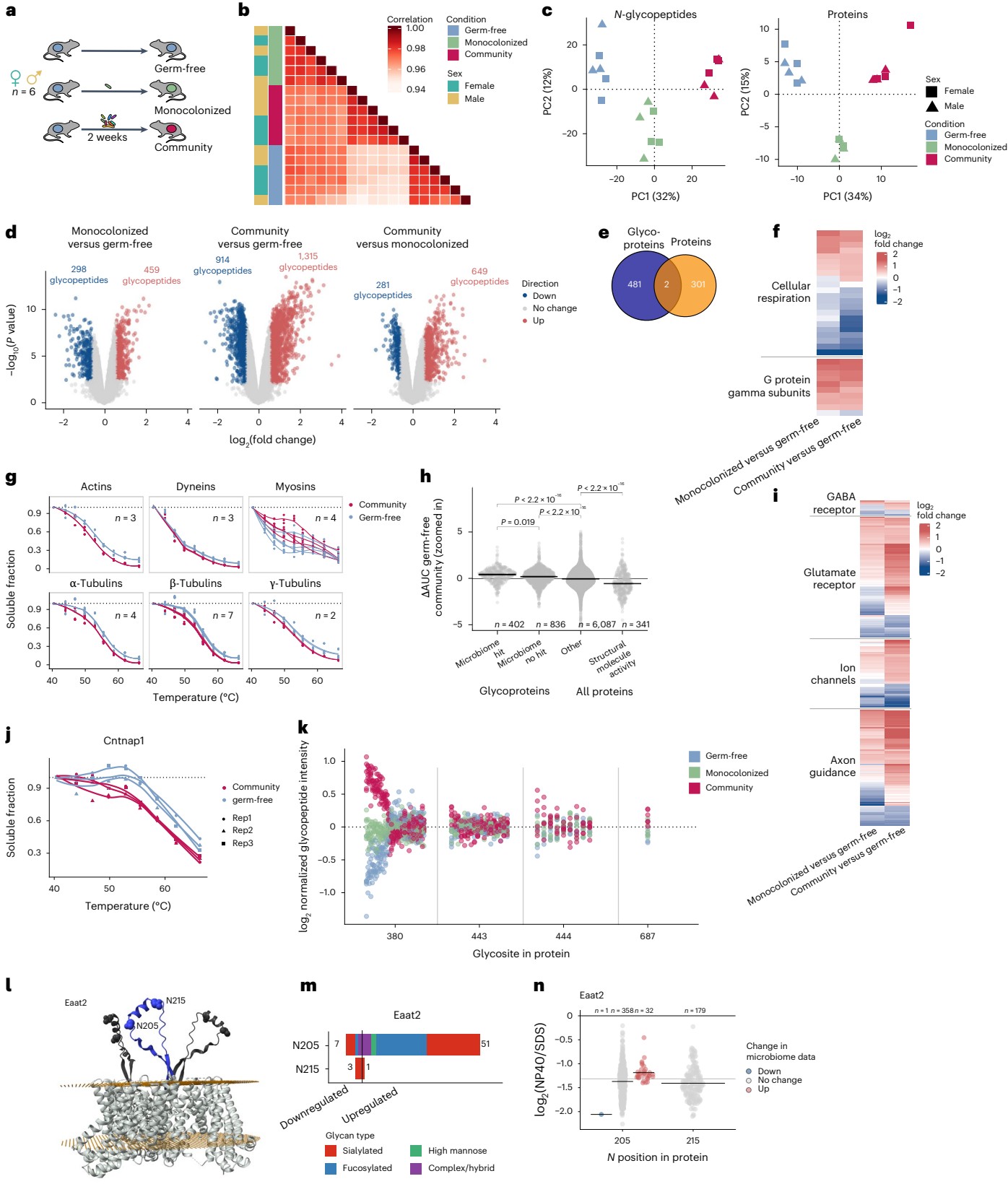

AlphaFold predictions[25] enabled us to define site-specific structural features such as the fact that sites with a higher degree of microheterogeneity are more likely to be located on highly ordered structures and to have high solvent accessibility. The latter aligns with previous work showing that solvent accessibility is an important parameter for further processing of high-mannose glycans to create mature glycan forms[56,57]. We foresee that our extensive dataset will be a useful resource to further understand and predict the reciprocal impact between protein structures on glycosylation.

Our affordable and versatile workflow for multiplexed quantitative analysis of protein glycoforms makes it possible to systematically quantify protein glycosylation levels across a large number and range of samples. This approach holds substantial promise for identifying disease-specific glycosylation patterns. Likewise, the profiling of glycosylation across tissues is relevant for characterizing abnormal, disease-driven glycosylation patterns. This could, in the future, support the identification of biomarkers[58] and, more generally, improve our understanding of the association between deregulation of glycosylation and disease states. The discovery of tissue-specific glycosites also opens avenues for exploring the roles of glycosylation in modulating protein activity across diverse tissues.

Previous studies quantified glycosylation changes in cell lines after perturbations directly targeting glycosyltransferases (for example, inhibition[29] or gene knockout[28]). Here, we quantified the dynamics of the glycoproteome in vivo in response to controlled perturbation. The remodeling of the brain glycoproteome upon changes of the gut microbiome composition suggests high plasticity of glycosylation in response to a broad range of stimuli. While the exact impact of the observed glycosylation changes on protein functions remains to be elucidated, the fact that many of these changes occur on neurotransmitter receptors or proteins involved in axon guidance, for which glycosylation has been previously shown to be essential for protein activity[48,49], reinforces the link between the gut microbiome and brain physiology.

Previous work coined the term metaheterogeneity to reflect the diversity of glycosylation among different sites within the same protein[59]. In our data, we could observe that the differences between glycoproteomes across different tissues are driven by a subset of sites with high tissue specificity in terms of their glycosylation pattern. Similarly, perturbations can give rise to quantitative changes in site-specific glycoform compositions, illustrating how metaheterogeneity can be modulated in response to perturbations. Investigating the mechanisms behind these site-specific modulations opens exciting avenues for future research.

In summary, the presented workflow constitutes a framework for addressing many unresolved questions, such as the kinetics of glycosylation changes in response to various perturbations or the impact of glycosylation microheterogeneity modulation on protein function. The latter point represents a particularly difficult challenge as site mutagenesis, commonly used to assess the functionality of other modifications (for example, phosphorylation[60]), cannot address this issue and no alternative strategy to alter glycosylation on a specific site has so far been described. Here, we conducted a proteome-wide characterization of glycoform solubility, revealing a widespread link between glycosylation microheterogeneity and protein state in vivo. We believe that this method lays the groundwork for the field of functional glycoproteomics, providing a powerful tool for generating hypotheses about glycoform functionality. Future efforts can also focus on measuring the effect of glycosylation changes on other biophysical properties of proteins, such as thermal stability[44,61], which can inform on changes in ligand-binding properties[44] and protein interactions[62].

## Online content

Any methods, additional references, Nature Portfolio reporting summaries, source data, extended data, supplementary information,

acknowledgements, peer review information; details of author contributions and competing interests; and statements of data and code availability are available at https://doi.org/10.1038/s41594-025-01485-w.

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

## Methods

### Cell culture

All cell lines used in this study were verified to be negative for *Mycoplasma* contamination. HeLa Kyoto cells (a gift from the Ellenberg group, European Molecular Biology Laboratory (EMBL), previously authenticated[63]) and HEK293T cells (obtained from the American Type Culture Collection, CRL-3216) were cultured in DMEM (Sigma-Aldrich, D5648) containing 4.5 mg ml$^{-1}$ glucose, 10% (vol/vol) FBS (Gibco, 10270) and 1 mM L-glutamine (Gibco, 25030081) at 37 °C with 5% CO$_2$. HeLaK and HEK293T cells (0.5 million) were seeded in 150-mm dishes and grown for 2 days. The cells were washed with ice-cold PBS (2.67 mM KCl, 1.5 mM KH$_2$PO$_4$, 137 mM NaCl and 8.1 mM NaH$_2$PO$_4$, pH 7.4) and collected by scraping. The cells were pelleted by centrifugation at 300$g$ for 3 min. The cell pellets were flash-frozen in liquid N$_2$ and stored at −80 °C.

### Mouse experiments

All mouse experiments were performed using approved protocols by the EMBL ethics committee (license 21-002_HD_MZ). Germ-free C57BL/6 mice were maintained and bred in gnotobiotic isolators (CbC) with a 12-h light–dark cycle. Germ-free status was monitored by PCR (using the 16S primer pair; Supplementary Data 7) and culture-based methods. Mice were provided with standard, autoclaved chow (1318P FORTI, Altromin) ad libitum. Animals of mixed gender at an age between 8 and 17 weeks were used for all experiments. Informed consent was obtained from the human donor and colonization with human stool samples was performed by two oral gavages in 48 h with 200 µl of homogenized and cryopreserved stool samples from a human donor according to the approved protocol (2015-009) by the EMBL Bioethics Internal Advisory Committee. Colonization with laboratory strains was performed by oral gavage with 200 µl of cryopreserved bacteria from overnight cultures in modified Gifu anaerobic medium (Nissui Pharmaceutical) of *B. uniformis* DSM6597 or with a mixed in vitro culture containing eight bacterial species mixed in equal ratios, based on optical density measurements at 600 nm. All inocula were made in advance and stored at −80 °C until use. Colonized animals were single-housed for 14 days before they were killed and brain, liver, kidney and cecal samples were collected and stored at −70 °C until further analysis.

### Determination of gut bacterial community composition by targeted microbiome profiling

The relative cecal abundance of each species was determined from purified DNA by qPCR (Supplementary Data 7). DNA was extracted using a commercially available genomic DNA (gDNA) extraction kit (96 DNA kit, ZymoBiomics) following the manufacturer's instructions. Species-specific primers were designed against intergenic regions of the published genome of each species (Supplementary Data 7). All primers were tested in silico and by qPCR for their reactivity to nonspecific sites within the genomes of other strains in the community. For each qPCR assay, a standard curve was calculated from gDNA isolated from each species. To this aim, gDNA was diluted to 0.5 ng µl$^{-1}$ (as quantified by the Quant-iT double-stranded DNA BR assay kit and a Qubit fluorometer (Invitrogen). Ten-point standard curves were prepared from twofold serial dilutions of this 0.5 ng µl$^{-1}$ starting point. qPCR reactions were run in 20-µl volumes, containing 10 µl of SYBR green PCR master mix (Thermo Fisher Scientific), 6.8 µl of nuclease-free water, 1.2 µl of primers at 333 nM final concentration each and 2.0 µl of template DNA. Amplification was performed using the StepOnePlus real-time PCR system (Applied Biosystems) machine running StepOne software (version 2.3) according to the manufacturer's recommended protocol: 95 °C for 10 min, followed by 40 cycles of 15 s at 95 °C and 1 min at 60 °C. Melting curve analysis followed after amplification to ensure a single product in each reaction. The qPCR estimate of the amount of DNA from each strain was estimated by normalization of the $C_t$ values by the estimated molecular weight of each strain's genome, thereby converting the $C_t$ values to genome equivalents.

Relative abundance was calculated as the proportion of a single-strain genome equivalent to the total genome equivalents in the sample.

### Sample preparation

Samples (cell lines or tissues) were lysed in a buffer composed of 4 M guanidinium isothiocyanate, 50 mM HEPES, 10 mM TCEP, 1% *N*-lauroylsarcosine, 5% isoamyl alcohol and 40% acetonitrile adjusted to pH 8.5 with 10 M sodium hydroxide. The volume of lysis buffer used corresponded to approximately five times the sample volume. Human cell pellets were pipetted up and down while mouse tissues were homogenized using a bead beater with a mix of 2.8 mm and 1.6 mm zirconium silicate beads for 30 s at 6 m s$^{-1}$ (Beadruptor Elite, Omni International). For the SDS lysis, samples were lysed in a buffer composed of 2% SDS, 50 mM HEPES and 10 mM TCEP. After 10 min of incubation at room temperature in a shaker at 1,000 rpm, samples were centrifuged at 16,100$g$ at room temperature for 10 min or filtered in multiscreenHTS-HV 0.45-µm 96-well filter plates with PVDF membranes (Merck Millipore) to remove cell debris and nucleic acid aggregates. Protein concentrations were determined using a tryptophan fluorescence assay[64] and the samples were diluted to a maximum concentration of 25 µg µl$^{-1}$, if needed. Volumes of 80 µl of lysate per well were transferred into a multiscreenHTS-HV 0.45-µm 96-well filter plate and 220 µl of ice-cold acetonitrile was added to induce protein precipitation. After 10 min, the plate was centrifuged to remove solution and protein precipitates were washed twice with 200 µl of 80% acetonitrile and two times with 200 µl of 70% ethanol (1,000$g$ for 2 min for each step). Then, the digestion buffer composed of 100 mM HEPES pH 8.5, 5 mM TCEP, 20 mM chloroacetamide and trypsin (TPCK-treated trypsin, Thermo Fisher Scientific) was added to the protein precipitates in the filter plate. The ratio of trypsin to protein was fixed to 1:25 w/w and the maximum final protein concentration was set to 10 µg µl$^{-1}$. Trypsin was reconstituted at a stock concentration of 10 µg µl$^{-1}$ and added to the ice-cold digestion buffer just before addition to the precipitates. Tryptic digestion was carried out overnight at room temperature under mild shaking (600 rpm). After digestion, samples were acidified with 1% TFA and desalted using Sep-Pak tC18 columns (Waters), eluted using 0.1% TFA in 40% acetonitrile and dried using a vacuum concentrator.

For glycopeptide enrichment, dried peptides were resuspended in the glycopeptide enrichment buffer composed of 50 mM carbonate buffer pH 10.5 in 50% acetonitrile. The optimal peptide concentration was determined to be 5 mg ml$^{-1}$. Silica beads functionalized with PBA (Bondesil-PBA 40 µm, Agilent) were used for glycopeptide enrichment at an optimized ratio of 1:2.5 (w/w) peptides to PBA beads. Before addition to the samples, beads were washed three times with the glycopeptide enrichment buffer. The sample–bead mixtures were incubated in Eppendorf tubes or 96-well plates at room temperature for 1 h under sufficient shaking to prevent bead sedimentation. After incubation, the sample–bead mixtures were transferred to a multiscreenHTS-HV 0.45-µm 96-well filter plate, nonbound peptides were filtered off and beads were washed seven times with 200 µl of glycopeptide enrichment buffer, followed by centrifugation, without the need for incubation, at a speed of 100$g$ for 1 min. Glycopeptides were eluted two times with 50 µl of 1% TFA in 50% acetonitrile for 15 min at room temperature under shaking at 800 rpm before being dried in the 96-well elution plate or glass inserts using a vacuum concentrator. For single-shot measurements of glycopeptides, we found that the number of identification reached a plateau after 1 mg of input. For the PGC-fractionated sample, we enriched 2 mg of peptide per TMT channel or 30 mg of peptides in the case of label-free PGC experiments.

For the TMT labeling, because of the presence of residual acidic salts, glycopeptides were resuspended for 5 min in 10 µl of 400 mM HEPES pH 10 before addition of the TMTpro reagents (Thermo Fisher Scientific). To quantify the full proteome, 15-µg aliquots were taken after sample desalting and before glycopeptide enrichment, dried and resuspended in 10 µl of 100 mM HEPES pH 8.5 before TMT labeling.

For all samples, 4 µl of TMTpro reagents at a concentration of 20 µg µl⁻¹ in acetonitrile were added to each well. The labeling reaction was carried out for 1 h at room temperature under mild shaking before being quenched by 5 µl of 5% hydroxylamine for 15 min. Samples belonging to the same TMT experiment were then pooled and dried using a vacuum concentrator. Before fractionation, the full proteome samples were desalted using Sep-Pak tC18 columns (Waters), while the labeled glycopeptides were dissolved into 100 µl of 10% TFA and desalted using C18 StageTips[65] made in-house and packed with 1 mg of C18 bulk material (ReproSil-Pur 120 C18-AQ 5 µm, Dr. Maisch) on top of the C18 resin disk (AttractSPE disks bio C18, Affinisep).

### 2FF time course

Expi293F (suspension cells derived from HEK293 cells, Thermo Fisher Scientific) were cultured in FreeStyle 293 expression medium (serum-free medium) until reaching a concentration of 0.5 million cells per ml. A total of 10 millions cells were harvested and divided into two groups to serve as the $t = 0$ time point. The remaining cells were split into four flasks of equal volume. In two flasks, 200 µM 2FF (Synchem) was added and the other two flasks received an equal volume of DMSO as a control. Then, 5 million cells of both treated and control cells were harvested at 8, 24, 48 and 72 h after treatment (resulting in biological duplicates for each time point). After harvesting, the cells were spun down at 300g for 5 min, washed with cold PBS and spun down again, followed by flash-freezing in liquid nitrogen before sample preparation including lysis, digestion, glycopeptide enrichment and TMT labeling.

### Determination of surface-exposed glycoforms

HEK293 cells were washed off from plates by gently pipetting PBS up and down, washed with PBS at room temperature and aliquoted in PCR tubes (5 million cells per tube in a volume of 200 µl of PBS). Both PNGase and proteinase K treatments were realized in quadruplicate, with the controls being incubated in the same conditions. For the PNGase experiment, 5 µl of rapid PNGase F (nonreducing format, New England Biolabs) was added to the living cells, followed by an incubation of 30 min at 40 °C, with the cells being resuspended by gently pipetting up and down every 5 min. After incubation, cells were centrifuged at 1,000g for 1 min, the supernatant was discarded and 100 µl of lysis buffer was added to each tube.

For the proteinase K experiment, 5 µg of proteinase K was added to the living cells, followed by an incubation of 1 min at room temperature. After incubation, AEBSF protease inhibitor (cOmplete mini EDTA-free, Roche) was added at the recommended concentration, the cells were centrifuged at 1,000g for 1 min at 4 °C and the supernatant was discarded. Cells were then lysed by the addition of 1% SDS in PBS supplemented with protease inhibitors and the lysate was boiled (tubes were immersed in boiling water) for 15 min to inactivate proteinase K. Proteins were precipitated by the addition of acetonitrile to a final concentration of 60%, pelleted and resuspended in 100 µl of lysis buffer, before processing as described above.

### Solubility proteome profiling

Mouse brain tissue was homogenized on ice in PBS supplemented with protease and phosphatase inhibitors (cOmplete mini EDTA-free and PhosSTOP, Roche) using a glass douncer. A volume corresponding to 4 mg of protein at a concentration of 50 µg µl⁻¹ was aliquoted into eight PCR tubes (four corresponding to the NP40-only samples and four corresponding to the SDS samples). NP40 was added to all tubes at a final concentration of 0.8%. For the SDS-treated aliquots, SDS was added to a final concentration of 1%, along with benzonase to reduce sample viscosity. All samples were centrifuged at 100,000g at 4 °C for 20 min. The supernatant was then transferred to fresh tubes, where the proteins were precipitated by the addition of acetonitrile at a final concentration of 60% for 20 min on ice. After precipitation, proteins were pelleted and resuspended in 100 µl of lysis buffer. The samples were then processed as described above.

### Thermal proteome profiling

Mouse brain tissues of germ-free mouse and mouse colonized by the eight-member bacterial community were homogenized on ice in PBS supplemented with protease and phosphatase inhibitors (cOmplete mini EDTA-free and PhosSTOP, Roche) using a glass douncer. The lysates were then aliquoted in PCR plates (20 µl per well) and heated to nine temperatures (40.4, 44.0, 46.9, 49.8, 52.9, 55.5, 58.6, 62.0 and 66.3 °C) for 3 min. Next, hydrophobic magnetic beads (Sera-Mag, Cytiva) were added and the samples were gently mixed by pipetting up and down, followed by an incubation of 10 min on ice. Then, 40 µl of lysis buffer (1.2% NP40, 2 mM MgCl₂, 1% benzonase and protease and phosphatase inhibitors in PBS) was added to each tube. After 20 min of incubation on ice, the plate was centrifuged at 3,000g for 10 min at 4 °C and the supernatant was transferred to a multiscreenHTS-HV 0.45-µm 96-well filter plate with PVDF membranes (Merck Millipore). After filtration, the samples were denatured by addition of the same volume of denaturing buffer (6 M Gnd-SCN, 100 mM HEPES and 10 mM TCEP) and the protein was precipitated by addition of acetonitrile to a final concentration of 70%. The samples were then washed and digested as described above. Regarding TMT labeling, samples corresponding to the germ-free mouse brain were labeled with channels 1–9 while samples corresponding to the community colonized mouse brain were labeled with channels 10–18 before pooling. The thermal proteome profiling experiments were performed in triplicate.

### PGC fractionation

Samples were reconstituted in 18 µl of buffer A (0.05% TFA in MS-grade water supplemented with 2% acetonitrile) and the injection volume was fixed to 16 µl. The glycopeptides were separated on a Hypercarb column (100 mm, 1.0-mm inner diameter, 3-µm particle size; Thermo Fisher Scientific) at a temperature of 50 °C and a flow rate of 75 µl min⁻¹ using an Ultimate 3000 liquid chromatography (LC) system (Thermo Fisher Scientific). The linear separation gradient started 1 min after injection and increased from 13% buffer B (0.05% TFA in acetonitrile) to 42% buffer B after 95 minutes, before increasing to 80% buffer B in 5 min. The column was washed with 80% buffer B for 5 min before being re-equilibrated for 5 min with 100% buffer A. Fractions were collected from 4.5 to 100.5 min with a 2-min collection period, resulting in 48 fractions that were subsequently pooled into 24 fractions, where each $n$ fraction was pooled with the $n + 24$ fraction. Samples were dried using a vacuum concentrator before LC–MS/MS analysis. For the tissue atlas, samples were separated into 96 fractions, collected every 2 min (linear gradient from 13% to 42% B) and pooled into 48 fractions.

### LC–MS/MS analysis

All samples were resuspended in a loading buffer containing 1% TFA, 50 mM citric acid and 2% acetonitrile in MS-grade water. Peptides were separated using an UltiMate 3000 RSLCnano system (Thermo Fisher Scientific) equipped with a trapping cartridge (Precolumn; C18 PepMap 100, 5 µm, 300-µm inner diameter × 5 mm, 100 Å) and an analytical column (Waters nanoEase HSS C18 T3, 75 µm × 25 cm, 1.8 µm, 100 Å). Solvent A was 0.1% formic acid supplemented with 3% DMSO in LC–MS-grade water and solvent B was 0.1% formic acid supplemented with 3% DMSO in LC–MS-grade acetonitrile. Peptides were loaded onto the trapping cartridge (30 µl min⁻¹ solvent A for 3 min) and eluted with a constant flow of 300 nl min⁻¹. Peptides were separated using a linear gradient of 8–28%, 5–25% and 6–26% B for 156 min for the full proteome, label-free glycopeptides and TMT-labeled glycopeptides, respectively, followed by an increase to 40% B within 4 min before washing at 85% B for 4 min and re-equilibration to initial conditions. The LC system was coupled to a Fusion Lumos Tribrid or Exploris 480 MS instrument (Thermo Fisher Scientific) operated in positive ion mode with a spray

voltage of 2.4 kV and a capillary temperature of 275 °C. Full-scan MS spectra were acquired in profile mode in the Orbitrap using a resolution of 120,000, with a mass range of 375–1,500 $m/z$, 700–2,000 $m/z$ and 800–2,000 $m/z$ in the case of the full proteome, label-free glycopeptides and TMT-labeled glycopeptides, respectively. The maximum injection time was 50 ms and automatic gain control (AGC) was set to $4 \times 10^5$ charges. The MS instrument was operated in data-dependent acquisition mode and precursors with charge states 2–7 and a minimum intensity of $2 \times 10^5$ were selected for subsequent HCD fragmentation with a maximum duty cycle time of 3 s. Peptides were isolated using the quadrupole with an isolation window of 0.7 $m/z$ and 1.4 $m/z$ in the case of TMT-labeled samples and label-free samples, respectively. Precursors were fragmented with a normalized collision energy of 34% for the TMT-labeled full proteome, a normalized collision energy of 38% or stepped collision energy of 22%, 30% and 38% for the label-free glycopeptides (Exploris 480) and a stepped collision energy of 25%, 36% and 45% (Fusion Lumos) or 23%, 34% and 42% (Exploris 480) for the TMT-labeled glycopeptides. A dynamic exclusion window of 45 s was used for the full proteome sample and 30 s for glycopeptides. MS/MS spectra were acquired in profile mode with a resolution of 30,000 (label-free) or 45,000 (TMT) in the Orbitrap. The maximum injection time was set to 100 ms for the full proteome and label-free glycopeptides and 200 ms for TMT-labeled glycopeptides. The AGC target was set to $1 \times 10^5$ charges for the full proteome and label-free glycopeptides and to $2.5 \times 10^5$ charges for TMT-labeled glycopeptides.

## Data analysis

**Database search.** Raw files were converted to mzML format using MSConvert from Proteowizard[66], using peak picking from the vendor algorithm and keeping the 1,000 most intense peaks. Files were then searched using MSFragger version 4.0 in Fragpipe version 20.0 against the Swiss-Prot *Homo sapiens* database (20,443 entries) in the case of human cell samples or the *Mus musculus* reference proteome with one protein per gene sequence (22,084 entries) in the case of mouse tissue samples.

The default glyco-N-HCD, glyco-O-HCD and glyco-N-TMT workflows were used for the label-free N-glyco, label-free O-glyco and TMT-labeled N-glyco database searches, respectively. After mass calibration, the top 300 peaks were used for database search and the MS2 fragment mass tolerance was lowered to 7 ppm for the MSFragger search, while both precursor and fragment mass tolerance were lowered to 10 ppm for the PTM-Shepherd glycan assignment.

In the case of *N*-glycosylation, a custom database containing 1,038 unique glycan compositions (Supplementary Data 1) was used while the default MSFragger database was used for *O*-glycosylation. The 1,038 *N*-glycan compositions database consisted of the 1,799 glycan compositions database downloaded from GlyTouCan.org (https://glytoucan.org/), arbitrarily filtered such that the following applied:

- The number of HexNac molecules should be inferior or equal to two times the number of Hex + 1;
- The number of NeuGc + NeuAc or fucose molecules should be inferior or equal to the number of (Hex/2 + 1);
- In total the number of NeuGc + NeuAc + fucose molecules should be inferior or equal to 5.

A glycopeptide FDR of 0.01 was used and, after glycan composition assignment using PTM-Shepherd, the glycan $q$ value cutoff was set to 0.05. The same glycan $q$-value cutoff was recently used by the developers of MSFragger[67].

In the case of TMT-labeled samples, TMTpro modification was fixed on lysine and variable on the peptide N terminus and channel normalization was disabled. The psm.tsv and protein.tsv output files were used for subsequent data analysis. In the case of *O*-glycosylation, all glycopeptides containing the *N*-glycosylation sequon (N-X-S/T) and peptides without a target match were removed to prevent misassignment.

Peptides with a unique amino acid sequence combined with a unique glycan composition were considered as unique glycopeptides. Spectra could be visualized with the PDV-viewer integrated in Fragpipe.

For the Byonic search of the stepped HCD data (mouse brain fractionated by PGC into 48 fractions), the calibrated mzmL files were used (output from Fragpipe). The default Byonic search parameters were used and the results were further filtered as in Riley et al.[16]: 1% FDR using the 2D-FDR score, Byonic score above 150, |logProb| value above 1 and a minimum peptide length of six amino acids.

**Glycan type classification.** For downstream analysis, we extracted glycopeptides from the psm.tsv files and categorized them into seven or eight glycan classes for *N*-glycosylation and *O*-glycosylation, respectively. The categories were defined as follows (hierarchical classification): (1) phospho, containing a phosphate group; (2) sialylated, containing a NeuAc or NeuGc sugar; (3) fucosylated, containing fucose; (4) high mannose, combining a HexNAc(2) core with 4–12 hexose molecules; (5) paucimannose, combining a HexNAc(2) core with 1–3 hexose molecules; (6) small, composed of only a core with less than three HexNAc molecules; and (7) complex or hybrid, comprising the remainder. For *O*-glycosylation data, categories were defined as follows (hierarchical classification): (1) *O*-phospho, containing a phosphate group; (2) *O*-sialylated, containing a NeuAc or NeuGc sugar; (3) *O*-fucosylated, containing fucose; (4) *O*-sulfated, containing a sulfation; (5) *O*-GlcNAc, for glycans containing a single HexNAc; (6) *O*-hexose, composed of a core with less than three HexNAc molecules and at least one hexose molecule; and (6) *O*-hybrid, comprising the remainder. This means that a glycoform being both sialylated and fucosylated would be part of the sialylated glycan class but not counted as a fucosylated glycoform.

**Pathway and domain enrichment analyses.** All included enrichment analyses were performed using the STRINGdb R package[68], with the full *M. musculus* or *H. sapiens* network. The default hypergeometric test and *P*-value adjustment procedure was used. Significant pathways were extracted using a 5% FDR cutoff. For the domain enrichment analysis, we performed an overrepresentation analysis on the contingency matrix of InterPro domains[69] and glycan classes using Fisher's exact test (Supplementary Data 9). *P* values were adjusted for multiple testing using the Benjamini–Hochberg method. Significant domains were extracted using an adjusted *P*-value cutoff of 0.1%.

**Comparison of glycosylation profiles.** To compare the similarity of glycosylation profiles of sites on one protein, we prefiltered for proteins with more than one site. We calculated the pairwise correlation of the number of all possible observed modifications between all sites of a protein using the Kendall rank correlation coefficient. To generate random correlations, we shuffled all glycosites to randomly assign glycan compositions to them. To test whether the observed difference of random and true correlations of sites is significant, we performed a Wilcoxon signed-rank test.

**Comparison to database information.** Evidence annotation of known glycosites was downloaded from UniProt and PhosphoSitePlus[70]. Additional information regarding the type of evidence (for example, sequence analysis) was only available for the UniProt-derived annotations. To annotate each glycoprotein with a subcellular localization, we obtained data from the COMPARTMENTS database[71]. We defined the nucleus, cytoplasm, mitochondrion, ER, Golgi apparatus, lysosome, plasma membrane and extracellular as locations of interest and used those to filter the database. Additionally, we focused only on locations with a confidence score exceeding 2.5. To ensure a predominantly unique localization per protein, we removed additional locations of plasma membrane or extracellular proteins considering these compartments as the final location for a glycoprotein. To check proteins belonging to a specific organelle (for example, lysosome), we used the

AnnotationDbi R package to download respective GO term information (https://bioconductor.org/packages/AnnotationDbi). Glycoenzyme annotations were downloaded from the Glygen database[72].

To investigate the distance between phosphosites and glycosites for different glycan types, we downloaded all known *M. musculus* phosphosites from UniProt. We defined a phosphosite as close if it was located ±5 residues around the glycosite in the case of *N*-glycosylation and ±5 residues around the peptide start position in the case of *O*-glycosylation.

**2FF treatment data analysis.** For the quantitative proteomics and glycoproteomics data of HEK293f cells exposed to 2FF over time, we first normalized the TMT reporter intensities for glycopeptides and proteins with complete quantification across all conditions (that is, TMT reporter ion intensity different from zero for all channels) using the normalizeVSN function of the limma R package[73]. To check the reproducibility of quantification, the pairwise Spearman correlation of all conditions and replicates was calculated. We corrected glycopeptide intensities per protein by removing abundance-derived intensity using a linear regression for all glycopeptides with a matched total protein intensity. The corrected and normalized glycopeptide intensities and the normalized protein intensities were used to generate PCA plots and the differential analysis using the limma R package[73]. For the design of the differential expression (DE) analysis, we considered the treatment condition and duration and the sample replicate. Contrasts were set for the comparison of the DMSO control against 2FF treatment per time point. A significant change was assigned for glycopeptides with an absolute $\log_2$ fold change $> \log_2(1.5)$ and an adjusted *P* value $< 0.05$. For clustering analysis, $\log_2$ fold changes of glycopeptides that were significantly affected for at least one time point were clustered using neural gas clustering implemented in the cclust R package (https://cran.rstudio.com/web/packages/cclust/index.html). To explore down-regulation of fucosylation in detail, $\log_2$ fold changes of fucosylated peptides that were significantly downregulated for at least one time point were clustered as described before.

**Surface accessibility data analysis.** The quantitative glycoproteomics data of HEK293T cells after PNGase or proteinase K treatment were analyzed separately per treatment dataset before comparing the two after differential abundance analysis. For both datasets, we first normalized the TMT reporter intensities for glycopeptides and proteins using the normalizeQuantiles function of the limma R package[73]. To check the reproducibility of quantification, the pairwise Spearman correlation of all control samples of the proteinase K experiment was calculated. The normalized glycopeptide intensities were used for the differential analysis using the limma R package[73]. Data distribution was assumed to be normal but this was not formally tested. For both datasets, we only considered glycopeptides with complete quantification across all conditions. For the design of the DE analysis, we considered the treatment and the replicate of the sample. Contrasts were set for the comparison of the control and treatment samples. A significant change was assigned for glycopeptides with an absolute $\log_2$ fold change $> \log_2(1.5)$ and an adjusted *P* value $< 0.05$. If we detected more than one significantly upregulated glycopeptide on a given site, we classified it as affected by the treatment and used this classification to select glycopeptides for further analysis.

**Tissue data analysis.** For the quantitative proteomics and glycoproteomics data of different mice tissues, we first removed the low-abundance signals of glycopeptides to avoid detection because of TMT channel leaking. To do so, we computed the $\log_2$ fold change for each TMT channel to the TMT channel exhibiting the highest relative intensity per glycopeptide and per protein in the full proteome data. Then, we filtered out glycopeptides and proteins with a $\log_2$ fold change cutoff $< \log_2(1/16)$ between two TMT channels to eliminate potential

noise. To determine the number of tissues in which a glycopeptide was identified, we filtered for glycopeptides with quantification in all three replicates of a given tissue. Next, we adjusted glycopeptide intensity for protein abundance while maintaining differences between glycosites. To this end, we calculated a corrected intensity as follows:

$$\text{Intensity}_{corrected} = \frac{\dfrac{\text{Glycopeptide intensity}}{\text{Protein intensity}}}{\dfrac{\text{Median glycopeptide } intensity}{\text{Median protein intensity}}} \\ \times \text{Median glycopeptide intensity}$$

These corrected intensities were normalized using the normalizeVSN function of the limma R package[73] and then used to generate PCA plots and calculate the fractional abundance of glycopeptides per tissue. To compare glycosylation profiles of glycosites across tissues and, thus, determine their tissue specificity, we computed the pairwise Pearson correlation of glycosylation patterns (correlating the relative intensities of all glycopeptides on a given glycosite between tissues) for sites having more than five quantified glycopeptides. We generated this correlation information for either both mice combined or per mouse. To assess differences of tissue glycosylation between the two profiled mice, we used the normalized and corrected glycopeptide intensities for differential abundance analysis using the limma R package[73]. Data distribution was assumed to be normal but this was not formally tested. For the design of the DE analysis, we considered the tissue, the mouse and the replicate of the sample. Contrasts were set for the comparison of the two mice per tissue. A significant change was assigned for glycopeptides with an absolute $\log_2$ fold change $> \log_2(2)$ and an adjusted *P* value $< 0.05$.

**Solubility proteome profiling data analysis.** For the quantitative proteomics and glycoproteomics data of the mouse brain solubility profiling experiment, we first computed solubility ratios per glycopeptide and protein. To do so, we compared glycopeptide or protein intensity in the NP40 lysis condition compared to the SDS lysis condition ($\log_2$ NP40/SDS ratio). Next, we computed the differential solubility per glycopeptide by comparing its solubility ratio to the solubility ratio of the protein in the full proteome dataset using limma. To do this, we first filtered the glycopeptide data for entries with a matched measured protein and then normalized glycopeptide and protein solubility ratios using the normalizeMedianValues of the limma R package. Data distribution was assumed to be normal but this was not formally tested. For the design of the differential analysis, we considered the dataset (glycoproteome or full proteome). Contrasts were set for the comparison of the two datasets ($\log_2$ fold change glycoform solubility − protein solubility). A significant change was assigned for glycopeptides with an absolute $\log_2$ fold change $> \log_2(1.5)$ and an adjusted *P* value $< 0.05$.

**Microbiome data analysis.** For the quantitative proteomics and glycoproteomics data of brains of mice exposed to different microbiome species, we first normalized the TMT reporter intensities for glycopeptides and proteins using the normalizeVSN function of the limma R package[73]. To check the reproducibility of quantification, the pairwise Spearman correlation of all conditions and replicates was calculated. We corrected glycopeptide intensities per protein by removing abundance-derived intensity using a linear regression for all glycopeptides with a matched total protein intensity. The corrected and normalized glycopeptide intensities and the normalized protein intensities were used to generate PCA plots and the differential analysis using the limma R package[73]. The intensity matrices were filtered for features with complete quantification across all conditions (TMT reporter ion intensity different from 0 for all channels). Data distribution was assumed to be normal but this was not formally tested. For the design of the DE analysis, we considered the microbiome group

and the sample replicate. Contrasts were set for the comparison of the different microbiome groups. A significant change was assigned for glycopeptides or proteins with an absolute $\log_2$ fold change $> \log_2(1.5)$ and an adjusted $P$ value $< 0.05$.

**Thermal proteome profiling data analysis.** For the mouse brain thermal proteome profiling experiment, we combined three replicates measured in three TMT sets by calculating an internal reference sample per TMT set on the basis of protein intensity in the first temperature. The internal reference corresponds to the mean protein intensity divided by the mean protein intensity in the given replicate. This internal reference was then used to align the data of the three replicates and remove the batch effect of the used TMT set. Subsequently, the corrected data were normalized using the normalizeQuantiles function of the limma R package. The normalized protein intensities were then used for statistical analysis using GPMelt[74] to compare thermal protein stability in brains of germ-free mice compared to mice exposed to a microbiome (eight-member community). To calculate the ΔAUC between the two conditions, we first subtracted the abundance change computed on the basis of the normalized intensities of the first two temperatures from all temperatures to consider thermal (de)stabilization effects. The ΔAUC then corresponds to the difference of summed intensities of the two conditions. Data distribution was assumed to be normal but this was not formally tested.

**Structural and functional analysis of lowly and highly glycosylated asparagines.** We filtered and divided glycosylated asparagines identified in the deep glycoproteomics analysis into two groups: (1) lowly glycosylated, for asparagines with a single glycoform, and (2) highly glycosylated, for asparagines with a number of glycoforms above the 90th percentile of the distribution of number of glycoforms per dataset. This resulted in an upper threshold of 44 glycoforms for mouse brain data and 36 glycoforms for human cell lines data. The total number of sites per dataset was 2,203 lowly glycosylated and 747 highly glycosylated sites in mouse data and 1,196 lowly glycosylated and 488 highly glycosylated sites in human data.

Next, we used the structuremap python package to generate all site-level structural annotations (version 0.0.9) (https://github.com/MannLabs/structuremap)[26]. We retrieved predicted structures for all proteins identified in the N-glycosylation data from the AlphaFold Protein Structure Database[75]. Residues were annotated using predicted secondary structures and a pPSE metric described previously[26], which scores the order for each residue on the basis of the number of residues in their structural proximity, defined by a directional partial sphere with a specific radius (Å) and angle (°), which also considers AlphaFold prediction error. Two pPSE metrics were generated: one for a wider and bigger partial sphere (24 Å, 180°) and another for a narrower and smaller partial sphere (12 Å, 70°). As suggested in the original study[26], we used a threshold of 34.27 on the smoothed pPSE (180°, 24 Å) to determine IDRs. We also retrieved topological domain annotations from UniProt. We filtered these annotations to include only the two classes of interest, which were 'cytoplasmic' and 'extracellular'.

We built a model to predict the extent of site glycosylation (highly glycosylated or lowly glycosylated) using structural features as input. We used a model based on gradient-boosting decision trees as implemented by the XGBoost python package[76] (version 2.0.3) (https://github.com/dmlc/xgboost). Gradient boosting is a machine learning technique that builds an ensemble of decision trees in sequence, where each tree attempts to correct errors made by the previous one, resulting in a strong predictive model. The model was created using default XGBoost parameters, trained on mouse data and validated in human data, under the assumption that a structural signal would be conserved across organisms. To evaluate the model's performance, we used the AUROC from the scikit-learn package.

The AUROC is a metric that measures how well the model can distinguish between two classes (in this case, highly and lowly glycosylated), with a score of 1 representing perfect separation and a score of 0.5 representing random guessing.

For the functional analysis of highly and lowly glycosylated asparagines, we retrieved AlphaMissense predictions for all possible amino acid substitutions in the human proteome obtained online (https://console.cloud.google.com/storage/browser/dm_alphamissense)[27]. The average AlphaMissense score across potential amino acid substitutions was then used for each glycosylated asparagine. This score served as the probability of a mutation at a specific site being clinically pathogenic.

**Experiment design and statistical rationale.** All experiments were performed in triplicate to perform statistical analysis. No statistical methods were used to predetermine sample sizes but our sample sizes are similar to those routinely reported in proteomics publications. Data collection and analysis were not performed blind to the conditions of the experiments to ensure correct labeling of the samples. No samples were removed before data analysis.

### Reporting summary

Further information on research design is available in the Nature Portfolio Reporting Summary linked to this article.

## Data availability

All results and data are available online (https://apps.embl.de/glycoapp/). All raw files, search parameters and search outputs were deposited to the ProteomeXchange Consortium through the PRIDE partner repository with the dataset identifier PXD052447.

## Code availability

All code used to perform the computational analyses described and to reproduce the figures is available from GitHub (https://github.com/miralea/DQGlyco) and archived on Zenodo (https://doi.org/10.5281/zenodo.14443693)[77].

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

## Acknowledgements

We thank A. Mateus and P. Cossart for insightful discussions and feedback on the paper. This work was supported by the EMBL. C.M.P. was supported by a fellowship from the EMBL Interdisciplinary Postdoc (EI3POD) program under Marie Skłodowska-Curie Actions COFUND (grant number 664726). M.G.-R. is supported through state funds approved by the State Parliament of Baden-Württemberg for the Innovation Campus Health+Life Science Alliance Heidelberg Mannheim. A.T. is supported by a European Research Council (ERC) consolidator grant, uCARE. M.M.S. is supported by the Allen Distinguished Investigator award through the Paul G. Allen Frontiers Group. M.Z. is supported by an ERC starting grant, GutTransForm (grant no. 101078353). We would like to thank all members of the M.M.S., M.Z. and A.T. groups for helpful discussions, the Proteomics Core Facility at the EMBL for expert help and G. Maftei for assistance with animal experiments.

## Author contributions

C.M.P. and M.M.S. designed the study. C.M.P. developed the methodology and performed the (glyco)proteomics experiments. C.M.P., M.L.B. and M.G-R. developed the data analysis strategies and analyzed the data. A.B-N. performed the mouse experiments and was supervised by M.Z. I.B. cultured the human cells. C.L.S. and M.L.B. analyzed the thermal proteome profiling data. M.L.B. developed the data visualization application. I.B. finalized the figures for publication. M.Z. and A.T. provided scientific input on the study design and data interpretation. C.M.P. and M.M.S. drafted the paper with contribution from M.L.B., M.G-R., I.B., M.Z. and A.T., which was reviewed and edited by all authors. C.M.P. and M.M.S. supervised the study.

## Funding

## Competing interests

The authors declare no competing interests.

## Additional information

**Correspondence and requests for materials** should be addressed to Clément M. Potel or Mikhail M. Savitski.

Clement Potel

# Reporting Summary

## Statistics

For all statistical analyses, confirm that the following items are present in the figure legend, table legend, main text, or Methods section.

| n/a | Confirmed | |
|---|---|---|
| ☐ | ☒ | The exact sample size (*n*) for each experimental group/condition, given as a discrete number and unit of measurement |
| ☐ | ☒ | A statement on whether measurements were taken from distinct samples or whether the same sample was measured repeatedly |
| ☐ | ☒ | The statistical test(s) used AND whether they are one- or two-sided<br>*Only common tests should be described solely by name; describe more complex techniques in the Methods section.* |
| ☐ | ☒ | A description of all covariates tested |
| ☐ | ☒ | A description of any assumptions or corrections, such as tests of normality and adjustment for multiple comparisons |
| ☐ | ☒ | A full description of the statistical parameters including central tendency (e.g. means) or other basic estimates (e.g. regression coefficient) AND variation (e.g. standard deviation) or associated estimates of uncertainty (e.g. confidence intervals) |
| ☐ | ☒ | For null hypothesis testing, the test statistic (e.g. *F*, *t*, *r*) with confidence intervals, effect sizes, degrees of freedom and *P* value noted<br>*Give P values as exact values whenever suitable.* |
| ☒ | ☐ | For Bayesian analysis, information on the choice of priors and Markov chain Monte Carlo settings |
| ☒ | ☐ | For hierarchical and complex designs, identification of the appropriate level for tests and full reporting of outcomes |
| ☐ | ☒ | Estimates of effect sizes (e.g. Cohen's *d*, Pearson's *r*), indicating how they were calculated |

*Our web collection on statistics for biologists contains articles on many of the points above.*

## Software and code

Policy information about availability of computer code

| | |
|---|---|
| Data collection | Vendor acquisition softwares: Thermo Scientific Xcalibur 4.6 and Tune 4.2 |
| Data analysis | MSFragger v20<br>MSConvert 3.0.19057-5e3190638<br>Python 3.9.12<br>R 4.2.1<br>Structuremap package (v0.0.9)<br>XGBoost python package (v2.0.3)<br>All code used to perform the computational analyses described and to reproduce the figures is available at https://github.com/miralea/DQGlyco |

For manuscripts utilizing custom algorithms or software that are central to the research but not yet described in published literature, software must be made available to editors and reviewers. We strongly encourage code deposition in a community repository (e.g. GitHub). See the Nature Portfolio guidelines for submitting code & software for further information.

## Data

Policy information about availability of data

All manuscripts must include a data availability statement. This statement should provide the following information, where applicable:
- Accession codes, unique identifiers, or web links for publicly available datasets
- A description of any restrictions on data availability
- For clinical datasets or third party data, please ensure that the statement adheres to our policy

All raw files, databases (downloaded from Uniprot:  Swissprot Homo sapiens database (20,443 entries) and Mus musculus reference proteome with one protein per gene sequence (22,084 entries)) and search outputs were deposited to the ProteomeXchange Consortium via the PRIDE partner repository with the dataset identifier PXD052447.

## Human research participants

Policy information about studies involving human research participants and Sex and Gender in Research.

| | |
|---|---|
| Reporting on sex and gender | NA |
| Population characteristics | NA |
| Recruitment | NA |
| Ethics oversight | NA |

Note that full information on the approval of the study protocol must also be provided in the manuscript.

# Field-specific reporting

Please select the one below that is the best fit for your research. If you are not sure, read the appropriate sections before making your selection.

☒ Life sciences  ☐ Behavioural & social sciences  ☐ Ecological, evolutionary & environmental sciences

For a reference copy of the document with all sections, see nature.com/documents/nr-reporting-summary-flat.pdf

# Life sciences study design

All studies must disclose on these points even when the disclosure is negative.

| | |
|---|---|
| Sample size | No sample size calculation was performed. All experiments were performed in triplicates to perform statistical analysis. No statistical methods were used to pre-determine sample sizes but our sample sizes are similar to those routinely reported in proteomics publications. |
| Data exclusions | Peptides and proteins with incomplete TMT quantification were excluded, as well as proteins that matched the contaminant database. |
| Replication | All proteomics experiments were conducted in biological triplicates and replications were successful. |
| Randomization | No randomization was performed, as standard for this type of experiments. Experiments were measured sequentially on mass spectrometers as is the general practice in proteomics. |
| Blinding | No blinding was performed as this study would not be impacted by blinding. |

# Reporting for specific materials, systems and methods

We require information from authors about some types of materials, experimental systems and methods used in many studies. Here, indicate whether each material, system or method listed is relevant to your study. If you are not sure if a list item applies to your research, read the appropriate section before selecting a response.

## Materials & experimental systems

| n/a | Involved in the study |
|-----|----------------------|
| ☒ | ☐ Antibodies |
| ☐ | ☒ Eukaryotic cell lines |
| ☒ | ☐ Palaeontology and archaeology |
| ☐ | ☒ Animals and other organisms |
| ☒ | ☐ Clinical data |
| ☒ | ☐ Dual use research of concern |

## Methods

| n/a | Involved in the study |
|-----|----------------------|
| ☒ | ☐ ChIP-seq |
| ☒ | ☐ Flow cytometry |
| ☒ | ☐ MRI-based neuroimaging |

## Eukaryotic cell lines

Policy information about cell lines and Sex and Gender in Research

| | |
|---|---|
| Cell line source(s) | Hela-Kyoto (Schmitz et al., 2010) - was a gift from Jan Ellenberg's group, EMBL - original cell line (RRID: CVCL_1922) was received as gift from Prof. S. Narumiya, Kyoto University and authenticated by sequencing.<br>HEK293T cells (ATCC-CRL-3216) were obtained from the American Type Culture Collection (ATCC). Expi293F were obtained from Thermo Fisher Scientific. HEK293 and HeLa-Kyoto are not on the list of commonly misidentified cell lines. |
| Authentication | Hela-Kyoto cells have been authenticated by sequencing. HEK293 cells were not authenticated. |
| Mycoplasma contamination | All cell lines tested negative for mycoplasma contamination. |
| Commonly misidentified lines<br>(See ICLAC register) | No commonly misidentified lines were used in this study. |

## Animals and other research organisms

Policy information about studies involving animals; ARRIVE guidelines recommended for reporting animal research, and Sex and Gender in Research

| | |
|---|---|
| Laboratory animals | Germfree (GF) C57BL/6 mice were maintained and bred in gnotobiotic isolators (CbC) with a 12-hour light/dark cycle. GF status was monitored by PCR (using 16S primer pair) and culture-based methods. Mice were provided with standard, autoclaved chow (1318 P FORTI, Altromin) ad libitum. Animals of mixed gender at an age between 8 and 17 weeks were used for all experiments. |
| Wild animals | The study did not involved wild animals. |
| Reporting on sex | Mice of both male and female sex were used for this study. |
| Field-collected samples | The study did not involved samples collected from the field. |
| Ethics oversight | All mouse experiments were performed using approved protocols by the EMBL ethics committee (licence 21-002_HD_MZ). |

Note that full information on the approval of the study protocol must also be provided in the manuscript.

