## [Peer Review File · Nature Structural & Molecular Biology]

Uncovering protein glycosylation dynamics and heterogeneity via Deep Quantitative Glycoprofiling (DQGlyco)

Corresponding Author: Dr Mikhail Savitski

Version 0:

Decision Letter:

12th Sep 2024

Dear Dr. Savitski,

Thank you again for submitting your manuscript "Deep quantitative glycoproteomics reveals gut microbiome induced remodeling of the brain glycoproteome". We now have comments (below) from the 3 reviewers who evaluated your paper. In light of those reports, we remain interested in your study and would like to see your response to the comments of the referees, in the form of a revised manuscript.

In general, we would ask that you address all discussion points and methodological concerns raised by the reviewers. From an editorial perspective, we would also ask that you include additional mechanistic or functional data involving one or more of the proteins identified by your preliminary analysis (ex. EEAT2, in terms of mouse brain health). We feel that including this downstream validation would help to expand the impact of the study, and serve as a key example of how your approach could be used to glean novel biological insight (while also addressing the comments from Reviewer #1). We would also encourage you to include additional applications of DQGlyco to other tissues or disease contexts, to further evaluate its generalizability, though we would place greater emphasis on incorporating functional data.

In any case, please be sure to address/respond to all concerns of the referees in full in a point-by-point response and highlight all changes in the revised manuscript text file. We are committed to providing a fair and constructive peer-review process. Do not hesitate to contact us if there are specific requests from the reviewers that you believe are technically impossible or unlikely to yield a meaningful outcome.

We appreciate the requested revisions are extensive. We thus expect to see your revised manuscript within 6 months. If you cannot send it within this time, please let us know. We will be happy to consider your revision as long as nothing similar has been accepted for publication at NSMB or published elsewhere. Should your manuscript be substantially delayed without notifying us in advance and your article is eventually published, the received date would be that of the revised, not the original, version.

Reporting Summary:

When submitting the revised version of your manuscript, please pay close attention to our [href="https://www.nature.com/nature-portfolio/editorial-policies/image-integrity">Digital Image Integrity Guidelines. and to the following points below:](https://www.nature.com/nature-portfolio/editorial-policies/image-integrity)

Please note that all key data shown in the main figures as cropped gels or blots should be presented in uncropped form, with molecular weight markers. These data can be aggregated into a single supplementary figure. While these data can be displayed in a relatively informal style, they must refer back to the relevant figures. These data should be submitted with the last revision, prior to acceptance, but you may want to start putting it together at this point.

We require deposition of coordinates (and, in the case of crystal structures, structure factors) into the Protein Data Bank with the designation of immediate release upon publication (HPUB). Electron microscopy-derived density maps and coordinate data must be deposited in EMDB and released upon publication. Deposition and immediate release of NMR chemical shift assignments are highly encouraged. Deposition of deep sequencing and microarray data is mandatory, and the datasets must be released prior to or upon publication. To avoid delays in publication, dataset accession numbers must be supplied with the final accepted manuscript and appropriate release dates must be indicated at the galley proof stage. Please find the complete NRG policies on data availability at <http://www.nature.com/authors/policies/availability.html>.

Link Redacted

Best regards,

George Inglis

George Inglis, PhD
Senior Editor

[Research Cross-Journal Editorial Team](https://www.nature.com/nsmb/research-cross-journal-editorial-team)
Nature Structural & Molecular Biology

Referee expertise:

Referee #1: Glycoproteomics in cancer

Referee #2: Glycoproteomic methods

Referee #3: Proteomics in host-pathogen interactions

Reviewers' Comments:

Reviewer #1:

Remarks to the Author:

This manuscript describes the development and application of Deep Quantitative Glycoproteomics (DQGlyco), a high-throughput sample preparation method to enrich glycopeptides combined with mass spectrometry analysis. DQGlyco is applied to the mouse brain tissue where an impressive number of unique glycopeptides are identified. Analysis is performed on this dataset to determine correlation between glycosylation microheterogeneity and protein structure. Tissue-specificity of

glycosylation is also examined. Lastly mouse brains with different gut microbiota conditions are analyzed and claims are made that different microbiome conditions result in extensive remodeling of the brain glycoproteome.

Overall this is an interesting study with an impressive depth of analysis. The data set is also likely to be useful to the scientific community. However, my major criticism of the manuscript is that it covers several large areas (method development, protein structure and glycosylation, glycosylation differences between tissues, microbiome effect on mouse brain tissue), but is lacking substantial depth in any given area. The manuscript is largely descriptive and it is somewhat difficult to follow what the main point is. Some additional specific comments:

- 1) The first two paragraphs of Results section are out of place and don't belong in the Results section.
- 2) The description of the AI analysis for protein structure and glycosylation not well described. It was difficult to follow the analysis and findings.
- 3) The claim that gut microbiome changes glycosylation in brain proteins is a substantial stretch.
- 4) Cell line authentication was not described.
- 5) Scores for glycopeptide ID's in supplementary tables should be provided.
- 6) The classification of glycopeptides is not clear. Is something double counted if it is both sialylated and fucosylated for example?
- 7) The number of biological and technical replicates for each experiment is not clear.
- 8) While substantially more glycopeptides are identified in this study compared to previous studies, glycopeptides in the mouse brain have been studied quite extensively, thereby limiting the novelty and making this study somewhat incremental.

Reviewer #2:

Remarks to the Author:

This manuscript by Potel et al. describes a new workflow for glycoproteomics, combining an improved method for glycopeptide enrichment with optimized deep fractionation and mass spectrometric acquisition methods to profile an order of magnitude more glycopeptides than previous studies in similar sample types. The authors then use this exceptionally deep glycoproteomic profiling to provide a variety of biological insights, providing an excellent illustration of the potential of glycoproteomics. The manuscript is compelling and clearly supports these rather exceptional claims with clear and convincing evidence throughout. Other than a few minor clarifications and areas for additional discussion, detailed below, I see no barriers to publication.

Specific comments:

1. The improvement in glycopeptides identified from the PBA enrichment and optimized MS1 scan range, while impressive, appear to be smaller than the gain from the PGC fractionation in Fig 1. This is particularly notable given that the previous studies being compared (Riley et al, etc.) also used deep fractionation, but via high pH C18 rather than PGC. Some additional discussion of the interplay of these factors would be very valuable to clarify whether similarly deep profiling could be accomplished by PGC fractionation with traditional enrichment methods (HILIC, lectins) and/or without the nucleic acid precipitation, or if it is really the combination of all of these improvements together that is necessary to achieve the order of magnitude increase in glycopeptides identified. Additionally, some discussion of any challenges or key considerations for implementing this method in other labs would provide a valuable perspective on the potential of the method.
2. The discussion of peptides bearing O-GlcNAc modifications should clarify explicitly that a single HexNAc can be either O-GlcNAc or O-GalNAc, and while the oxonium ion ratios in Fig S2G clearly show a difference in trend between intra- and extracellular proteins, it is not as clear-cut as described in the text given that median ratio for extracellular proteins remains well below 1. Calling these putative O-GlcNAc modifications would be more accurate, and the limitations of oxonium ion analysis (namely, co-fragmentation) should be mentioned.
3. The choice to use a more liberal glycan q-value threshold of 0.05 is not unreasonable, but the text should note that the confidence in the glycan compositions identified is lower than the peptides, given the higher chosen threshold. A comparison of the results for one analysis between setting a threshold of 0.01 and 0.05 could be a useful addition to the supporting information to inform the reader as to the impact of this choice.

Reviewer #3:

Remarks to the Author:

Potel et al. present a study focused on deep quantitative glycoproteomics including a number of technical advances with an application related to microbiome induced changes in mouse brain. The technical advances include a glycopeptide enrichment based on existing covalent chemistry (PBA) combined with highthroughput plate-based sample processing methods that reduces contaminating species such as nucleic acids. While the enrichment methodology already contains

several improvements the authors then employ a multidimensional approach using porous graphitic carbon which substantially improves glycopeptide coverage. The authors benchmark by comparison to other recent papers demonstrating the improvements, and importantly also show that their method demonstrates less bias for specific glycopeptide/protein classes than existing methods. They assess site-level heterogeneity and protein level heterogeneity as well as setting glycoform variation in the context of predicted structural features including a machine learning model to predict site level heterogeneity based on structural features. Finally, the authors integrate isobaric labelling with their multidimensional method, including labelling adaptations to minimize label requirements, to enable quantitative comparisons. The strategy is then applied to microbiome perturbation experiment in mouse. They show first that there strong distinction in tissue level specificity of the glycoproteome, and further differences in specific glycoforms in brains of mice depending on microbiome status.

In my view this manuscript is very strong and represents a very substantial improvement over the state of the art in glycoproteomics methodology. The increase in depth achieved, and removal of systematic bias for glycopeptide coverage, by very careful and systematic optimization of purification, sample prep, and MS methodology is fairly remarkable. I think the manuscript is very logical and well written, well illustrated, and packs in a large body of work that clearly shows the progression of concepts. The conclusions are surrounding micro- and macro-heterogeneity (also related to structure), tissue specificity, and effects of microbiome on brain are novel and striking. All data and code are made available in appropriate repositories, and results are explorable via a web app. My view is that this will have an immediate and strong impact on the field. As such, I think there is rather little to complain about and I suggest that the manuscript should be accepted in short order. I have put a few minor clarifying comments/questions below.

1. One technical question would be whether anything semi-quantitative can be said about the distribution of glycoforms at the site level. Given that the most extreme case they observe 600+ glycoforms at a single site, can anything be said about whether any forms are more common (akin to major/minor isoform or splice variants). While the response factor for a given peptide sequence dependent on the glycoform it harbours will presumably change, but perhaps a comparison of intensities would be informative. Is anything known about this issue? Could this be summarized at glycoform class level?
2. The description of the methodological advances in the first results section could be slightly more easy to digest if revised elaborating further all steps. While its fine to leave majority of details in the methods section, I think it would be more understandable if conceptual aspects were more completely stated (i.e. a key advance is the removal of nucleic acids by precipitation but the details of how this is done is breezed over, enrichment by covalent coupling to PBA is described but not method for elution, etc). My experience on reading was to have to dig around in methods several times to understand the complete workflow conceptually so a revision of this section for conceptual clarity might help the less initiated reader.
3. I am missing a bit some info on how much input material is needed for both their single shot or multidimensional strategies (i.e. how much protein from how many cells and how much tissue is needed/optimal). This would be an important consideration for anyone wanting to try this.
4. In methods they mention that identified O-linked peptides containing a N-linked motif were removed. It would be interesting to know the rate at which rate this occurs.

signed
Ben Collins

Version 1:

Decision Letter:

11th Oct 2024

Dear Dr. Savitski,

Thank you for submitting your revised manuscript, "Deep quantitative glycoproteomics reveals gut microbiome induced remodeling of the brain glycoproteome". After careful consideration and discussion with my colleagues, I am sorry to have to tell you that we do not feel that the referees' comments have been sufficiently addressed to justify sending this revision back for peer review.

This unusual course of action is taken occasionally to avoid unproductive rounds of review that result in reviewer fatigue and damage the chances of the manuscript obtaining a fair and objective evaluation. Such situations are not in an author's best interest so we try to avoid them when it seems prudent to do so.

In order to consider this manuscript further we would request that you please do your best to fully address all of the comments of the editors and reviewers. In particular, we would again strongly encourage you to include additional mechanistic or functional data exploring the downstream relevance of at least one of your target proteins from the brain-microbiome analysis (ex. EEAT2). While we appreciate the inclusion of the new data in HEK293 cells, we would still ask that you apply DQGlyco to specific disease models or primary tissues, to better highlight its general applicability. On a related note, please update the main text (Title, Abstract, and Discussion) to emphasize DQGlyco itself, rather than the brain glycoproteomics analysis.

Should you be able to adequately respond to these and the reviewers' other concerns, we would be happy to look at a revised manuscript, though we would ask that you provide a marked-up version of the manuscript file alongside the updated rebuttal letter.

As before, we shall hope to receive your revised version with 6 months, though please let us know if you would need additional time. We will be happy to consider your revision so long as nothing similar has been accepted for publication at Nature Structural & Molecular Biology or published elsewhere. Should your manuscript be substantially delayed without notifying us in advance and your article is eventually published, the received date may be that of the revised, not the original, version.

If you are not interested in submitting a suitably revised manuscript in the future please let me know immediately so we can close your file. If you have any questions, please contact me.

Please use the link below to submit a suitably revised manuscript and updated response to referees when they are ready.

Link Redacted

Best regards,

George

George Inglis, PhD
Senior Editor

<https://www.nature.com/nsmb/research-cross-journal-editorial-team> Research Cross-Journal Editorial Team
Nature Structural & Molecular Biology

Version 2:

Decision Letter:

Dear Dr. Savitski,

Thank you for submitting your revised manuscript "Deep Quantitative Glycoprofiling: Uncovering Glycosylation Dynamics and Microheterogeneity" (NSMB-A49568B). It has now been seen by the original referees and their comments are below. Please note that Reviewer #2 was unable to comment on the revised manuscript, so we asked Reviewer #1 to assess the rebuttal in their place. In any case, the reviewers find that the paper has improved in revision, and therefore we'll be happy in principle to publish it in Nature Structural & Molecular Biology, pending minor revisions to comply with our editorial and formatting guidelines. However, we would prefer to consider this as a Technical Report going forward, rather than an Article.

We are now performing detailed checks on your paper and will aim to send you a checklist detailing our editorial and formatting requirements in about a week. Please do not upload the final materials and make any revisions until you receive this additional information from us.

To facilitate our work at this stage, it is important that we have a copy of the main text as a word file. If you could please send along a word version of this file as soon as possible, we would greatly appreciate it; please make sure to copy the NSMB account (cc'ed above).

Best regards,

George

George Inglis, PhD
Senior Editor

<https://www.nature.com/nsmb/research-cross-journal-editorial-team> Research Cross-Journal Editorial Team
Nature Structural & Molecular Biology

Reviewer #1 (Remarks to the Author):

The reviewers have addressed my comments and the manuscript is much clearer and improved in its revised form.

Reviewer #3 (Remarks to the Author):

The authors have made a carefull and comprehensive revision answering all queries. I recommend acceptance.

Version 3:

Decision Letter:

Dear Dr. Savitski,

Thank you for your patience, we are now happy to accept your revised paper "Uncovering protein glycosylation dynamics and heterogeneity via Deep Quantitative Glycoprofiling (DQGlyco)" for publication as a Technical Report in Nature Structural & Molecular Biology.

Your paper will be published online soon after we receive proof corrections and will appear in print in the next available issue. You can find out your date of online publication by contacting the production team shortly after sending your proof corrections.

An online order form for reprints of your paper is available at <https://www.nature.com/reprints/author->

reprints.html"><https://www.nature.com/reprints/author-reprints.html>. Please let your coauthors and your institutions' public affairs office know that they are also welcome to order reprints by this method.

Best regards,

George

George Inglis, PhD

Senior Editor

[Research Cross-Journal Editorial Team](https://www.nature.com/nsmb/research-cross-journal-editorial-team)
Nature Structural & Molecular Biology

Reviewer #1:

Remarks to the Author:

This manuscript describes the development and application of Deep Quantitative Glycoprofiling (DQGlyco), a high-throughput sample preparation method to enrich glycopeptides combined with mass spectrometry analysis. DQGlyco is applied to the mouse brain tissue where an impressive number of unique glycopeptides are identified. Analysis is performed on this dataset to determine correlation between glycosylation microheterogeneity and protein structure. Tissue-specificity of glycosylation is also examined. Lastly mouse brains with different gut microbiota conditions are analyzed and claims are made that different microbiome conditions result in extensive remodeling of the brain glycoproteome.

Overall this is an interesting study with an impressive depth of analysis. The data set is also likely to be useful to the scientific community. However, my major criticism of the manuscript is that it covers several large areas (method development, protein structure and glycosylation, glycosylation differences between tissues, microbiome effect on mouse brain tissue), but is lacking substantial depth in any given area. The manuscript is largely descriptive and it is somewhat difficult to follow what the main point is. Some additional specific comments:

1) The first two paragraphs of the results section are out of place and don't belong in the Results section.

We thank the reviewer for this comment and moved the two paragraphs in the introduction section.

Action taken: Moved the two paragraphs to the introduction section.

2) The description of the AI analysis for protein structure and glycosylation not well described. It was difficult to follow the analysis and findings.

We thank the reviewer for this comment and described the AI analysis in more detail, both in the main text and in the method section.

Action taken: We expanded the results paragraph as well as the methods section to provide a more detailed explanation of the components and variables used in this analysis.

3) The claim that gut microbiome changes glycosylation in brain proteins is a substantial stretch.

While we agree that the mechanisms underlying the connection between gut microbiome compositions and glycoproteome alterations remain to be investigated, we think that given our experimental design (6 biological replicates in each group: germ-free, monocolonized, and 8-members colonized) the data shows that a clear remodeling of the brain glycoproteome occurs upon gut colonization.

4) Cell line authentication was not described.

The HeLaK cell line used in this study has been authenticated before (doi 10.1534/g3.113.005777), while the HEK293T was obtained from ATCC and the Expi293 cells were purchased from Thermo.

Action taken: We added this information in the methods section.

5) Scores for glycopeptide ID's in supplementary tables should be provided.

We thank the reviewer for this comment and added the MSFragger hyperscore of glycopeptides in the supplementary tables.

Action taken: MSFragger hyperscores were added to the supplementary tables.

6) The classification of glycopeptides is not clear. Is something double counted if it is both sialylated and fucosylated for example?

We thank the reviewer for this comment and apologize if the classification description was not clear. We used a hierarchical classification to group glycoforms in different glycan classes, as follow (from the method section): (i) phospho, containing a phosphate group, (ii) sialylated, containing a NeuAc or NeuGc sugar, (iii) fucosylated, containing fucose, (iv) high mannose, combining a HexNAc(2) core with between four and 12 hexose molecules, (v) paucimannose, combining a HexNAc(2) core with between one and three hexose molecules, (vi) small, composed of only a core with less than three HexNAc molecules and (vii) complex/hybrid, the rest. For O-glycosylation data, categories were defined as follows (hierarchical classification): (i) O-phospho, containing a phosphate group, (ii) O-sialylated, containing a NeuAc or NeuGc sugar, (iii) O-fucosylated, containing fucose, (iv) O-sulfated, containing a sulfation, (v) O-GlcNAc, for glycans containing a single HexNAc, (vi) O-hexose, composed of a core with less than three HexNAc molecules and at least one hexose molecule (vii) O-hybrid, the rest.

This means that a glycoform being both sialylated and fucosylated will be part of the sialylated glycan class, but not counted as a fucosylated glycoform.

Action taken: The last sentence above was added to the method section.

7) The number of biological and technical replicates for each experiment is not clear.

We thank the reviewer for this comment. We added the number of biological and technical replicates in the method section or figures legends.

Action taken: The number of replicates was added in the method section and/or in figures legends.

8) While substantially more glycopeptides are identified in this study compared to previous studies, glycopeptides in the mouse brain have been studied quite extensively, thereby limiting the novelty and making this study somewhat incremental.

We agree that the mouse brain glycoproteome has been profiled before, and one of the reasons why we selected this tissue was to benchmark our approach. By doing so, we were able to clearly demonstrate that our approach improved coverage by 25-fold. We would also like to emphasize that all previous studies in the mouse brain have been qualitative, and that our strategy enables for the first time the in-depth quantitative profiling of the brain glycoproteome dynamics in response to a controlled perturbation in vivo. Furthermore, we did not only profile the brain glycoproteome, but also other tissues (mouse liver and kidney) and cell lines (HeLa, HEK293T, HEK293f).

Action taken: Following the request of the editor, we also added additional data. We first profiled the modulation of the glycoproteome upon treatment with a fucosylation inhibitor (2FF), revealing site- and glycoform-specific regulation of glycosylation. Then, we introduced the concept of biophysical glycoproteomics, an approach enabling the determination of the solubility (here the differential solubility of glycoforms in a non-denaturing detergent - NP40 - vs a strong denaturing detergent - SDS) of different glycoforms in vivo. Such an approach was recently developed in our lab to quantify the impact of phosphorylation on protein solubility and decipher phosphorylation-mediated mechanisms of protein phase separation (DOI 10.1038/s41589-022-01062-y).

As a change of solubility can be linked to a change in protein state, this constitutes the first effort to systematically link differences of microheterogeneity with alterations in protein states (i.e. activity, localization or interactions). While SPP does not allow to obtain a definitive answer regarding the specific function of a given glycoform, it can

help to pinpoint functionally important glycosylation events on a proteome-wide scale and inform us on the potential role of this glycoform on protein function (e.g. if a glycoform promotes or prevents interactions with ECM, this could be reflected in an increase or decrease of solubility of this glycoform, respectively). Overall, this method revealed extensive differences of solubility between glycoforms, supporting at a proteome-wide scale previous anecdotal evidence that microheterogeneity likely influences protein function.

We copy here the new paragraphs from the main text, along with the corresponding figures:

“Time course of fucosylation inhibition reveals site- and glycoform-specific regulation of glycosylation

We first used our quantitative workflow to study the inhibition of fucosylation over time following treatment with 2-fluorofucose (2FF, Figure 4A). 2FF readily diffuses into cells, where it is metabolized into the donor substrate GDP-2FF that blocks fucosyltransferase activity. Moreover, the intracellular accumulation of GDP-2FF has been shown to inhibit the production of endogenous GDP-fucose leading to a general suppression of protein fucosylation, which inhibits several biological processes such as tumor growth, adhesion and metastasis. The profiling of the glycoproteome over three days of treatment of HEK293f cells revealed a gradual decrease of fucosylation, alongside extensive remodeling across all glycan classes (Figure 4B-D). Notably, the downregulation of fucosylation occurred at varying rates, which we grouped into 3 main clusters (Figure 4E), revealing a widespread site-specific and glycoform-specific modulation. This is illustrated by fucosylated glycoforms of the same protein belonging to different downregulated clusters (Figure 4F), meaning that such differences in kinetics of fucosylation downregulation cannot be attributed to differences in protein turnover rates. A clear example of this observation is the modulation of glycoforms on the proteins MRC2 and NRCAM, where fucosylated glycoforms on different sites follow distinct kinetics (Figure 4G,H).

It is generally accepted that changes in protein glycosylation occur mainly via modulation of glycoenzymes abundance, localization or activity. In such cases, similar trends would be expected at the glycoproteome level, especially for glycosites within the same protein. This site- and glycoform-specific modulation could be explained by glycoform-specific turnover rate (i.e. glycoforms are subjected to endocytosis and recycled at different rates), or by more complex modulation at the fucosyltransferase level.”

“Biophysical glycoproteomics

Modulation of site microheterogeneity has been shown to regulate many cellular processes, but determining its functional impact remains challenging. This is mainly due

to the fact that only few glycosylation events have a functional annotation, that it is currently impossible to selectively alter a given glycoform on a given glycosite both in vitro and in vivo, and that one cannot perform mutagenesis to assess functionality of glycosite microheterogeneity. The quantitative glycoproteomics workflow developed in this study enables us to systematically determine biophysical properties of glycoforms in vivo by assessing the solubility of more than 30,000 glycoforms using Solubility Proteome Profiling (SPP, (Sridharan et al., 2019), Supplementary Data 5). Here, the differential solubility of a given proteoform in a mild detergent - NP40 - and a strong denaturing detergent - SDS - is determined (Figure XA), an approach previously used to systematically study the functional relevance of protein phosphorylation (Sridharan et al., 2022). Although this approach does not directly reveal the impact of a change in glycosylation on specific protein function or activity, variations in protein solubility due to a specific glycosylation likely suggest differences in protein state, such as localization or interaction.

Application of glyco-SPP showed that high mannose glycoforms tend to be more soluble than more processed glycans (Figure 6B), likely due to the fact that mature glycoforms are mostly less soluble than intracellular glycoforms, as a result of interactions with the extracellular matrix. This is particularly well illustrated in the case of the cadherin 13 protein, (CDH13), involved in cell adhesion: glycoforms on the site N86, belonging to the propeptide, are more soluble than glycoforms on other sites belonging to the mature proteoforms (Figure XC). In general, distinct glycoforms on the same site exhibit varied solubility, suggesting that difference in microheterogeneity can be linked to changes in protein activity, localization or interactions. Notably, we observed several instances of site-specific impact of glycosylation on solubility, wherein most glycosylation events on a given site tend to influence protein solubility in the same direction, as illustrated for the site N94 of Thy-1 or the sites N73 and N80 of CD47 (Figure XD,E).”

Reviewer #2:

Remarks to the Author:

This manuscript by Potel et al. describes a new workflow for glycoproteomics, combining an improved method for glycopeptide enrichment with optimized deep fractionation and mass spectrometric acquisition methods to profile an order of magnitude more glycopeptides than previous studies in similar sample types. The authors then use this exceptionally deep glycoproteomic profiling to provide a variety of biological insights, providing an excellent illustration of the potential of glycoproteomics. The manuscript is compelling and clearly supports these rather exceptional claims with clear and convincing evidence throughout. Other than a few minor clarifications and areas for additional discussion, detailed below, I see no barriers to publication.

Specific comments:

1. The improvement in glycopeptides identified from the PBA enrichment and optimized MS1 scan range, while impressive, appear to be smaller than the gain from the PGC fractionation in Fig 1. This is particularly notable given that the previous studies being compared (Riley et al, etc.) also used deep fractionation, but via high pH C18 rather than PGC. Some additional discussion of the interplay of these factors would be very valuable to clarify whether similarly deep profiling could be accomplished by PGC fractionation with traditional enrichment methods (HILIC, lectins) and/or without the nucleic acid precipitation, or if it is really the combination of all of these improvements together that is necessary to achieve the order of magnitude increase in glycopeptides identified. Additionally, some discussion of any challenges or key considerations for implementing this method in other labs would provide a valuable perspective on the potential of the method.

We thank the reviewer for the positive feedback. We agree that the off-line fractionation of glycopeptides using a PGC column provides the most important gain in identification. We are confident that a PGC fractionation of a glycopeptide sample following a HILIC or lectin-based enrichment would also boost the number of identification significantly, albeit to a lower extent than with the PBA enrichment. We show in Figure 1C that PBA-based enrichment achieves high specificity thanks to stringent washes following covalent capture by PBA functionalized beads (ca 90% of peptides are glycopeptides), which is not possible with traditional enrichment methods. This is illustrated in the figure S1F, where we show that a single shot measurement of a non fractionated mouse brain glycopeptide sample following a PBA enrichment provides more than double the identification numbers when compared to off-line fractionated samples obtained after HILIC (Liu et al.) and lectin-based (Riley et al.) enrichments. In addition, it appears that the PBA based enrichment method is less biased than lectin-based or HILIC enrichment techniques towards certain glycan composition or molecular weight. Indeed, the different lectins are biased towards certain types of glycan composition (ConA is for example exhibiting a bias towards high mannose glycans), and HILIC enrichment will favor the enrichment of highly hydrophilic glycans (negatively charged and high molecular weight glycans).

Overall, regarding the interplay between the different optimized parameters, we believe that the improvement of the MS acquisition methods, as well as the PGC fractionation would enhance all glycoproteomics enrichment platforms. However, the nucleic acid precipitation should mostly be beneficial for sample preparation prior to PBA enrichment. Indeed, PBA will form a covalent bond with diols present in nucleic acids, but lectin, and to a lesser extent HILIC enrichment should remain largely unaffected by the presence of nucleic acids in the sample.

Action taken: We added the previous paragraph in the discussion.

2. The discussion of peptides bearing O-GlcNAc modifications should clarify explicitly that a single HexNAc can be either O-GlcNAc or O-GalNAc, and while the oxonium ion ratios in Fig S2G clearly show a difference in trend between intra- and extracellular proteins, it is not as clear-cut as described in the text given that median ratio for extracellular proteins remains well below 1. Calling these putative O-GlcNAc modifications would be more accurate, and the limitations of oxonium ion analysis (namely, co-fragmentation) should be mentioned.

We thank the reviewer for this comment and modified the nomenclature as suggested.

Action taken: The term O-GlcNAc was replaced by O-HexNAc in the text and figures.

3. The choice to use a more liberal glycan q-value threshold of 0.05 is not unreasonable, but the text should note that the confidence in the glycan compositions identified is lower than the peptides, given the higher chosen threshold. A comparison of the results for one analysis between setting a threshold of 0.01 and 0.05 could be a useful addition to the supporting information to inform the reader as to the impact of this choice.

We thank the reviewer for this comment. We discussed extensively with the developers of MSFragger glyco regarding the q-value cutoff, and agreed to choose 0.05, a cutoff previously used in a recent publication from the same lab (Bedran et al., <https://doi.org/10.1038/s41467-023-39270-2>). According to the developers, the glycan q-values serves more as a guideline than a hard cutoff, and is particularly useful to filter out glycan compositions with a q-value = 1, which are likely to be misassigned. This is underlined by the skewed distribution of the glycan q-values, which differs significantly from typical peptides q-values distribution observed in proteomics. This is illustrated by the q-value distribution plotted below (from a fractionated mouse brain sample, single high energy HCD fragmentation), in which more than 230,000 PSMs have the exact same glycan q-value of 0.0234599165. We think that the assessment of glycan composition assignment confidence is an important issue, and hope that the extensive datasets generated (more than 1.5 millions glycoPSMs, from label-free and TMT-labeled samples, fragmented with single high energy or stepped HCD collision energies) will aid in addressing this challenge and developing appropriate scoring models.

Action taken: We added in the first result paragraph of the main text that the different thresholds used for N-glycopeptides identification are: “1% FDR at the N-glycopeptide level, glycan q-values ≤ 0.05 ”.

Reviewer #3:

Remarks to the Author:

Potel et al. present a study focused on deep quantitative glycoproteomics including a number of technical advances with an application related to microbiome induced changes in mouse brain. The technical advances include a glycopeptide enrichment based on existing covalent chemistry (PBA) combined with highthroughput plate-based sample processing methods that reduces contaminating species such as nucleic acids. While the enrichment methodology already contains several improvements the authors then employ a multidimensional approach using porous graphitic carbon which substantially improves glycopeptide coverage. The authors benchmark by comparison to other recent papers demonstrating the improvements, and importantly also show that their method demonstrates less bias for specific glycopeptide/protein classes than existing methods. They assess site-level heterogeneity and protein level heterogeneity as well as setting glycoform variation in the context of predicted structural features including a machine learning model to predict site level heterogeneity based on structural features. Finally, the authors integrate isobaric labelling with their multidimensional method, including labelling adaptations to minimize label requirements, to enable quantitative comparisons. The strategy is then applied to microbiome perturbation experiment in mouse. They show first that there strong distinction in tissue level specificity of the glycoproteome, and further differences in specific glycoforms in brains of mice depending on microbiome status.

In my view this manuscript is very strong and represents a very substantial improvement over the state of the art in glycoproteomics methodology. The increase in depth achieved, and removal of systematic bias for glycopeptide coverage, by very careful and systematic optimization of purification, sample prep, and MS methodology is fairly remarkable. I think the manuscript is very logical and well written, well illustrated, and packs in a large body of work that clearly shows the progression of concepts. The conclusions are surrounding micro- and macro-heterogeneity (also related to structure), tissue specificity, and effects of microbiome on brain are novel and striking. All data and code are made available in appropriate repositories, and results are explorable via a web app. My view is that this will have an immediate and strong impact on the field. As such, I think there is rather little to complain about and I suggest that the manuscript should be accepted in short order. I have put a few minor clarifying comments/questions below.

1. One technical question would be whether anything semi-quantitative can be said about the distribution of glycoforms at the site level. Given that the most extreme case they observe 600+ glycoforms at a single site, can anything be said about whether any forms are more common (akin to major/minor isoform is splice variants). While the

response factor for a given peptide sequence dependent on the glycoform it harbours will presumably change, but perhaps a comparison of intensities would be informative. Is anything known about this issue? Could this be summarized at glycoform class level?

We thank the reviewer for the positive feedback. We agree that information about stoichiometry is important to understand biological implications of observed glycoform alterations. However, we can here only perform relative quantification, and using MS1 or summed TMT intensities to compare glycoform abundance would only be valid if the glycan composition does not impact ionization efficiency. Based on the general MS1 intensities of the different glycan classes, one can for example clearly see that the presence of sialylated glycan negatively affects ionization efficiency in positive mode, likely due to presence of negative charges (MS1 glycopeptide intensity per glycan class compared to high mannose glycopeptide intensities):

Nevertheless, using MS1 intensities to calculate glycoform fractional intensity (ratio of glycoform intensity over the sum of all glycoforms intensities on a given site) revealed that most glycoforms have a low fractional intensity:

When focusing on the glycoform having the highest fractional intensity per site (here only the sites with at least 10 glycoforms were considered), it becomes clear that glycosylation of a specific site is rarely dominated by a glycoform of high stoichiometry:

Action taken: We added the discussion above to the main text, in the “Deep mouse brain glycoproteome analysis” result section (figures in Figure S2).

2. The description of the methodological advances in the first results section could be slightly more easy to digest if revised elaborating further all steps. While its fine to leave majority of details in the methods section, I think it would be more understandable if conceptual aspects were more completely stated (i.e. a key advance is the removal of nucleic acids by precipitation but the details of how this is done is breezed over, enrichment by covalent coupling to PBA is described but not method for elution, etc). My experience on reading was to have to dig around in methods several times to understand the complete workflow conceptually so a revision of this section for conceptual clarity might help the less initiated reader.

We thank the reviewer for this suggestion, and added additional information in the main text regarding the nucleic acid precipitation and glycopeptide elution parts.

Action taken: The main text was modified as follow:

“To address this, we optimized the sample lysis buffer by incorporating high concentration of chaotropic salts and organic solvent to induce nucleic acid precipitation while proteins remain in solution. Nucleic acid aggregates were then filtered out on 96-well filter plates, followed by protein precipitation by further increasing organic solvent concentration prior to tryptic enzymatic digestion.”

“Chemical coupling of glycopeptides to beads functionalized with reactive groups generally leverages the reversible reaction between phenyl-boronic acid (PBA) derivatives and 1,2- or 1,3-diols present in sugar molecules. This results in the formation of a covalent bond between functionalized beads and glycopeptides at high pH, followed by elution of the glycopeptides at low pH.”

3. I am missing a bit some info on how much input material is needed for both their single shot or multidimensional strategies (i.e. how much protein from how many cells and how much tissue is needed/optimal). This would be an important consideration for anyone wanting to try this.

We thank the reviewer for his comment. In our hands, the number of identification reaches a plateau at 1mg of input for a single shot analysis, but of course this will depend on the sample type (cell-line, tissue, organism,...) and the instrumentation (i.e. an Astral will reach the same number of identification than an Orbitrap Exploris or tribrid with less input). Regarding the fractionated samples, in this study we enriched 2mg of peptides per replicate before TMT labeling and pooling (the total input before fractionation being then equal to 36 mg for the tissue and microbiome experiment, or 16 mg of input for the SPP). For the label free samples fractionated with PGC, we enriched 30 mg of mouse brain peptides.

Action taken: We added the inputs used in the method section, as followed: “For single-shot measurements of glycopeptides, we found that the number of identification reached a plateau after 1 mg of input. For the PGC fractionated sample, we enriched 2 mg of peptide per TMT channel, or 30 mg of peptides in the case of label-free PGC experiments.”

4. In methods they mention that identified O-linked peptides containing a N-linked motif were removed. It would be interesting to know the rate at which rate this occurs.

We thank the reviewer for his comment. We found that it is a critical step, as the search engine will search separately for either N- or O-linked glycopeptides.

As in the samples analyzed the majority of glycopeptides are N-glycopeptides, and given the fact that several N- and O-glycan compositions have the same molecular weight, by searching for O-glycopeptides, MSFragger will identify many candidates O-glycopeptides based on the peptide backbone fragments and offset mass which are in reality N-glycopeptides (all containing a serine or threonine in the sequence), with the same peptide sequence having a N-glycan composition of similar mass.

By removing all O-glycopeptides having a N-X-S/T sequon in the peptide sequence (or N-K/R at the peptide C-term followed by S/T in the protein sequence), we ensure that this is not the case. We chose this conservative strategy to minimize the number of misidentifications. By doing so, 85% of the O-glycopeptides identified by MSFragger were filtered out.

Reviewer #1:

Remarks to the Author:

This manuscript describes the development and application of Deep Quantitative Glycoprofiling (DQGlyco), a high-throughput sample preparation method to enrich glycopeptides combined with mass spectrometry analysis. DQGlyco is applied to the mouse brain tissue where an impressive number of unique glycopeptides are identified. Analysis is performed on this dataset to determine correlation between glycosylation microheterogeneity and protein structure. Tissue-specificity of glycosylation is also examined. Lastly mouse brains with different gut microbiota conditions are analyzed and claims are made that different microbiome conditions result in extensive remodeling of the brain glycoproteome.

Overall this is an interesting study with an impressive depth of analysis. The data set is also likely to be useful to the scientific community. However, my major criticism of the manuscript is that it covers several large areas (method development, protein structure and glycosylation, glycosylation differences between tissues, microbiome effect on mouse brain tissue), but is lacking substantial depth in any given area. The manuscript is largely descriptive and it is somewhat difficult to follow what the main point is. Some additional specific comments:

1) The first two paragraphs of the results section are out of place and don't belong in the Results section.

We thank the reviewer for this comment and moved the two paragraphs in the introduction section.

Action taken: Moved the two paragraphs to the introduction section.

2) The description of the AI analysis for protein structure and glycosylation not well described. It was difficult to follow the analysis and findings.

We thank the reviewer for this comment and described the AI analysis in more detail, both in the main text and in the method section.

Action taken: We expanded the results paragraph as well as the methods section to provide a more detailed explanation of the components and variables used in this analysis.

3) The claim that gut microbiome changes glycosylation in brain proteins is a substantial stretch.

While we agree that the mechanisms underlying the connection between gut microbiome compositions and glycoproteome alterations remain to be investigated, we think that given our experimental design (6 biological replicates in each group: germ-free, monocolonized, and 8-members colonized) the data shows that a clear remodeling of the brain glycoproteome occurs upon gut colonization. Nevertheless, following editorial discussion, we changed the title and focus of the manuscript to reflect the technical nature of our work.

Action taken: We changed the title of the manuscript, and added new data, as detailed in the response to point 7 of the reviewer.

4) Cell line authentication was not described.

The HeLaK cell line used in this study has been authenticated before (doi 10.1534/g3.113.005777), while the HEK293T was obtained from ATCC and the Expi293 cells were purchased from Thermo.

Action taken: We added this information in the methods section.

5) Scores for glycopeptide ID's in supplementary tables should be provided.

We thank the reviewer for this comment and added the MSFragger hyperscore of glycopeptides in the supplementary tables.

Action taken: MSFragger hyperscores were added to the supplementary tables.

6) The classification of glycopeptides is not clear. Is something double counted if it is both sialylated and fucosylated for example?

We thank the reviewer for this comment and apologize if the classification description was not clear. We used a hierarchical classification to group glycoforms in different glycan classes, as follow (from the method section): (i) phospho, containing a phosphate group, (ii) sialylated, containing a NeuAc or NeuGc sugar, (iii) fucosylated, containing fucose, (iv) high mannose, combining a HexNAc(2) core with between four and 12 hexose molecules, (v) paucimannose, combining a HexNAc(2) core with between one and three hexose molecules, (vi) small, composed of only a core with less

than three HexNAc molecules and (vii) complex/hybrid, the rest. For O-glycosylation data, categories were defined as follows (hierarchical classification): (i) O-phospho, containing a phosphate group, (ii) O-sialylated, containing a NeuAc or NeuGc sugar, (iii) O-fucosylated, containing fucose, (iv) O-sulfated, containing a sulfation, (v) O-GlcNAc, for glycans containing a single HexNAc, (vi) O-hexose, composed of a core with less than three HexNAc molecules and at least one hexose molecule (vii) O-hybrid, the rest. This means that a glycoform being both sialylated and fucosylated will be part of the sialylated glycan class, but not counted as a fucosylated glycoform.

Action taken: The last sentence above was added to the method section.

7) The number of biological and technical replicates for each experiment is not clear.

We thank the reviewer for this comment. We added the number of biological and technical replicates in the method section or figures legends.

Action taken: The number of replicates was added in the method section and/or in figures legends.

8) While substantially more glycopeptides are identified in this study compared to previous studies, glycopeptides in the mouse brain have been studied quite extensively, thereby limiting the novelty and making this study somewhat incremental.

We agree that the mouse brain glycoproteome has been profiled before, and one of the reasons why we selected this tissue was to benchmark our approach. By doing so, we were able to clearly demonstrate that our approach improved coverage by 25-fold. We would also like to emphasize that all previous studies in the mouse brain have been qualitative, and that our strategy enables for the first time the in-depth quantitative profiling of the brain glycoproteome dynamics in response to a controlled perturbation in vivo. Furthermore, we did not only profile the brain glycoproteome, but also other tissues (mouse liver and kidney) and cell lines (HeLa, HEK293T, HEK293f).

Action taken: Following the request of the editor, we also added additional data. We first profiled the modulation of the glycoproteome upon treatment with a fucosylation inhibitor (2FF), revealing site- and glycoform-specific regulation of glycosylation. In addition, we treated intact living HEK293f cells with either PNGase F or proteinase K to characterize the mature, surface exposed glycoforms. Finally, we introduced the concept of biophysical glycoproteomics, an approach enabling the determination of the solubility (here the differential solubility of glycoforms in a non-denaturing detergent - NP40 - vs a strong denaturing detergent - SDS) of different glycoforms in vivo. Such an

approach was recently developed in our lab to quantify the impact of phosphorylation on protein solubility and decipher phosphorylation-mediated mechanisms of protein phase separation (DOI 10.1038/s41589-022-01062-y).

As a change of solubility can be linked to a change in protein state, this constitutes the first effort to systematically link differences of microheterogeneity with alterations in protein states (i.e. activity, localization or interactions). While SPP does not allow to obtain a definitive answer regarding the specific function of a given glycoform, it can help to pinpoint functionally important glycosylation events on a proteome-wide scale and inform us on the potential role of this glycoform on protein function (e.g. if a glycoform promotes or prevents interactions with ECM, this could be reflected in an increase or decrease of solubility of this glycoform, respectively). Overall, this method revealed extensive differences of solubility between glycoforms, supporting at a proteome-wide scale previous anecdotal evidence that microheterogeneity likely influences protein function.

We copy here the new paragraphs from the main text, along with the corresponding main figures:

“Time course of fucosylation inhibition reveals site- and glycoform-specific regulation of glycosylation

We first used our quantitative workflow to study the inhibition of fucosylation over time following treatment with 2-fluorofucose (2FF, Figure 4A). 2FF readily diffuses into cells, where it is metabolized into the donor substrate GDP-2FF that blocks fucosyltransferase activity. Moreover, the intracellular accumulation of GDP-2FF has been shown to inhibit the production of endogenous GDP-fucose leading to a general suppression of protein fucosylation, which inhibits several biological processes such as tumor growth, adhesion and metastasis. The profiling of the glycoproteome over three days of treatment of HEK293f cells revealed a gradual decrease of fucosylation, alongside extensive remodeling across all glycan classes (Figure 4B-D). Notably, the downregulation of fucosylation occurred at varying rates, which we grouped into 3 main clusters (Figure 4E), revealing a widespread site-specific and glycoform-specific modulation. This is illustrated by fucosylated glycoforms of the same protein belonging to different downregulated clusters (Figure 4F), meaning that such differences in kinetics of fucosylation downregulation cannot be attributed to differences in protein turnover rates. A clear example of this observation is the modulation of glycoforms on the proteins MRC2 and NRCAM, where fucosylated glycoforms on different sites follow distinct kinetics (Figure 4G,H).

It is generally accepted that changes in protein glycosylation occur mainly via modulation of glycoenzymes abundance, localization or activity. In such cases, similar trends would be expected at the glycoproteome level, especially for glycosites within the same protein. This site- and glycoform-specific modulation could be explained by glycoform-specific turnover rate (i.e. glycoforms are subjected to endocytosis and recycled at different rates), or by more complex modulation at the fucosyltransferase level.”

Determination of cell surface-exposed glycoforms

N-glycans are added to asparagine side chains of proteins as high-mannose in the ER before being further processed in the Golgi. Once the cells are lysed, it becomes impossible to differentiate mature, functional glycoforms (i.e. present on proteins which made it to their final destination) from immature glycoforms (i.e. intermediate species on

proteins going through the secretory pathway). Due to the complexity of the glycosylation machinery, it is not possible to predict the composition of mature glycans and it is postulated that glycoforms of any compositions or structures could potentially be surface-exposed (Schjoldager et al., 2020). Significant differences have been observed between cell surface and intracellular glycans (de Haan et al., 2023), but obtaining a comprehensive system-wide, site-specific overview of mature glycoforms has been challenging so far and the compositions or the proportion of mature glycoforms in a given glycoproteomics experiment remains an open question in the field. Here, we applied our quantitative workflow to intact living human HEK293 cells subjected to either PNGase F treatment for 30 mins (glycosidase targeting N-glycans) or proteinase K for 1 min (broad specificity enzyme used in limited proteolysis experiments (Schopper et al., 2017)). Our rationale was that only cell surface-exposed glycoforms would be affected, while glycoforms present in intracellular organelles (i.e. in the ER or Golgi) would be protected, resulting in unchanged abundance (Figure 5A). As a quality control, we demonstrated that the vast majority of proteins affected by the enzymes were annotated as belonging to the plasma membrane (Figure 5B).

While some sites remained inaccessible and were not affected by the enzymatic treatments, we identified 3,990 glycopeptides on 826 sites and 513 proteins showing significant changes in abundance upon treatment (Supplementary Data 4, Figure S7A,B). To enhance confidence we decided to consider for subsequent analysis only sites for which at least 2 glycoforms were affected (3,113 glycopeptides on 299 sites and 168 proteins, Figure S6B). For example, on the zinc transporter ZIP6, the N241 site was consistently affected by both enzymatic treatment, while the N67 and N266 sites remained unaltered (Figure 5C). Regarding the site N67, this can be explained by the fact that this site is located on a propeptide, the N-terminal cleavage triggering the location of ZIP6 to the plasma membrane (Hogstrand et al., 2013). Similarly, the site N34 of Desmocollin-2 (DSC2) remained affected, because localized on the propeptide region whereas glycans on the N392, N546 and N629 sites were revealed to be surface-exposed (Figure 5D).

Analyzing these accessible sites, we could demonstrate that high-mannose glycoforms are in general less surface exposed, although this tendency seems to be site-dependent. Conversely, most of the fucosylated/sialylated glycoforms are, at least partially, surface-exposed, aligning with recent literature findings (de Haan et al., 2023) (Figure 5E). Sites exhibiting at least two high mannose surface-exposed glycoforms were in general less accessible, supporting the notion that site accessibility may influence glycan maturation (Figure S7C). We could in addition identify glycan compositions predominantly characterized as surface-exposed in our dataset (Figure S7D). For both enzymes, no discernible bias towards preferential cleavage of given

glycan compositions or molecular weight was observed (Figure S7E/F). This suggests that the measured fold change upon treatment likely reflects the degree of surface exposition of a given glycoform.

Notably, in instances where sites were impacted by both enzymatic treatments, we observed good correlation between the two experiments, both at the glycopeptide level (fold change of a given glycopeptide between treatment and control conditions) and glycan compositions level (number of times a given glycan composition was observed on a glycopeptide significantly changing upon treatment) (Figure 5F,G). PNGase and proteinase K were also found to be complementary (Figure S7G, Figure S7H), with proteinase K effectively targeting less accessible glycosites (Figure S7I), likely due to the fact that cleavage at any amino acids in the vicinity of the glycosite will result in a change in glycopeptide abundance while PNGase needs access to the glycosylated asparagine. In summary, we introduce a method for determining surface-exposed glycans, and provide a system-wide, site-specific view of mature and immature glycans in a human cell line, enhancing our understanding of the biological functions of glycosylation.

Biophysical glycoproteomics

Modulation of site microheterogeneity has been shown to regulate many cellular processes, but determining its functional impact remains challenging. This is mainly due to the fact that only few glycosylation events have a functional annotation, that it is currently impossible to selectively alter a given glycoform on a given glycosite both in vitro and in vivo, and that one cannot perform mutagenesis to assess functionality of glycosite microheterogeneity. The quantitative glycoproteomics workflow developed in this study enables us to systematically determine biophysical properties of glycoforms in vivo by assessing the solubility of more than 30,000 glycoforms using Solubility Proteome Profiling (SPP, (Sridharan et al., 2019), Supplementary Data 5). Here, the differential solubility of a given proteoform in a mild detergent - NP40 - and a strong denaturing detergent - SDS - is determined (Figure XA), an approach previously used to

systematically study the functional relevance of protein phosphorylation (Sridharan et al., 2022). Although this approach does not directly reveal the impact of a change in glycosylation on specific protein function or activity, variations in protein solubility due to a specific glycosylation likely suggest differences in protein state, such as localization or interaction.

Application of glyco-SPP showed that high mannose glycoforms tend to be more soluble than more processed glycans (Figure 6B), likely due to the fact that mature glycoforms are mostly less soluble than intracellular glycoforms, as a result of interactions with the extracellular matrix. This is particularly well illustrated in the case of the cadherin 13 protein, (CDH13), involved in cell adhesion: glycoforms on the site N86, belonging to the propeptide, are more soluble than glycoforms on other sites belonging to the mature proteoforms (Figure XC). In general, distinct glycoforms on the same site exhibit varied solubility, suggesting that difference in microheterogeneity can be linked to changes in protein activity, localization or interactions. Notably, we observed several instances of site-specific impact of glycosylation on solubility, wherein most glycosylation events on a given site tend to influence protein solubility in the same direction, as illustrated for the site N94 of Thy-1 or the sites N73 and N80 of CD47 (Figure XD,E).”

Reviewer #2:

Remarks to the Author:

This manuscript by Potel et al. describes a new workflow for glycoproteomics, combining an improved method for glycopeptide enrichment with optimized deep fractionation and mass spectrometric acquisition methods to profile an order of

magnitude more glycopeptides than previous studies in similar sample types. The authors then use this exceptionally deep glycoproteomic profiling to provide a variety of biological insights, providing an excellent illustration of the potential of glycoproteomics. The manuscript is compelling and clearly supports these rather exceptional claims with clear and convincing evidence throughout. Other than a few minor clarifications and areas for additional discussion, detailed below, I see no barriers to publication.

Specific comments:

1. The improvement in glycopeptides identified from the PBA enrichment and optimized MS1 scan range, while impressive, appear to be smaller than the gain from the PGC fractionation in Fig 1. This is particularly notable given that the previous studies being compared (Riley et al, etc.) also used deep fractionation, but via high pH C18 rather than PGC. Some additional discussion of the interplay of these factors would be very valuable to clarify whether similarly deep profiling could be accomplished by PGC fractionation with traditional enrichment methods (HILIC, lectins) and/or without the nucleic acid precipitation, or if it is really the combination of all of these improvements together that is necessary to achieve the order of magnitude increase in glycopeptides identified. Additionally, some discussion of any challenges or key considerations for implementing this method in other labs would provide a valuable perspective on the potential of the method.

We thank the reviewer for the positive feedback. We agree that the off-line fractionation of glycopeptides using a PGC column provides the most important gain in identification. We are confident that a PGC fractionation of a glycopeptide sample following a HILIC or lectin-based enrichment would also boost the number of identification significantly, albeit to a lower extent than with the PBA enrichment. We show in Figure 1C that PBA-based enrichment achieves high specificity thanks to stringent washes following covalent capture by PBA functionalized beads (ca 90% of peptides are glycopeptides), which is not possible with traditional enrichment methods. This is illustrated in the figure S1F, where we show that a single shot measurement of a non fractionated mouse brain glycopeptide sample following a PBA enrichment provides more than double the identification numbers when compared to off-line fractionated samples obtained after HILIC (Liu et al.) and lectin-based (Riley et al.) enrichments. In addition, it appears that the PBA based enrichment method is less biased than lectin-based or HILIC enrichment techniques towards certain glycan composition or molecular weight. Indeed, the different lectins are biased towards certain types of glycan composition (ConA is for example exhibiting a bias towards high mannose glycans), and HILIC enrichment will favor the enrichment of highly hydrophilic glycans (negatively charged and high molecular weight glycans).

Overall, regarding the interplay between the different optimized parameters, we believe

that the improvement of the MS acquisition methods, as well as the PGC fractionation would enhance all glycoproteomics enrichment platforms. However, the nucleic acid precipitation should mostly be beneficial for sample preparation prior to PBA enrichment. Indeed, PBA will form a covalent bond with diols present in nucleic acids, but lectin, and to a lesser extent HILIC enrichment should remain largely unaffected by the presence of nucleic acids in the sample.

Action taken: We added the previous paragraph in the discussion.

2. The discussion of peptides bearing O-GlcNAc modifications should clarify explicitly that a single HexNAc can be either O-GlcNAc or O-GalNAc, and while the oxonium ion ratios in Fig S2G clearly show a difference in trend between intra- and extracellular proteins, it is not as clear-cut as described in the text given that median ratio for extracellular proteins remains well below 1. Calling these putative O-GlcNAc modifications would be more accurate, and the limitations of oxonium ion analysis (namely, co-fragmentation) should be mentioned.

We thank the reviewer for this comment and modified the nomenclature as suggested.

Action taken: The term O-GlcNAc was replaced by O-HexNAc in the text and figures.

3. The choice to use a more liberal glycan q-value threshold of 0.05 is not unreasonable, but the text should note that the confidence in the glycan compositions identified is lower than the peptides, given the higher chosen threshold. A comparison of the results for one analysis between setting a threshold of 0.01 and 0.05 could be a useful addition to the supporting information to inform the reader as to the impact of this choice.

We thank the reviewer for this comment. We discussed extensively with the developers of MSFragger glyco regarding the q-value cutoff, and agreed to choose 0.05, a cutoff previously used in a recent publication from the same lab (Bedran et al., <https://doi.org/10.1038/s41467-023-39270-2>). According to the developers, the glycan q-values serves more as a guideline than a hard cutoff, and is particularly useful to filter out glycan compositions with a q-value = 1, which are likely to be misassigned. This is underlined by the skewed distribution of the glycan q-values, which differs significantly from typical peptides q-values distribution observed in proteomics. This is illustrated by the q-value distribution plotted below (from a fractionated mouse brain sample, single high energy HCD fragmentation), in which more than 230,000 PSMs have the exact same glycan q-value of 0.0234599165. We think that the assessment of glycan composition assignment confidence is an important issue, and hope that the extensive datasets generated (more than 1.5 millions glycoPSMs, from label-free and

TMT-labeled samples, fragmented with single high energy or stepped HCD collision energies) will aid in addressing this challenge and developing appropriate scoring models.

Action taken: We added in the first result paragraph of the main text that the different thresholds used for N-glycopeptides identification are: “1% FDR at the N-glycopeptide level, glycan q-values ≤ 0.05 ”.

Reviewer #3:

Remarks to the Author:

Potel et al. present a study focused on deep quantitative glycoproteomics including a number of technical advances with an application related to microbiome induced changes in mouse brain. The technical advances include a glycopeptide enrichment based on existing covalent chemistry (PBA) combined with highthroughput plate-based sample processing methods that reduces contaminating species such as nucleic acids. While the enrichment methodology already contains several improvements the authors then employ a multidimensional approach using porous graphitic carbon which substantially improves glycopeptide coverage. The authors benchmark by comparison to other recent papers demonstrating the improvements, and importantly also show that their method demonstrates less bias for specific glycopeptide/protein classes than existing methods. They assess site-level heterogeneity and protein level heterogeneity as well as setting glycoform variation in the context of predicted structural features including a machine learning model to predict site level heterogeneity based on structural features. Finally, the authors integrate isobaric labelling with their multidimensional method, including labelling adaptations to minimize label requirements, to enable quantitative comparisons. The strategy is then applied to microbiome perturbation experiment in mouse. They show first that there strong distinction in tissue

level specificity of the glycoproteome, and further differences in specific glycoforms in brains of mice depending on microbiome status.

In my view this manuscript is very strong and represents a very substantial improvement over the state of the art in glycoproteomics methodology. The increase in depth achieved, and removal of systematic bias for glycopeptide coverage, by very careful and systematic optimization of purification, sample prep, and MS methodology is fairly remarkable. I think the manuscript is very logical and well written, well illustrated, and packs in a large body of work that clearly shows the progression of concepts. The conclusions are surrounding micro- and macro-heterogeneity (also related to structure), tissue specificity, and effects of microbiome on brain are novel and striking. All data and code are made available in appropriate repositories, and results are explorable via a web app. My view is that this will have an immediate and strong impact on the field. As such, I think there is rather little to complain about and I suggest that the manuscript should be accepted in short order. I have put a few minor clarifying comments/questions below.

1. One technical question would be whether anything semi-quantitative can be said about the distribution of glycoforms at the site level. Given that the most extreme case they observe 600+ glycoforms at a single site, can anything be said about whether any forms are more common (akin to major/minor isoform is splice variants). While the response factor for a given peptide sequence dependent on the glycoform it harbours will presumably change, but perhaps a comparison of intensities would be informative. Is anything known about this issue? Could this be summarized at glycoform class level?

We thank the reviewer for the positive feedback. We agree that information about stoichiometry is important to understand biological implications of observed glycoform alterations. However, we can here only perform relative quantification, and using MS1 or summed TMT intensities to compare glycoform abundance would only be valid if the glycan composition does not impact ionization efficiency. Based on the general MS1 intensities of the different glycan classes, one can for example clearly see that the presence of sialylated glycan negatively affects ionization efficiency in positive mode, likely due to presence of negative charges (MS1 glycopeptide intensity per glycan class compared to high mannose glycopeptide intensities):

Nevertheless, using MS1 intensities to calculate glycoform fractional intensity (ratio of glycoform intensity over the sum of all glycoforms intensities on a given site) revealed that most glycoforms have a low fractional intensity:

When focusing on the glycoform having the highest fractional intensity per site (here only the sites with at least 10 glycoforms were considered), it becomes clear that glycosylation of a specific site is rarely dominated by a glycoform of high stoichiometry:

Action taken: We added the discussion above to the main text, in the “Deep mouse brain glycoproteome analysis” result section (figures in Figure S2).

2. The description of the methodological advances in the first results section could be slightly more easy to digest if revised elaborating further all steps. While its fine to leave majority of details in the methods section, I think it would be more understandable if conceptual aspects were more completely stated (i.e. a key advance is the removal of nucleic acids by precipitation but the details of how this is done is breezed over, enrichment by covalent coupling to PBA is described but not method for elution, etc). My experience on reading was to have to dig around in methods several times to understand the complete workflow conceptually so a revision of this section for conceptual clarity might help the less initiated reader.

We thank the reviewer for this suggestion, and added additional information in the main text regarding the nucleic acid precipitation and glycopeptide elution parts.

Action taken: The main text was modified as follow:

“To address this, we optimized the sample lysis buffer by incorporating high concentration of chaotropic salts and organic solvent to induce nucleic acid precipitation while proteins remain in solution. Nucleic acid aggregates were then filtered out on

96-well filter plates, followed by protein precipitation by further increasing organic solvent concentration prior to tryptic enzymatic digestion.”

“Chemical coupling of glycopeptides to beads functionalized with reactive groups generally leverages the reversible reaction between phenyl-boronic acid (PBA) derivatives and 1,2- or 1,3-diols present in sugar molecules. This results in the formation of a covalent bond between functionalized beads and glycopeptides at high pH, followed by elution of the glycopeptides at low pH.”

3. I am missing a bit some info on how much input material is needed for both their single shot or multidimensional strategies (i.e. how much protein from how many cells and how much tissue is needed/optimal). This would be an important consideration for anyone wanting to try this.

We thank the reviewer for his comment. In our hands, the number of identification reaches a plateau at 1mg of input for a single shot analysis, but of course this will depend on the sample type (cell-line, tissue, organism,...) and the instrumentation (i.e. an Astral will reach the same number of identification than an Orbitrap Exploris or tribrid with less input). Regarding the fractionated samples, in this study we enriched 2mg of peptides per replicate before TMT labeling and pooling (the total input before fractionation being then equal to 36 mg for the tissue and microbiome experiment, or 16 mg of input for the SPP). For the label free samples fractionated with PGC, we enriched 30 mg of mouse brain peptides.

Action taken: We added the inputs used in the method section, as followed: “For single-shot measurements of glycopeptides, we found that the number of identification reached a plateau after 1 mg of input. For the PGC fractionated sample, we enriched 2 mg of peptide per TMT channel, or 30 mg of peptides in the case of label-free PGC experiments.”

4. In methods they mention that identified O-linked peptides containing a N-linked motif were removed. It would be interesting to know the rate at which rate this occurs.

We thank the reviewer for his comment. We found that it is a critical step, as the search engine will search separately for either N- or O-linked glycopeptides.

As in the samples analyzed the majority of glycopeptides are N-glycopeptides, and given the fact that several N- and O-glycan compositions have the same molecular weight, by searching for O-glycopeptides, MSFragger will identify many candidates O-glycopeptides based on the peptide backbone fragments and offset mass which are

in reality N-glycopeptides (all containing a serine or threonine in the sequence), with the same peptide sequence having a N-glycan composition of similar mass.

By removing all O-glycopeptides having a N-X-S/T sequon in the peptide sequence (or N-K/R at the peptide C-term followed by S/T in the protein sequence), we ensure that this is not the case. We chose this conservative strategy to minimize the number of misidentifications. By doing so, 85% of the O-glycopeptides identified by MSFragger were filtered out.

Reviewer #1:

Remarks to the Author:

This manuscript describes the development and application of Deep Quantitative Glycoprofiling (DQGlyco), a high-throughput sample preparation method to enrich glycopeptides combined with mass spectrometry analysis. DQGlyco is applied to the mouse brain tissue where an impressive number of unique glycopeptides are identified. Analysis is performed on this dataset to determine correlation between glycosylation microheterogeneity and protein structure. Tissue-specificity of glycosylation is also examined. Lastly mouse brains with different gut microbiota conditions are analyzed and claims are made that different microbiome conditions result in extensive remodeling of the brain glycoproteome.

Overall this is an interesting study with an impressive depth of analysis. The data set is also likely to be useful to the scientific community. However, my major criticism of the manuscript is that it covers several large areas (method development, protein structure and glycosylation, glycosylation differences between tissues, microbiome effect on mouse brain tissue), but is lacking substantial depth in any given area. The manuscript is largely descriptive and it is somewhat difficult to follow what the main point is. Some additional specific comments:

1) The first two paragraphs of the results section are out of place and don't belong in the Results section.

We thank the reviewer for this comment and moved the two paragraphs in the introduction section.

Action taken: Moved the two paragraphs to the introduction section.

2) The description of the AI analysis for protein structure and glycosylation not well described. It was difficult to follow the analysis and findings.

We thank the reviewer for this comment and described the AI analysis in more detail, both in the main text and in the method section.

Action taken: We expanded the results paragraph as well as the methods section to provide a more detailed explanation of the components and variables used in this analysis.

3) The claim that gut microbiome changes glycosylation in brain proteins is a substantial stretch.

While we agree that the mechanisms underlying the connection between gut microbiome compositions and glycoproteome alterations remain to be investigated, we think that given our experimental design (6 biological replicates in each group: germ-free, monocolonized, and 8-members colonized) the data shows that a clear remodeling of the brain glycoproteome occurs upon gut colonization. Nevertheless, following editorial discussion, we changed the title and focus of the manuscript to reflect the technical nature of our work.

Action taken: We changed the title of the manuscript, and added new data, as detailed in the response to point 7 of the reviewer.

4) Cell line authentication was not described.

The HeLaK cell line used in this study has been authenticated before (doi 10.1534/g3.113.005777), while the HEK293T was obtained from ATCC and the Expi293 cells were purchased from Thermo.

Action taken: We added this information in the methods section.

5) Scores for glycopeptide ID's in supplementary tables should be provided.

We thank the reviewer for this comment and added the MSFragger hyperscore of glycopeptides in the supplementary tables.

Action taken: MSFragger hyperscores were added to the supplementary tables.

6) The classification of glycopeptides is not clear. Is something double counted if it is both sialylated and fucosylated for example?

We thank the reviewer for this comment and apologize if the classification description was not clear. We used a hierarchical classification to group glycoforms in different glycan classes, as follow (from the method section): (i) phospho, containing a phosphate group, (ii) sialylated, containing a NeuAc or NeuGc sugar, (iii) fucosylated, containing fucose, (iv) high mannose, combining a HexNAc(2) core with between four and 12 hexose molecules, (v) paucimannose, combining a HexNAc(2) core with between one and three hexose molecules, (vi) small, composed of only a core with less

than three HexNAc molecules and (vii) complex/hybrid, the rest. For O-glycosylation data, categories were defined as follows (hierarchical classification): (i) O-phospho, containing a phosphate group, (ii) O-sialylated, containing a NeuAc or NeuGc sugar, (iii) O-fucosylated, containing fucose, (iv) O-sulfated, containing a sulfation, (v) O-GlcNAc, for glycans containing a single HexNAc, (vi) O-hexose, composed of a core with less than three HexNAc molecules and at least one hexose molecule (vii) O-hybrid, the rest. This means that a glycoform being both sialylated and fucosylated will be part of the sialylated glycan class, but not counted as a fucosylated glycoform.

Action taken: The last sentence above was added to the method section.

7) The number of biological and technical replicates for each experiment is not clear.

We thank the reviewer for this comment. We added the number of biological and technical replicates in the method section or figures legends.

Action taken: The number of replicates was added in the method section and/or in figures legends.

8) While substantially more glycopeptides are identified in this study compared to previous studies, glycopeptides in the mouse brain have been studied quite extensively, thereby limiting the novelty and making this study somewhat incremental.

We agree that the mouse brain glycoproteome has been profiled before, and one of the reasons why we selected this tissue was to benchmark our approach. By doing so, we were able to clearly demonstrate that our approach improved coverage by 25-fold. We would also like to emphasize that all previous studies in the mouse brain have been qualitative, and that our strategy enables for the first time the in-depth quantitative profiling of the brain glycoproteome dynamics in response to a controlled perturbation in vivo. Furthermore, we did not only profile the brain glycoproteome, but also other tissues (mouse liver and kidney) and cell lines (HeLa, HEK293T, HEK293f).

Action taken: Following the request of the editor, we also added additional data. We first profiled the modulation of the glycoproteome upon treatment with a fucosylation inhibitor (2FF), revealing site- and glycoform-specific regulation of glycosylation. In addition, we treated intact living HEK293f cells with either PNGase F or proteinase K to characterize the mature, surface exposed glycoforms. Finally, we introduced the concept of biophysical glycoproteomics, an approach enabling the determination of the solubility (here the differential solubility of glycoforms in a non-denaturing detergent - NP40 - vs a strong denaturing detergent - SDS) of different glycoforms in vivo. Such an

approach was recently developed in our lab to quantify the impact of phosphorylation on protein solubility and decipher phosphorylation-mediated mechanisms of protein phase separation (DOI 10.1038/s41589-022-01062-y).

As a change of solubility can be linked to a change in protein state, this constitutes the first effort to systematically link differences of microheterogeneity with alterations in protein states (i.e. activity, localization or interactions). While SPP does not allow to obtain a definitive answer regarding the specific function of a given glycoform, it can help to pinpoint functionally important glycosylation events on a proteome-wide scale and inform us on the potential role of this glycoform on protein function (e.g. if a glycoform promotes or prevents interactions with ECM, this could be reflected in an increase or decrease of solubility of this glycoform, respectively). Overall, this method revealed extensive differences of solubility between glycoforms, supporting at a proteome-wide scale previous anecdotal evidence that microheterogeneity likely influences protein function.

We copy here the new paragraphs from the main text, along with the corresponding main figures:

“Time course of fucosylation inhibition reveals site- and glycoform-specific regulation of glycosylation

We first used our quantitative workflow to study the inhibition of fucosylation over time following treatment with 2-fluorofucose (2FF, Figure 4A). 2FF readily diffuses into cells, where it is metabolized into the donor substrate GDP-2FF that blocks fucosyltransferase activity. Moreover, the intracellular accumulation of GDP-2FF has been shown to inhibit the production of endogenous GDP-fucose leading to a general suppression of protein fucosylation, which inhibits several biological processes such as tumor growth, adhesion and metastasis. The profiling of the glycoproteome over three days of treatment of HEK293f cells revealed a gradual decrease of fucosylation, alongside extensive remodeling across all glycan classes (Figure 4B-D). Notably, the downregulation of fucosylation occurred at varying rates, which we grouped into 3 main clusters (Figure 4E), revealing a widespread site-specific and glycoform-specific modulation. This is illustrated by fucosylated glycoforms of the same protein belonging to different downregulated clusters (Figure 4F), meaning that such differences in kinetics of fucosylation downregulation cannot be attributed to differences in protein turnover rates. A clear example of this observation is the modulation of glycoforms on the proteins MRC2 and NRCAM, where fucosylated glycoforms on different sites follow distinct kinetics (Figure 4G,H).

It is generally accepted that changes in protein glycosylation occur mainly via modulation of glycoenzymes abundance, localization or activity. In such cases, similar trends would be expected at the glycoproteome level, especially for glycosites within the same protein. This site- and glycoform-specific modulation could be explained by glycoform-specific turnover rate (i.e. glycoforms are subjected to endocytosis and recycled at different rates), or by more complex modulation at the fucosyltransferase level.”

Determination of cell surface-exposed glycoforms

N-glycans are added to asparagine side chains of proteins as high-mannose in the ER before being further processed in the Golgi. Once the cells are lysed, it becomes impossible to differentiate mature, functional glycoforms (i.e. present on proteins which made it to their final destination) from immature glycoforms (i.e. intermediate species on

proteins going through the secretory pathway). Due to the complexity of the glycosylation machinery, it is not possible to predict the composition of mature glycans and it is postulated that glycoforms of any compositions or structures could potentially be surface-exposed (Schjoldager et al., 2020). Significant differences have been observed between cell surface and intracellular glycans (de Haan et al., 2023), but obtaining a comprehensive system-wide, site-specific overview of mature glycoforms has been challenging so far and the compositions or the proportion of mature glycoforms in a given glycoproteomics experiment remains an open question in the field. Here, we applied our quantitative workflow to intact living human HEK293 cells subjected to either PNGase F treatment for 30 mins (glycosidase targeting N-glycans) or proteinase K for 1 min (broad specificity enzyme used in limited proteolysis experiments (Schopper et al., 2017)). Our rationale was that only cell surface-exposed glycoforms would be affected, while glycoforms present in intracellular organelles (i.e. in the ER or Golgi) would be protected, resulting in unchanged abundance (Figure 5A). As a quality control, we demonstrated that the vast majority of proteins affected by the enzymes were annotated as belonging to the plasma membrane (Figure 5B).

While some sites remained inaccessible and were not affected by the enzymatic treatments, we identified 3,990 glycopeptides on 826 sites and 513 proteins showing significant changes in abundance upon treatment (Supplementary Data 4, Figure S7A,B). To enhance confidence we decided to consider for subsequent analysis only sites for which at least 2 glycoforms were affected (3,113 glycopeptides on 299 sites and 168 proteins, Figure S6B). For example, on the zinc transporter ZIP6, the N241 site was consistently affected by both enzymatic treatment, while the N67 and N266 sites remained unaltered (Figure 5C). Regarding the site N67, this can be explained by the fact that this site is located on a propeptide, the N-terminal cleavage triggering the location of ZIP6 to the plasma membrane (Hogstrand et al., 2013). Similarly, the site N34 of Desmocollin-2 (DSC2) remained affected, because localized on the propeptide region whereas glycans on the N392, N546 and N629 sites were revealed to be surface-exposed (Figure 5D).

Analyzing these accessible sites, we could demonstrate that high-mannose glycoforms are in general less surface exposed, although this tendency seems to be site-dependent. Conversely, most of the fucosylated/sialylated glycoforms are, at least partially, surface-exposed, aligning with recent literature findings (de Haan et al., 2023) (Figure 5E). Sites exhibiting at least two high mannose surface-exposed glycoforms were in general less accessible, supporting the notion that site accessibility may influence glycan maturation (Figure S7C). We could in addition identify glycan compositions predominantly characterized as surface-exposed in our dataset (Figure S7D). For both enzymes, no discernible bias towards preferential cleavage of given

glycan compositions or molecular weight was observed (Figure S7E/F). This suggests that the measured fold change upon treatment likely reflects the degree of surface exposition of a given glycoform.

Notably, in instances where sites were impacted by both enzymatic treatments, we observed good correlation between the two experiments, both at the glycopeptide level (fold change of a given glycopeptide between treatment and control conditions) and glycan compositions level (number of times a given glycan composition was observed on a glycopeptide significantly changing upon treatment) (Figure 5F,G). PNGase and proteinase K were also found to be complementary (Figure S7G, Figure S7H), with proteinase K effectively targeting less accessible glycosites (Figure S7I), likely due to the fact that cleavage at any amino acids in the vicinity of the glycosite will result in a change in glycopeptide abundance while PNGase needs access to the glycosylated asparagine. In summary, we introduce a method for determining surface-exposed glycans, and provide a system-wide, site-specific view of mature and immature glycans in a human cell line, enhancing our understanding of the biological functions of glycosylation.

Biophysical glycoproteomics

Modulation of site microheterogeneity has been shown to regulate many cellular processes, but determining its functional impact remains challenging. This is mainly due to the fact that only few glycosylation events have a functional annotation, that it is currently impossible to selectively alter a given glycoform on a given glycosite both in vitro and in vivo, and that one cannot perform mutagenesis to assess functionality of glycosite microheterogeneity. The quantitative glycoproteomics workflow developed in this study enables us to systematically determine biophysical properties of glycoforms in vivo by assessing the solubility of more than 30,000 glycoforms using Solubility Proteome Profiling (SPP, (Sridharan et al., 2019), Supplementary Data 5). Here, the differential solubility of a given proteoform in a mild detergent - NP40 - and a strong denaturing detergent - SDS - is determined (Figure XA), an approach previously used to

systematically study the functional relevance of protein phosphorylation (Sridharan et al., 2022). Although this approach does not directly reveal the impact of a change in glycosylation on specific protein function or activity, variations in protein solubility due to a specific glycosylation likely suggest differences in protein state, such as localization or interaction.

Application of glyco-SPP showed that high mannose glycoforms tend to be more soluble than more processed glycans (Figure 6B), likely due to the fact that mature glycoforms are mostly less soluble than intracellular glycoforms, as a result of interactions with the extracellular matrix. This is particularly well illustrated in the case of the cadherin 13 protein, (CDH13), involved in cell adhesion: glycoforms on the site N86, belonging to the propeptide, are more soluble than glycoforms on other sites belonging to the mature proteoforms (Figure XC). In general, distinct glycoforms on the same site exhibit varied solubility, suggesting that difference in microheterogeneity can be linked to changes in protein activity, localization or interactions. Notably, we observed several instances of site-specific impact of glycosylation on solubility, wherein most glycosylation events on a given site tend to influence protein solubility in the same direction, as illustrated for the site N94 of Thy-1 or the sites N73 and N80 of CD47 (Figure XD,E).”

Reviewer #2:

Remarks to the Author:

This manuscript by Potel et al. describes a new workflow for glycoproteomics, combining an improved method for glycopeptide enrichment with optimized deep fractionation and mass spectrometric acquisition methods to profile an order of

magnitude more glycopeptides than previous studies in similar sample types. The authors then use this exceptionally deep glycoproteomic profiling to provide a variety of biological insights, providing an excellent illustration of the potential of glycoproteomics. The manuscript is compelling and clearly supports these rather exceptional claims with clear and convincing evidence throughout. Other than a few minor clarifications and areas for additional discussion, detailed below, I see no barriers to publication.

Specific comments:

1. The improvement in glycopeptides identified from the PBA enrichment and optimized MS1 scan range, while impressive, appear to be smaller than the gain from the PGC fractionation in Fig 1. This is particularly notable given that the previous studies being compared (Riley et al, etc.) also used deep fractionation, but via high pH C18 rather than PGC. Some additional discussion of the interplay of these factors would be very valuable to clarify whether similarly deep profiling could be accomplished by PGC fractionation with traditional enrichment methods (HILIC, lectins) and/or without the nucleic acid precipitation, or if it is really the combination of all of these improvements together that is necessary to achieve the order of magnitude increase in glycopeptides identified. Additionally, some discussion of any challenges or key considerations for implementing this method in other labs would provide a valuable perspective on the potential of the method.

We thank the reviewer for the positive feedback. We agree that the off-line fractionation of glycopeptides using a PGC column provides the most important gain in identification. We are confident that a PGC fractionation of a glycopeptide sample following a HILIC or lectin-based enrichment would also boost the number of identification significantly, albeit to a lower extent than with the PBA enrichment. We show in Figure 1C that PBA-based enrichment achieves high specificity thanks to stringent washes following covalent capture by PBA functionalized beads (ca 90% of peptides are glycopeptides), which is not possible with traditional enrichment methods. This is illustrated in the figure S1F, where we show that a single shot measurement of a non fractionated mouse brain glycopeptide sample following a PBA enrichment provides more than double the identification numbers when compared to off-line fractionated samples obtained after HILIC (Liu et al.) and lectin-based (Riley et al.) enrichments. In addition, it appears that the PBA based enrichment method is less biased than lectin-based or HILIC enrichment techniques towards certain glycan composition or molecular weight. Indeed, the different lectins are biased towards certain types of glycan composition (ConA is for example exhibiting a bias towards high mannose glycans), and HILIC enrichment will favor the enrichment of highly hydrophilic glycans (negatively charged and high molecular weight glycans).

Overall, regarding the interplay between the different optimized parameters, we believe

that the improvement of the MS acquisition methods, as well as the PGC fractionation would enhance all glycoproteomics enrichment platforms. However, the nucleic acid precipitation should mostly be beneficial for sample preparation prior to PBA enrichment. Indeed, PBA will form a covalent bond with diols present in nucleic acids, but lectin, and to a lesser extent HILIC enrichment should remain largely unaffected by the presence of nucleic acids in the sample.

Action taken: We added the previous paragraph in the discussion.

2. The discussion of peptides bearing O-GlcNAc modifications should clarify explicitly that a single HexNAc can be either O-GlcNAc or O-GalNAc, and while the oxonium ion ratios in Fig S2G clearly show a difference in trend between intra- and extracellular proteins, it is not as clear-cut as described in the text given that median ratio for extracellular proteins remains well below 1. Calling these putative O-GlcNAc modifications would be more accurate, and the limitations of oxonium ion analysis (namely, co-fragmentation) should be mentioned.

We thank the reviewer for this comment and modified the nomenclature as suggested.

Action taken: The term O-GlcNAc was replaced by O-HexNAc in the text and figures.

3. The choice to use a more liberal glycan q-value threshold of 0.05 is not unreasonable, but the text should note that the confidence in the glycan compositions identified is lower than the peptides, given the higher chosen threshold. A comparison of the results for one analysis between setting a threshold of 0.01 and 0.05 could be a useful addition to the supporting information to inform the reader as to the impact of this choice.

We thank the reviewer for this comment. We discussed extensively with the developers of MSFragger glyco regarding the q-value cutoff, and agreed to choose 0.05, a cutoff previously used in a recent publication from the same lab (Bedran et al., <https://doi.org/10.1038/s41467-023-39270-2>). According to the developers, the glycan q-values serves more as a guideline than a hard cutoff, and is particularly useful to filter out glycan compositions with a q-value = 1, which are likely to be misassigned. This is underlined by the skewed distribution of the glycan q-values, which differs significantly from typical peptides q-values distribution observed in proteomics. This is illustrated by the q-value distribution plotted below (from a fractionated mouse brain sample, single high energy HCD fragmentation), in which more than 230,000 PSMs have the exact same glycan q-value of 0.0234599165. We think that the assessment of glycan composition assignment confidence is an important issue, and hope that the extensive datasets generated (more than 1.5 millions glycoPSMs, from label-free and

TMT-labeled samples, fragmented with single high energy or stepped HCD collision energies) will aid in addressing this challenge and developing appropriate scoring models.

Action taken: We added in the first result paragraph of the main text that the different thresholds used for N-glycopeptides identification are: “1% FDR at the N-glycopeptide level, glycan q-values ≤ 0.05 ”.

Reviewer #3:

Remarks to the Author:

Potel et al. present a study focused on deep quantitative glycoproteomics including a number of technical advances with an application related to microbiome induced changes in mouse brain. The technical advances include a glycopeptide enrichment based on existing covalent chemistry (PBA) combined with highthroughput plate-based sample processing methods that reduces contaminating species such as nucleic acids. While the enrichment methodology already contains several improvements the authors then employ a multidimensional approach using porous graphitic carbon which substantially improves glycopeptide coverage. The authors benchmark by comparison to other recent papers demonstrating the improvements, and importantly also show that their method demonstrates less bias for specific glycopeptide/protein classes than existing methods. They assess site-level heterogeneity and protein level heterogeneity as well as setting glycoform variation in the context of predicted structural features including a machine learning model to predict site level heterogeneity based on structural features. Finally, the authors integrate isobaric labelling with their multidimensional method, including labelling adaptations to minimize label requirements, to enable quantitative comparisons. The strategy is then applied to microbiome perturbation experiment in mouse. They show first that there strong distinction in tissue

level specificity of the glycoproteome, and further differences in specific glycoforms in brains of mice depending on microbiome status.

In my view this manuscript is very strong and represents a very substantial improvement over the state of the art in glycoproteomics methodology. The increase in depth achieved, and removal of systematic bias for glycopeptide coverage, by very careful and systematic optimization of purification, sample prep, and MS methodology is fairly remarkable. I think the manuscript is very logical and well written, well illustrated, and packs in a large body of work that clearly shows the progression of concepts. The conclusions are surrounding micro- and macro-heterogeneity (also related to structure), tissue specificity, and effects of microbiome on brain are novel and striking. All data and code are made available in appropriate repositories, and results are explorable via a web app. My view is that this will have an immediate and strong impact on the field. As such, I think there is rather little to complain about and I suggest that the manuscript should be accepted in short order. I have put a few minor clarifying comments/questions below.

1. One technical question would be whether anything semi-quantitative can be said about the distribution of glycoforms at the site level. Given that the most extreme case they observe 600+ glycoforms at a single site, can anything be said about whether any forms are more common (akin to major/minor isoform is splice variants). While the response factor for a given peptide sequence dependent on the glycoform it harbours will presumably change, but perhaps a comparison of intensities would be informative. Is anything known about this issue? Could this be summarized at glycoform class level?

We thank the reviewer for the positive feedback. We agree that information about stoichiometry is important to understand biological implications of observed glycoform alterations. However, we can here only perform relative quantification, and using MS1 or summed TMT intensities to compare glycoform abundance would only be valid if the glycan composition does not impact ionization efficiency. Based on the general MS1 intensities of the different glycan classes, one can for example clearly see that the presence of sialylated glycan negatively affects ionization efficiency in positive mode, likely due to presence of negative charges (MS1 glycopeptide intensity per glycan class compared to high mannose glycopeptide intensities):

Nevertheless, using MS1 intensities to calculate glycoform fractional intensity (ratio of glycoform intensity over the sum of all glycoforms intensities on a given site) revealed that most glycoforms have a low fractional intensity:

When focusing on the glycoform having the highest fractional intensity per site (here only the sites with at least 10 glycoforms were considered), it becomes clear that glycosylation of a specific site is rarely dominated by a glycoform of high stoichiometry:

Action taken: We added the discussion above to the main text, in the “Deep mouse brain glycoproteome analysis” result section (figures in Figure S2).

2. The description of the methodological advances in the first results section could be slightly more easy to digest if revised elaborating further all steps. While its fine to leave majority of details in the methods section, I think it would be more understandable if conceptual aspects were more completely stated (i.e. a key advance is the removal of nucleic acids by precipitation but the details of how this is done is breezed over, enrichment by covalent coupling to PBA is described but not method for elution, etc). My experience on reading was to have to dig around in methods several times to understand the complete workflow conceptually so a revision of this section for conceptual clarity might help the less initiated reader.

We thank the reviewer for this suggestion, and added additional information in the main text regarding the nucleic acid precipitation and glycopeptide elution parts.

Action taken: The main text was modified as follow:

“To address this, we optimized the sample lysis buffer by incorporating high concentration of chaotropic salts and organic solvent to induce nucleic acid precipitation while proteins remain in solution. Nucleic acid aggregates were then filtered out on

96-well filter plates, followed by protein precipitation by further increasing organic solvent concentration prior to tryptic enzymatic digestion.”

“Chemical coupling of glycopeptides to beads functionalized with reactive groups generally leverages the reversible reaction between phenyl-boronic acid (PBA) derivatives and 1,2- or 1,3-diols present in sugar molecules. This results in the formation of a covalent bond between functionalized beads and glycopeptides at high pH, followed by elution of the glycopeptides at low pH.”

3. I am missing a bit some info on how much input material is needed for both their single shot or multidimensional strategies (i.e. how much protein from how many cells and how much tissue is needed/optimal). This would be an important consideration for anyone wanting to try this.

We thank the reviewer for his comment. In our hands, the number of identification reaches a plateau at 1mg of input for a single shot analysis, but of course this will depend on the sample type (cell-line, tissue, organism,...) and the instrumentation (i.e. an Astral will reach the same number of identification than an Orbitrap Exploris or tribrid with less input). Regarding the fractionated samples, in this study we enriched 2mg of peptides per replicate before TMT labeling and pooling (the total input before fractionation being then equal to 36 mg for the tissue and microbiome experiment, or 16 mg of input for the SPP). For the label free samples fractionated with PGC, we enriched 30 mg of mouse brain peptides.

Action taken: We added the inputs used in the method section, as followed: “For single-shot measurements of glycopeptides, we found that the number of identification reached a plateau after 1 mg of input. For the PGC fractionated sample, we enriched 2 mg of peptide per TMT channel, or 30 mg of peptides in the case of label-free PGC experiments.”

4. In methods they mention that identified O-linked peptides containing a N-linked motif were removed. It would be interesting to know the rate at which rate this occurs.

We thank the reviewer for his comment. We found that it is a critical step, as the search engine will search separately for either N- or O-linked glycopeptides.

As in the samples analyzed the majority of glycopeptides are N-glycopeptides, and given the fact that several N- and O-glycan compositions have the same molecular weight, by searching for O-glycopeptides, MSFragger will identify many candidates O-glycopeptides based on the peptide backbone fragments and offset mass which are

in reality N-glycopeptides (all containing a serine or threonine in the sequence), with the same peptide sequence having a N-glycan composition of similar mass.

By removing all O-glycopeptides having a N-X-S/T sequon in the peptide sequence (or N-K/R at the peptide C-term followed by S/T in the protein sequence), we ensure that this is not the case. We chose this conservative strategy to minimize the number of misidentifications. By doing so, 85% of the O-glycopeptides identified by MSFragger were filtered out.